# Global determinants of insect mitochondrial genetic diversity

Connor M. French [1,2] ✉, Laura D. Bertola [1,3], Ana C. Carnaval[1,2], Evan P. Economo[4], Jamie M. Kass [4,15], David J. Lohman[1,2,5], Katharine A. Marske [6], Rudolf Meier [7,8], Isaac Overcast [2,9,10], Andrew J. Rominger [11,12], Phillip P. A. Staniczenko [13] & Michael J. Hickerson [1,2,14]

Understanding global patterns of genetic diversity is essential for describing, monitoring, and preserving life on Earth. To date, efforts to map macrogenetic patterns have been restricted to vertebrates, which comprise only a small fraction of Earth's biodiversity. Here, we construct a global map of predicted insect mitochondrial genetic diversity from cytochrome c oxidase subunit 1 sequences, derived from open data. We calculate the mitochondrial genetic diversity mean and genetic diversity evenness of insect assemblages across the globe, identify their environmental correlates, and make predictions of mitochondrial genetic diversity levels in unsampled areas based on environmental data. Using a large single-locus genetic dataset of over 2 million globally distributed and georeferenced mtDNA sequences, we find that mitochondrial genetic diversity evenness follows a quadratic latitudinal gradient peaking in the subtropics. Both mitochondrial genetic diversity mean and evenness positively correlate with seasonally hot temperatures, as well as climate stability since the last glacial maximum. Our models explain 27.9% and 24.0% of the observed variation in mitochondrial genetic diversity mean and evenness in insects, respectively, making an important step towards understanding global biodiversity patterns in the most diverse animal taxon.

Resolving global patterns of biodiversity is essential for understanding how life is distributed across the world, and where it is most important to protect it. As yet, global-scale assessments have largely focused on species richness[1], phylogenetic diversity[2,3], species abundances[4,5], and functional trait diversity[6,7]. These macroecological metrics have long been used to inform conservation and gain insights into mechanisms underlying eco-evolutionary patterns.

[1]Biology Department, City College of New York, New York, NY, USA. [2]Biology Ph.D. Program, Graduate Center, City University of New York, New York, NY, USA. [3]Section for Computational and RNA Biology, Department of Biology, University of Copenhagen, Copenhagen N 2200, Denmark. [4]Biodiversity and Biocomplexity Unit, Okinawa Institute of Science and Technology Graduate University, Onna, Okinawa, Japan. [5]Entomology Section, National Museum of Natural History, Manila, Philippines. [6]School of Biological Sciences, University of Oklahoma, Norman, OK, USA. [7]Institut für Biologie, Humboldt-Universität zu Berlin, Berlin, Germany. [8]Center for Integrative Biodiversity Discovery, Leibniz Institute for Evolution and Biodiversity Science, Museum für Naturkunde Berlin, Berlin, Germany. [9]Institut de Biologie de l'Ecole Normale Superieure, Paris, France. [10]Department of Vertebrate Zoology, American Museum of Natural History, New York, NY, USA. [11]School of Biology and Ecology, University of Maine, Orono, ME, USA. [12]Maine Center for Genetics in the Environment, University of Maine, Orono, ME, USA. [13]Department of Biology, Brooklyn College, Brooklyn, NY, USA. [14]Division of Invertebrate Zoology, American Museum of Natural History, New York, NY, USA. [15]Present address: Macroecology Laboratory, Graduate School of Life Sciences, Tohoku University, Sendai, Miyagi, Japan. ✉e-mail: cfrench@gradcenter.cuny.edu

Large-scale georeferenced DNA barcode surveys have great potential beyond their original use and are increasingly used in global studies of biodiversity[8,9]. In addition to accelerating taxonomy[10], DNA barcoding has become a tool in diagnosing conservation status and, more generally, better understanding ecology, evolution, and biogeography[11–13], but see ref. 14. More recently, DNA barcode surveys provide data for the emerging field of "macrogenetics"[15,16], which has the potential to become one of the many approaches used to monitor and protect genetic diversity[17,18]. Class III macrogenetic studies, as defined by Leigh et al.[19], summarize the geographic distribution of intraspecific genetic variation of a taxonomic group across broad geographic scales and use data from large numbers of species aggregated from public repositories. Typically used to find correlated environmental variables that predict this critical component of biodiversity[19,20], previous class III macrogenetic studies have focused on vertebrate groups, uncovering links between global patterns in intraspecific mitochondrial genetic diversity, species richness, and phylogenetic diversity[21,22], while documenting latitudinal gradients[23–25]. They have provided mixed support for the influence of human disturbance on mitochondrial genetic diversity[21,23,26,27], and suggest that climate stability[21,27] and species' range sizes[20,28] affect intraspecific mitochondrial genetic diversity on a global scale.

This bias in macrogenetic studies towards vertebrates leaves undocumented the bulk of the planet's animal biodiversity: insects. Insects are vital for maintaining critical ecosystem services and functions[29,30], yet existing insect macrogenetic studies have been restricted to regional scales due to the immense effort required to collect, identify, and sequence such a species-rich group[31–36]. Studies on the resilience of insect communities to global change[37,38], including biological invasions[39,40], habitat conversion[41], and climate change[42] arrive at conflicting conclusions. Moreover, comprehensive knowledge of species diversity, distributions, and population dynamics is largely lacking for large insect groups[43,44]. These constraints on understanding broad-scale insect biodiversity patterns point to a need for systematic global data syntheses[45,46].

Despite their known limitations[14,47,48], DNA barcoding and metabarcoding address the severe bottleneck posed by species identification through conventional morphological methods[46,49,50], and they are viable approaches for rapid, large-scale, global quantification of insect biodiversity[14,47,48,51]. While being mindful of these limits in our study design and data[14,15,19,52,53], we present a global class III macrogenetic analysis of insect intraspecific mitochondrial DNA patterns. Our study is especially timely given increasing evidence that many insect taxa may be in global decline with respect to occurrence, local richness, abundance, and biomass[38,54–60].

Nearly all class III macrogenetic studies of animal taxa are based on mitochondrial DNA (mtDNA) sequence data, by far the most abundant type of sequence data in public repositories[15]. Therefore, it is important to note that while mtDNA has useful properties[12], mtDNA diversity patterns are impacted by multiple processes that distort the patterns shaped by phylogeographic demographic histories, including selection, coalescent variance, and mutation rate variation across taxa[52,61–64]. While taking these complex dynamics into consideration, sampling the mitochondrial genetic diversity of thousands of taxa per locale is a promising initial step towards understanding an important component of biodiversity at global spatial scales[10,12,65,66].

We do so here by using open data from the Barcode of Life Consortium database (BOLD), a rich source of single-locus mtDNA that links quality-controlled genetic data with georeferenced metadata[8]. Leveraging this resource, we compiled and analyzed the largest animal macrogenetic dataset ever assembled: 2,415,425 globally distributed and georeferenced mtDNA sequences (cytochrome c oxidase subunit 1; *COI*) for 98,417 operational taxonomic units (OTUs) within the class Insecta. We used these data to generate a global map of insect mitochondrial genetic variation using the commonly used genetic diversity mean (GDM) and a new measure we adapted from macroecology: genetic diversity evenness (GDE). While GDM describes the average genetic diversity (GD) among species sampled from a spatial unit of arbitrary size (i.e., two-dimensional area), GDE represents the evenness of the distribution of per-species GD measures for all focal taxa that co-occur in the same sampled area[67]. So far, macrogenetic studies have not quantified variation in per-species GD and are therefore unable to distinguish if the high GDM observed in a geographic region is due to high diversity within most community members, or to the coexistence of a few taxa with extremely high diversity and many low-GD taxa (Fig. 1; see "Methods"). Much like species abundances and the evenness of abundances within a community, GDM and GDE present broadly complementary information when considered together, with potential utility in discriminating among processes generating natural community assemblages[68].

After calculating these two metrics across all OTUs for each cell, we explore how they correlate with global environmental and geographic variables. If GDM and GDE follow classic global biodiversity trends, we expect GDM and GDE to decrease with latitude[69] and human disturbance[70], while increasing with climate stability[71]. Once these environmental correlates of intraspecific insect mitochondrial genetic diversity are identified based on information extracted from insect sampling efforts across the globe, we use them to predict patterns of insect GDM and GDE in undersampled regions. We discuss and interpret our results in the context of classic biogeographic patterns such as the latitudinal diversity gradients in insect species richness, average and variability in range sizes, and late Pleistocene climate cycles.

## Results

GDM and GDE were calculated from COI sequences of insects native to the 193 km × 193 km equal-area resolution raster grid cells in which they were sampled. To account for the potential impact of sampling biases, six thresholds for the minimum number of OTUs per grid cell (10, 25, 50, 100, 150, and 200 OTUs) were considered. These cells were distributed heterogeneously across the globe and on every continent except Antarctica ($N_{100}$ = 245, Fig. 2, Supplementary Fig. 1). Genetic diversity patterns and modeling results were mostly consistent across the minimum number of OTUs thresholds (Supplementary Fig. 1, Supplementary Methods). For conciseness, we highlight results for a minimum number of 100 OTUs, which balanced per-cell and global sampling magnitude with per-cell and global sampling variance (Supplementary Fig. 2). This threshold resulted in the lowest bias (slope near 1 and y-intercept near zero) and highest accuracy (high $R^2$, low root mean squared error) when predicting trained models to withheld test data (Supplementary Fig. 3). This filtering criterion led to a final dataset that included 2,415,425 *COI* sequences from 98,417 OTUs sampled across 245 globally distributed grid cells.

On average, each sampled cell included ten insect orders, 689 OTUs, and 9859 individuals. Regions with both high GDM and high GDE (above the 90th percentile) were found in eastern North America, the North American desert southwest, southern South America, southern Africa, and southwestern Australia (Supplementary Fig. 4b). Areas with the lowest values of both observed GDM and GDE were mostly distributed in northern North America and Europe (Supplementary Fig. 4d). GDM and GDE were significantly and positively correlated (spatially modified t-test, two-sided; $r_{100}$ = 0.612, $p_{100}$ < 0.001).

**Insect mitochondrial genetic diversity correlates with latitude**
Absolute latitude was not significantly linearly correlated with GDE ($p_{100}$ = 0.064), but squared latitude had a negative relationship with GDE across the globe at $\alpha$ = 0.05 (spatially modified t-test, two-sided; Fig. 2; Table 1; $r_{100}$ = −0.360; $p_{100}$ = 0.022). In contrast to GDE, neither absolute latitude ($p_{100}$ = 0.924) nor squared latitude had a significant correlation with GDM (Fig. 2; Table 1; $p_{100}$ = 0.767). When considering the top three most-sampled orders independently, we found that

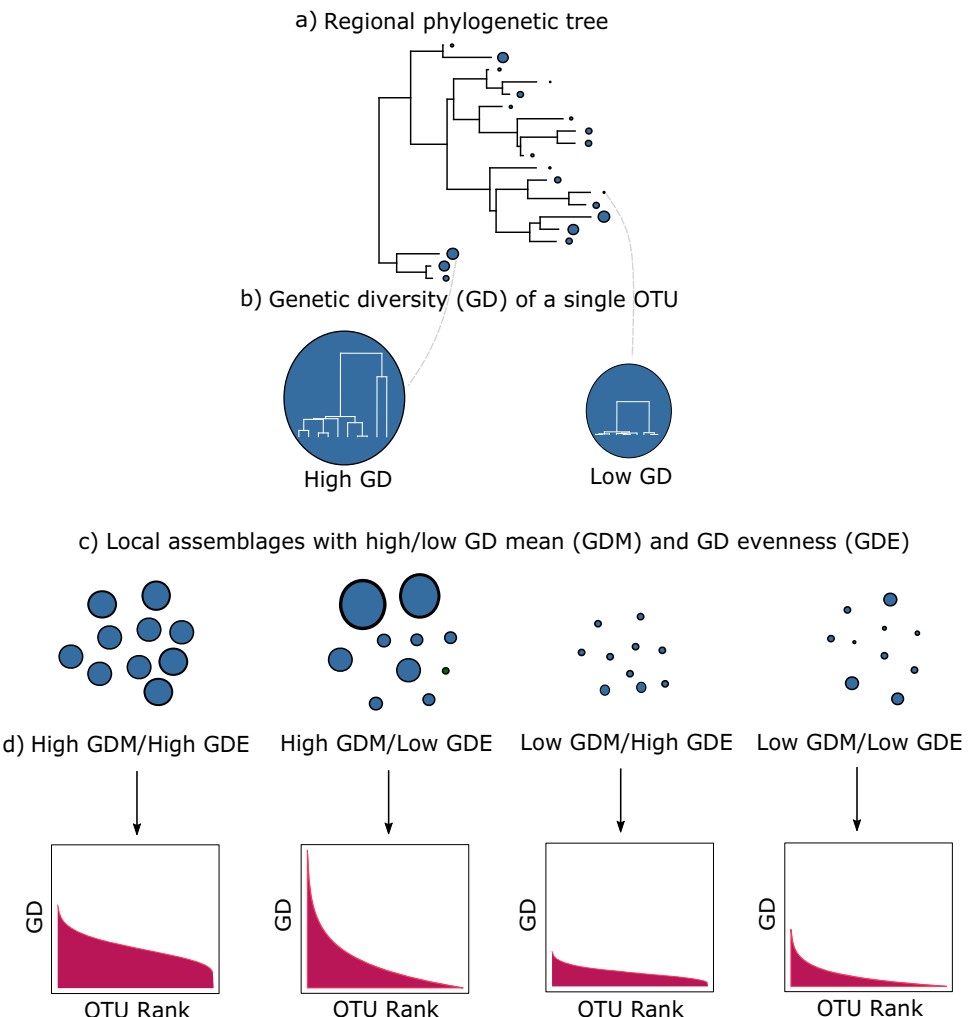

**Fig. 1 | Diagram illustrating genetic diversity mean (GDM) and genetic diversity evenness (GDE).** A local assemblage (**c**) is a set of operational taxonomic units (OTUs, analogous to species) sampled from a single grid cell that are a subset of a wider regional pool with evolutionary relationships shown in (**a**). OTUs have varying amounts of genetic diversity (GD), represented by blue circles with sizes corresponding to magnitude of GD. Longer branches among individuals within an OTU indicate a longer time to coalescence and therefore higher GD (**b**). Panel (**c**) illustrates four local assemblages sampled from four different grid cells from the same regional pool. The first local assemblage in (**c**) has high GDM and high GDE, represented by OTUs with high and similar GD and a corresponding relatively flat curve on the rank plot in (**d**). The second local assemblage in (**c**) has the same high GDM as the first assemblage in (**c**), but has lower GDE, indicated by dissimilar circle sizes and a steeper curve in the corresponding rank plot in (**d**). The third and fourth local assemblages in (**c**) have the same GDE as the first and second assemblages, respectively, but have lower GDM, indicated by the smaller circle sizes and lower height curves on the rank plots in (**d**). This illustrates the complementary nature of the two metrics, where GDM describes the average magnitude of GD in a local assemblage, while GDE describes the distribution of GD in that same local assemblage.

squared latitude had a negative relationship with GDE in Diptera ($r_{100} = -0.468$; $p_{100} = 0.001$) and Lepidoptera ($r_{100} = -0.360$; $p_{100} = 0.043$), but not in Hymenoptera ($p_{100} = 0.352$) (Fig. 3). GDM did not significantly vary with absolute latitude in any insect order (Fig. 3).

**Relationships between insect mitochondrial genetic diversity and the environment**

Higher GDE values were observed in areas that rarely freeze (Figs. 2d and 4c). To capture this relationship as a binary predictive variable, we divided the globe into areas that do or do not freeze, which is delineated by whether the long-term minimum temperature of the coldest month (MTCM) is above or below 0 °C[72]. We found that GDE is significantly higher in areas that do not freeze (spatially modified t-test; $r_{100} = 0.338$; $p_{100} = 0.013$), while GDM was not correlated with this binary metric ($P = 0.484$).

We found that GDM and GDE covary significantly with current and historical climate, and that both are positively correlated with

maximum temperature of the warmest month and climate stability variables (Figs. 2 and 4). We considered 49 environmental variables that could possibly structure the genetic diversity of insect assemblages (Supplementary Table 1). After removing strongly correlated variables ($r > 0.75$), we retained 11 environmental variables describing current climate, variables summarizing climate variation since the last glacial maximum (LGM, or "historical climate"), a habitat heterogeneity metric, a human habitat modification metric, and topographic variables. After removing additional variables that did not contribute substantial predictive power according to projection predictive variable selection (see Methods), we used Bayesian generalized linear mixed models (GLMMs) that account for spatial autocorrelation in the residuals to explain environmental relationships and make predictions[73]. The resulting GLMMs explained 27.9% of the training data variation in GDM (95% highest density interval (HDI): [14.6%, 40.7%]) and 24.0% of the training data variation in GDE (95% HDI: [10.1%, 39.0%]) (Fig. 4a, c). When projecting the models to withheld test

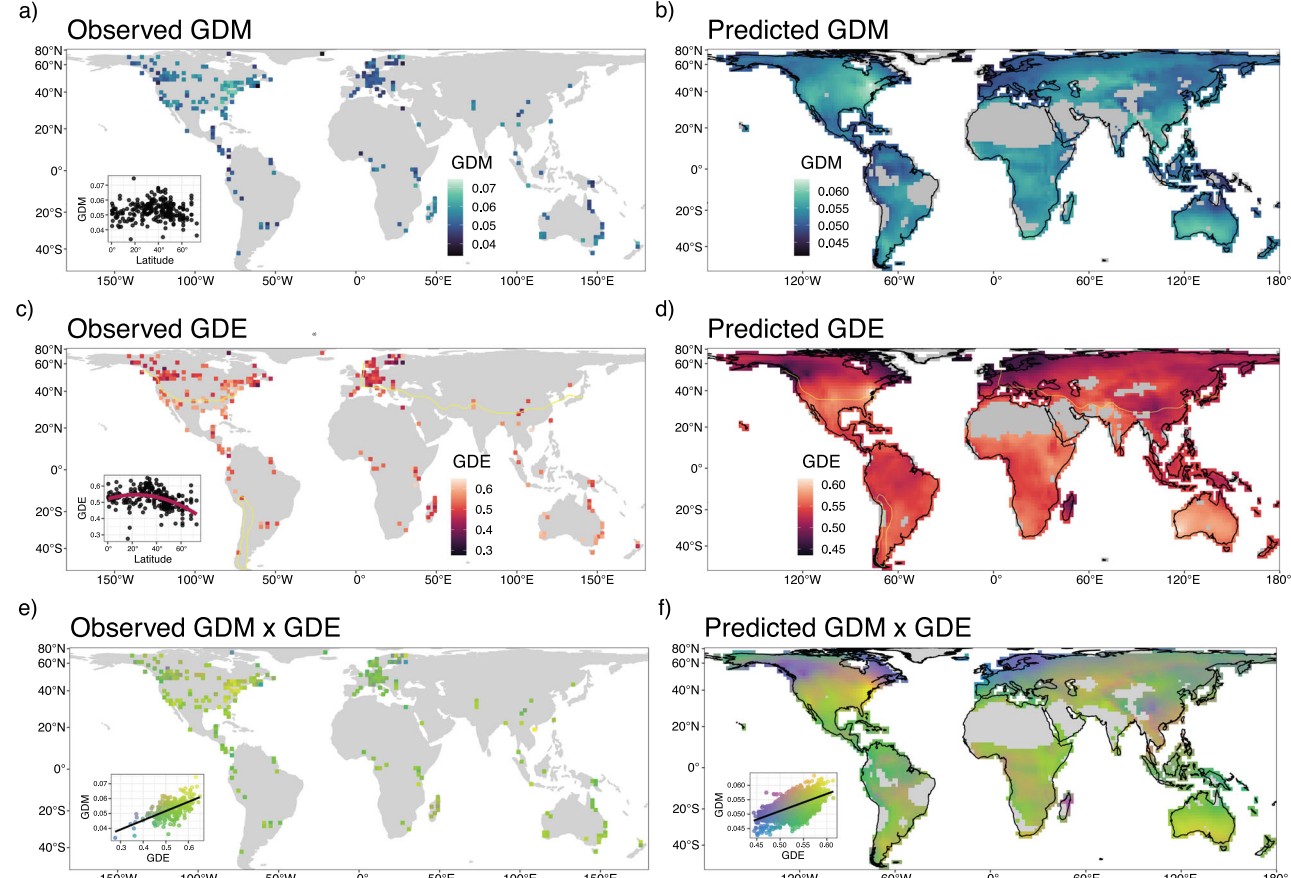

**Fig. 2 | Global maps of observed and predicted genetic diversity.** The observed (**a**, **c**, **e**) and projected (**b**, **d**, **e**) distributions of genetic diversity mean (GDM) (**a**, **b**), evenness (GDE) (**c**, **d**), and their composite (**e**, **f**) across the globe. Values for the projected maps were derived from a spatial Bayesian generalized linear mixed model with environmental predictor variables. For GDM (**b**), the best fit model included MTWM and precipitation seasonality, while for GDE (**d**), the best fit model included MTWM, temperature seasonality, and PWM. Latitudinal trends in GDM (**a**) and GDE (**c**) are included as insets of the observed maps, where latitude had no significant relationship with GDM and a negative quadratic relationship with GDE

(spatially modified t-test, Table 1). The yellow lines drawn across the maps of GDE (**c**, **d**) delineate areas that do or do not freeze, where areas north of the line and inside the polygon in South America have minimum temperatures that dip below 0 °C, and areas south of the line and outside the polygon have minimum temperatures that remain above 0 °C year-around. Areas that do not freeze on average have higher GDE than those that do freeze. We masked in gray areas with environments non-analogous to the environments used for modeling. MTWM maximum temperature of the warmest month, PWM precipitation of the wettest month. Source data are provided as a Source data file.

**Table 1 | Results for two-sided spatially modified t-test correlations between genetic diversity mean (GDM), genetic diversity evenness (GDE), and latitude**

| Model | Term | r | F-statistic | DOF | p-value |
|---|---|---|---|---|---|
| GDM ~ latitude | linear | −0.018 | 0.009 | 29.596 | 0.924 |
| GDM ~ latitude$^2$ | quadratic | 0.053 | 0.089 | 0.767 | 0.767 |
| GDE ~ latitude | linear | −0.280 | 3.617 | 42.562 | 0.064 |
| **GDE ~ latitude$^2$** | **quadratic** | **−0.360** | **5.730** | **38.550** | **0.022** |

Correlations were inferred for both linear and quadratic relationships. The bolded row indicates the relationship that is significant at α = 0.05.
r Pearson's correlation coefficient, DOF degrees of freedom.

data (75% training, 25% testing, stratified by continent), we found that predictions for all OTU thresholds were strongly correlated with observed data for GDE (Supplementary Fig. 3; $R^2_{100} = 0.515$, slope$_{100} = 0.961$, y-intercept$_{100} = 0.019$) and GDM (Supplementary Fig. 3; $R^2_{100} = 0.510$, slope$_{100} = 1.114$, y-intercept$_{100} = -0.007$).

The GLMM for GDM included eight variables related to current climate: precipitation seasonality, precipitation of the wettest month (PWM), precipitation of the driest month (PDM), maximum temperature of the warmest month (MTWM), and four variables summarizing climate change since the LGM: temperature trend, temperature variation, precipitation trend, and precipitation variation (Fig. 4b,

Supplementary Fig. 5). In contrast, the GLMM for GDE only included three variables, all related to current climate: temperature trend, PWM, and MTWM (Fig. 4d, Supplementary Fig. 5). Notably, predictor variables describing human habitat modification, habitat heterogeneity, and topography did not significantly predict either GD metric (Supplementary Tables 2 and 3).

There was no residual spatial autocorrelation in the final GLMMs (Table 2, Supplementary Fig. 6), and residual error in test data prediction did not have obvious spatial biases (Supplementary Fig. 3). All parameter posterior distributions had less than 13% overlap with their prior distributions, indicating high model identifiability (Supplementary Fig. 7).

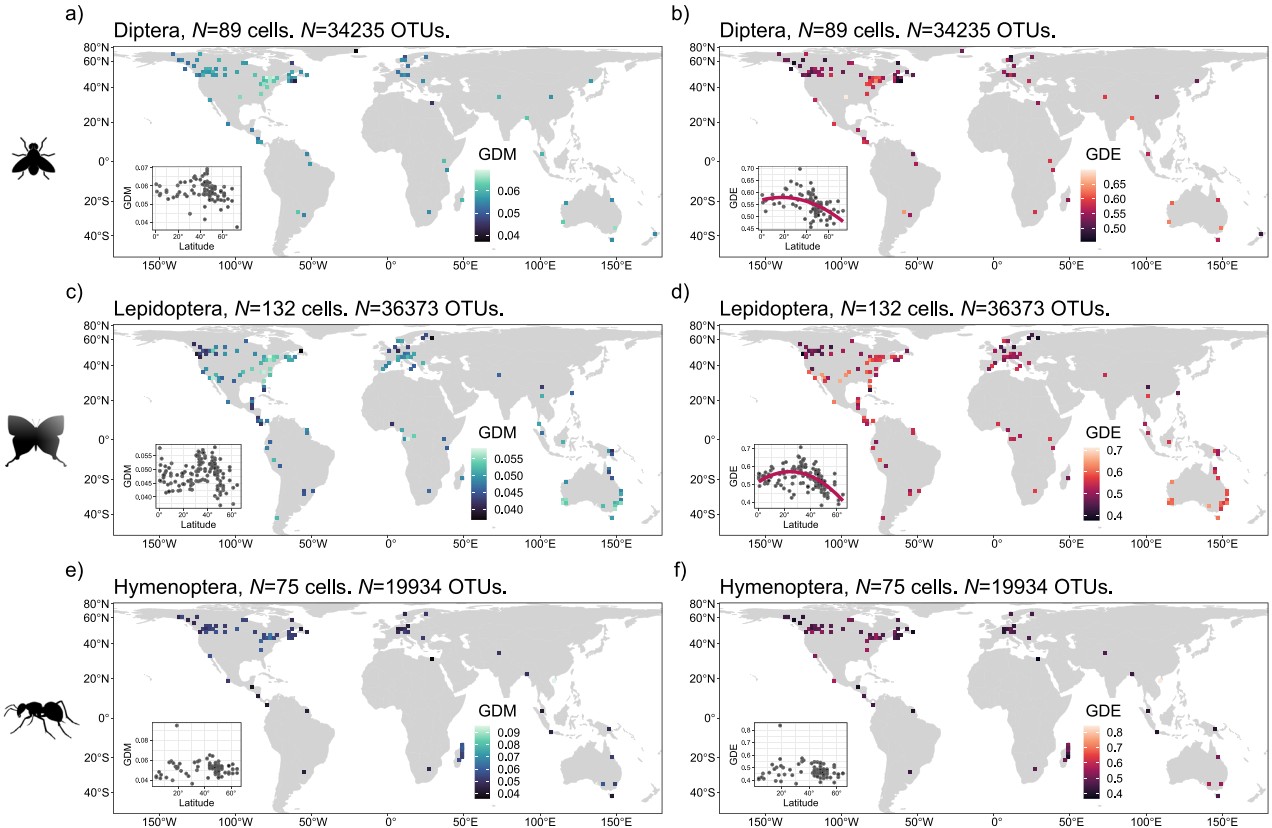

**Fig. 3 | Genetic diversity distributions for the top three most sampled taxa.** The observed distributions of genetic diversity mean (GDM) (**a**, **c**, **e**) and evenness (GDE) (**b**, **d**, **f**) for Diptera (**a**, **b**) (34.0% of OTUs), Lepidoptera (**c**, **d**) (32.4%), and Hymenoptera (**e**, **f**) (17.3%). The same filtering criteria were applied here as in the full analysis, where only cells with at least 100 OTUs and at least three sequences per OTU were considered. Latitude did not significantly vary with GDM for all orders (spatially modified t-test), but did vary with GDE for Diptera (**b**) and Lepidoptera (**d**). Source data are provided as a Source data file.

## Global predictions of insect mitochondrial genetic diversity

We then used the best-fit GLMM to predict and map the global distribution of GDM and GDE individually and jointly across the globe, including for unsampled areas (Fig. 2, Supplementary Fig. 8). To prevent model extrapolation into areas of non-analog environments and highlight areas where further sampling is warranted, we omitted predictions in all areas with environmental conditions that fell outside the model training range, including Antarctica, a large portion of northern Africa, the Arabian Peninsula, parts of central Asia, and interior Greenland (Fig. 2; shown in gray; Supplementary Fig. 9).

Areas predicted to have high levels of both GDM and GDE (above the 90th percentile) include eastern North America, the North American desert southwest, southern South America, southern Africa, and southwestern Australia (Supplementary Fig. 4a; Fig. 2). Areas predicted to have the lowest GDM and GDE values (below the 10th percentile for both) were found in northern North America and Europe (Supplementary Fig. 4c). When considered independently, GDM is predicted to be highest (above the 90th percentile) in eastern and southwestern North America, southeastern Asia, southern Australia, northern Madagascar, and southern Argentina (Supplementary Fig. 4e; Fig. 2), and is predicted to be lowest (below the 10th percentile) for the Nearctic and Palearctic tundra, Europe, Australasia, Central America, the northwest coast of South America, and northern sections of the Amazon (Fig. 1g). When GDE is considered independently, it is predicted to be the highest (above the 90th percentile) throughout subtropical Australia, the southeastern U.S., the deserts of southwestern U.S. and northern Mexico, the transition between Saharan and sub-Saharan Africa, and South Asia (Supplementary Fig. 4i). On the other hand, GDE is predicted to be lowest (below the 10th percentile) in

Europe and parts of the Nearctic and Palearctic tundra as well as northern Madagascar and a region in central China overlapping the Tibetan plateau (Supplementary Fig. 4k; Fig. 2). Maps of the upper and lower 95% highest density interval (HDI) predictions are available in Supplementary Fig. 8.

## Taxon-specific patterns of GD

Six insect orders represent 97.2% of all OTUs in this study (Supplementary Fig. 10; Supplementary Table 4). In order of prevalence, they include Diptera, Lepidoptera, Hymenoptera, Coleoptera (the four mega-diverse orders that include *ca.* 80% of known insect species), Hemiptera, and Trichoptera. The remaining 2.8% of OTUs belong to 20 additional insect orders. Although Coleoptera has more described species than the other three megadiverse orders, it comprises less than 10% of OTUs represented in the data, possibly due to the common practice of using Malaise traps in flying insect surveys, where Diptera, Hymenoptera, and Lepidoptera dominate sampling[74]. Across orders, 74.7% of all OTUs occupied a single grid cell with less than one percent occupying more than 11 grid cells (Supplementary Fig. 11).

To investigate the influence of the three most prevalent orders (Diptera, Lepidoptera, and Hymenoptera, 84.2% of total) we removed these orders from the full dataset and reanalyzed patterns of GDE and GDM. Using Welch's unequal variance t-tests, we found no significant difference in GDE estimates between the full dataset and the dataset with the most prevalent outliers removed (Supplementary Fig. 12; $p_{100} = 0.065$). However, GDM was slightly but significantly lower in the full dataset (mean$_{diff}$ = −0.004, df = 122.44, $P < 0.001$).

OTU sampling across the most abundant three orders varied geographically (Supplementary Fig. 13). When we calculated OTU

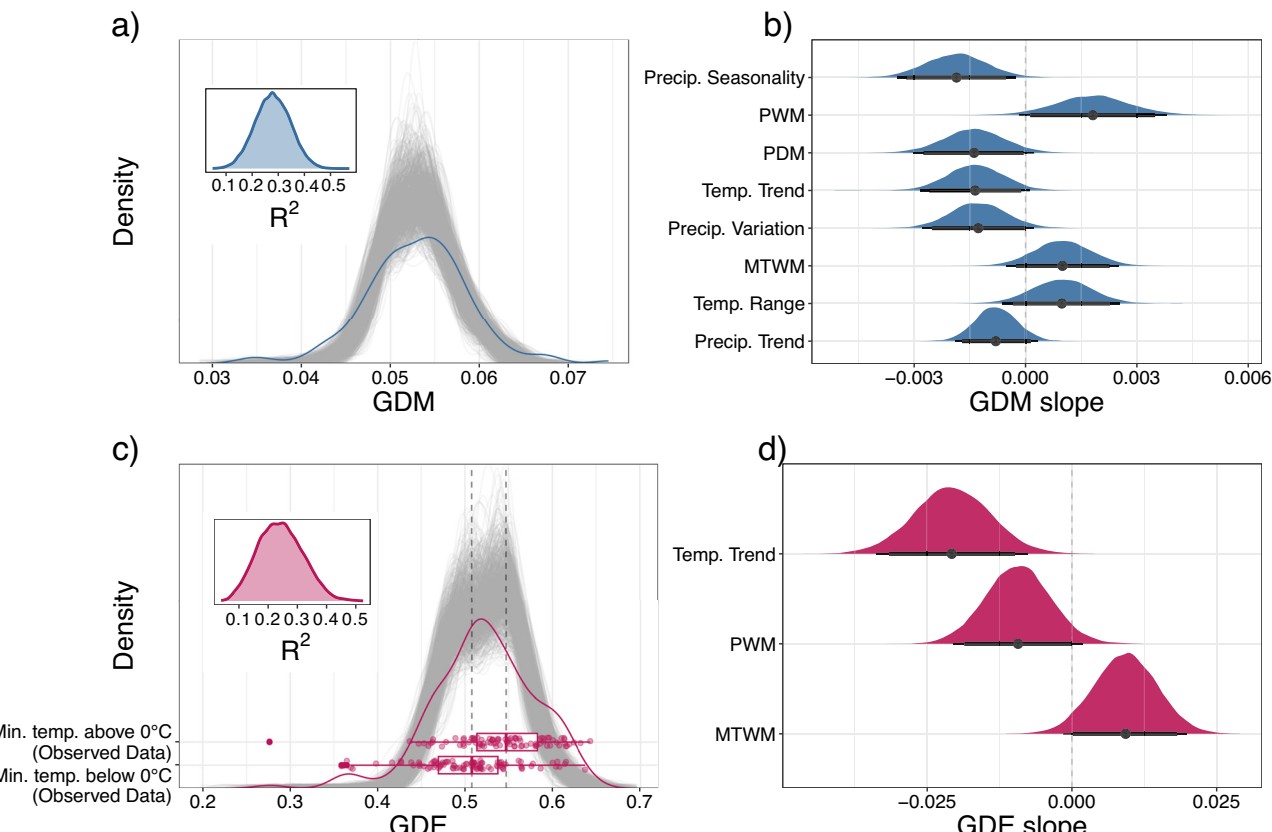

**Fig. 4 | Distributions of observed and predicted genetic diversity mean (GDM) and evenness (GDE).** The gray lines in (**a**) and (**c**) are 1000 random samples from the posterior distribution of the GDM and GDE models. The blue and red lines are the observed distributions of GDM and GDE, respectively. The boxplot overlaid on (**c**) illustrates the higher observed GDE in areas that do not freeze (minimum temperature >0 °C) versus GDE in areas that do freeze (minimum temperature <=0°). The boxplot center represents the median of the data, while the lower and upper hinges correspond to the first and third quartiles. The whiskers extend to the largest value no further than 1.5 times the inter-quartile range from the hinge. The observed differences in GDE are reflected in the posterior draws, which we

highlight with two gray, dashed lines drawn through the medians of the observed data. Bayesian $R^2$ posterior distributions are shown as insets in (**a**) and (**c**). The density plots in (**b**) and (**d**) summarize 1000 random samples from the posterior distributions of the slopes for each predictor variable. The thin bars under each density plot indicate the 95% highest posterior density interval (HDI) and the thick bars indicate the 90% HDI. The dots indicate the median of the posterior distribution. PWM precipitation of the wettest month, PDM precipitation of the driest month, MTWM maximum temperature of the warmest month, Temp. Trend historical temperature trend since the last glacial maximum, Precip. Trend historical precipitation trend since the LGM. Source data are provided as a Source data file.

**Table 2 | Results of the spatial linear modeling of environmental correlates for genetic diversity mean (GDM) and genetic diversity evenness (GDE)**

| Response | Predictor variables | Median $R^2$ | Lower 95% HDI | Upper 95% HDI | Moran's $I$ | $p$-value (Moran's $I$) |
|---|---|---|---|---|---|---|
| GDM | Precip. seasonality, PWM, PDM, temp. trend, precip. variation, MTWM, temp. range, precip. trend | 0.279 | 0.146 | 0.407 | −0.038 | 0.738 |
| GDE | Temp trend, PWM, MTWM | 0.240 | 0.101 | 0.390 | −0.043 | 0.708 |

Columns 3–5 contain a summary of the Bayesian $R^2$ model fit statistic for the Bayesian generalized linear mixed models. Residual spatial autocorrelation for each model was calculated using Moran's $I$ and a one-sided permutation test with 10,000 simulations was used to calculate the $p$-value.
HDI highest posterior density interval, PWM precipitation of the wettest month, PDM precipitation of the driest month, MTWM maximum temperature of the warmest month, Temp. Trend historical temperature trend since the last glacial maximum (LGM), Precip. Trend historical precipitation trend since the LGM.

sampling as the number of OTUs per order within each cell, Diptera dominated OTU sampling towards the far northern latitudes, while Lepidoptera dominated sampling south of these far northern latitudes, from North America and Europe towards the equator, and Hymenoptera typically accounted for fewer than 50% of OTUs sampled, with overrepresented sampling in Madagascar (Supplementary Fig. 13).

## Discussion

We found a quadratic latitudinal gradient in genetic diversity evenness (GDE) that peaks at subtropical latitudes and decreases near the equator and towards the poles (Fig. 2). However, genetic diversity mean (GDM) did not significantly vary with latitude, even though the

two metrics are significantly correlated. Although our approach is correlative and does not link observations with causality as one could with a process-explicit model[75], these results suggest that forces underlying intraspecific mitochondrial genetic diversity could be inherently different from those driving the classical negative latitudinal gradients in species richness and phylogenetic diversity found in most arthropod taxa, including ants, butterflies, and spiders[76–80], as well as plants[81], which are expected to be strongly linked to insect biogeographic patterns. Bees (within order Hymenoptera) may be a notable exception, as current estimates of species richness show a latitudinal gradient similar to GDE with highest richness at mid-latitudes[82]. However, when we fitted quadratic correlations to the three

most common orders separately, Hymenoptera was the only taxon with the two genetic diversity (GD) metrics that did not match the general trend (Fig. 3).

The negative quadratic latitudinal correlation of GDE in two of the three most sampled orders (Diptera and Lepidoptera) and the lack of an overall correlation with GDM suggests a departure from expectations of species genetic diversity correlation predictions, given the expected negative linear correlation of species richness and latitude[32,83–85]. However, many confounding factors could affect how species diversity metrics relate to GDM and GDE, and these factors may have both positive and negative effects, leading to large variation in the direction and strength of species genetic diversity correlations[86], especially at a global scale in such a large taxonomic group such as insects. Sampling biases, especially for the overrepresented North American and European regions may also influence this relationship by overfitting models towards patterns present in these regions, although we find no evidence of a correlation between sampling effort and the two GD metrics in our data (Supplementary Fig. 14, Supplementary Table 5).

The lack of a latitudinal gradient in GDM and the presence of one in GDE peaking in the subtropics also contrasts with recent macrogenetic studies of vertebrates, all of which find a negative latitudinal gradient of GDM and related metrics peaking in the tropics and declining poleward, including for mammals[20], amphibians[23], and fishes[22]. Why would GDE be lower in areas like the wet tropics where the species diversities of most insect groups reach their peaks[87,88]? Rapoport's Rule, the tendency for species' range sizes to increase with increasing latitude[89,90], might partially explain this result if species with larger ranges tend to harbor greater genetic diversity[91,92]. This expectation has recently been observed empirically[28], as well as investigated in a theoretical macrogenetic context whereby per-species GD relates to species' ranges following a power law[20]. If processes underlying Rapoport's Rule operate in the two large insect orders Diptera and Lepidoptera, coalescent times among sampled individuals for larger-ranged species at higher latitudes are expected to be older and therefore yield consistently higher values of per-OTU GD that aggregate to higher GDE within grid cells[93]. This expectation should hold even if species' ranges are larger than our chosen $193 \times 193$ km grid-cells and if subdivision exists within the range of a species[94,95]. Furthermore, the inflation of GD in a larger portion of OTUs within a grid-cell serves to increase evenness (GDE) by bridging the difference in GD between high GD and low GD OTUs within the sampled assemblage. This should also lead to higher GDM, but the presence of OTUs with extremely high or extremely low GD are more likely to lead to more statistical noise for the calculation of GDM than GDE. Although we do not find a significant correlation between latitude and GDM, we acknowledge that the positive correlation between GDE and GDM may contain some signal of an increase in GDM with latitude (Fig. 2d, e).

Multiple mechanisms may positively influence per-species GD, but they do not necessarily coincide with each other. For example, although larger ranges and climatic stability both have a positive effect on per-species GD, they may not co-occur. For example, a climatically stable area might also coincide with small range sizes, and conversely an area that experiences cycles of glaciation may also coincide with many large-ranged species. Additionally, we find that OTUs occupy more grid cells between 40° and 60° latitude (Supplementary Fig. 14), matching our expectation for larger ranges at higher latitudes, although we acknowledge that sampling bias may contribute to this pattern as well. Given that the observed peak in GDE is in the subtropics, but not at these higher latitudes, the impact of recent founder effects associated with post-glacial expansions on GD may outweigh the influence of larger range sizes at higher latitudes[71].

Indeed, our finding of lower GDE in high-latitude areas that were glaciated or tundra during the LGM is less surprising given that founder expansion dynamics predict lower GD values for species that expanded poleward from Pleistocene refugia[71,96,97]. While a significant poleward decline is found only in GDE, with the decline in GDM being apparent at extreme latitudes but not significant, this arises from many zero GD OTUs that are co-distributed with a small number of hyper-diverse OTUs in these northern regions. Indeed, this pattern is largely consistent with a gradient of lower haplotype richness in recently deglaciated areas found in European butterflies[31] based on the same *COI* data from BOLD used here. Similarly, aquatic insect species have lower intraspecific genetic diversities in recently deglaciated areas of Europe compared to Neotropical areas[33], as does an assemblage of *Anopheles* taxa co-distributed across the Indo-Burma biodiversity hotspot[98]. However, a deflation of GDE and GDM in temperate regions could also be driven by widespread positive selective sweeps on the mtDNA genome across insect species that expanded into these previously glaciated areas after the LGM, a scenario that is non-mutually exclusive to founder-effect dynamics[99].

In contrast to the declines of GDE in previously glaciated areas, we hypothesize that the peak of GDE in the subtropics, and more generally the correspondence of GDM and GDE with seasonally high temperatures (MTWM), results from climates that have remained stable since the LGM (Fig. 4). Likewise, declines in GDE in previously glaciated areas and the positive correlations between GDM and GDE with MTWM are also consistent with the evolutionary speed hypothesis, where warm temperatures are posited to decrease generation times and speed up mutation rates, leading to higher genetic diversity[100]. Similar patterns are found in macrogenetic studies of fish[22] and animals and plants in general[101]. Moreover, GDM and GDE have contrasting relationships with seasonally high precipitation (PWM), where GDM increases with PWM, while GDE decreases. The former corresponds with wet, hot regions, while GDE is highest where it is arid and hot (Figs. 2, 4).

GDM's correlation with high MTWM and PWM, low seasonal variation in precipitation (precipitation seasonality) and high long-term climate stability (precipitation variation, precipitation trend, and temperature trend) is reflected by predicted hotspots in the most long-term stable and least seasonal tropical or subtropical forest habitat (Supplementary Fig. 4e, Supplementary Fig. 5). In addition to increased evolutionary speed, the high resource availability and stability of these areas may allow for population persistence and the consequent accumulation of genetic diversity. In contrast, GDE's peak in subtropical, hot, and arid environments could be partially driven by processes related to geographic patterns in physiological tolerances that lead to increases in the size and uniformity of species ranges. Wide-ranging extratropical insect species can usually tolerate a broader range of climatic variation, whereas limited-range tropical insect species tend to have a narrow climatic niche, stronger habitat specializations, and narrower physiological tolerances[102]. Insect diapause is thought to provide this adaptive tolerance to wider abiotic conditions and may result in larger and more uniform range sizes across an assemblage[102,103]. If there is a positive relationship between range size and GD[91,92], this could provide a possible mechanistic relationship that connects Rapoport's Rule and the more uniform genetic diversities found in the subtropics. In this case, increasing GDE with latitude may be driven by uniformly larger range sizes that result from greater physiological tolerances in harsher environments. While this is consistent with both GDM and GDE being most strongly associated with the seasonally hot conditions (MTWM) found in the subtropics (Fig. 4), this relationship drastically breaks down in the colder temperate and subarctic regions (Fig. 2) that frequently freeze, especially for GDE (Fig. 4).

The regions predicted to have higher and more uniform GD correspond with some known hotspots of insect biodiversity. For instance, the deserts of southwestern US and northern Mexico have the highest butterfly phylogenetic endemism in North America[82,104]. Southwestern Australian deserts also have exceptionally high arthropod endemism[105], and are among the original biodiversity hotspots

identified by Meyers et al.[106]. One possible explanation is that these areas are more climatically stable and hence may harbor more uniform demographic histories that manifest as elevated GDM and GDE values. However, much of the global pattern in GDE that we find is dominated by Diptera and Lepidoptera (Fig. 3). Therefore, we caution against making sweeping generalizations about such large taxonomic groups.

Areas with higher levels of GDE could also potentially emerge from different points along the continuum of community assembly[107]. Although the spatial scale is not always in line with the grid cells we employ, Overcast et al.[68] found simulated and empirical (arthropod, annelid, and trees) communities to have elevated GDE under ecologically neutral conditions in contrast to non-neutral, or "niche-structured", conditions. In these specific model-dependent cases, the lower GDE in communities assembled via environmental filtering is likely caused by increased genetic diversity in hyper-dominant species with stronger local ecological adaptation. In line with this process-explicit modeling, our observation of GDE increasing from the tropics to the subtropics shows that equatorial insect communities may have stronger, local niche-structured mechanisms (i.e., less ecologically neutral conditions) than subtropical temperate insect communities. This would be consistent with the idea of stronger niche conservatism in the tropics[108].

While these are some of the many hypotheses emerging from our study, the correlative approaches we use are a crucial first step to developing a better understanding of the processes underlying biodiversity. To more directly test such hypotheses, process-explicit models, which can uncover causal factors that drive biodiversity patterns as well as discriminate among the processes that do not, will be an important next step[109,110]. These models will be especially valuable in revealing the underlying mechanisms of unexpected correlations found here and in other macrogenetic studies[111].

While mtDNA genetic polymorphism is only one component of genome-wide genetic diversity, the latter is critical to the survival of insects and their complex interactions with other organisms[112–114]. High genetic diversity may facilitate adaptation to changing climates, emerging diseases, and pollutants: three (of many) potential drivers of the "insect apocalypse"[58]. In addition, genetic diversity contributes to the diversity and stability of species interaction networks by affecting niche space and competition[115], community structure[116], and network complexity[117]. At larger ecological scales, insect genetic diversity may reflect ecosystem function and structure as reliably as other traditional macroecological metrics such as species richness[118]. It can also augment the resilience of ecosystems that provide continuing services for humankind[119], such as disease management, curbing the spread of invasive plants, aiding sustainable agriculture, pollinating food crops, and controlling pests[120].

While the metric of global human modification we considered did not significantly correlate with GDM or GDE, there are many facets of anthropogenic disturbance acting at different spatial scales that are difficult to summarize in a single metric[121]. For well-studied systems, shifts in GDM and GDE may reflect the loss of rare species with less genetic diversity or community shifts toward wide-ranging taxa, including invasive species, and could thus be used in long-term monitoring schemes[122,123]. Although large-scale data curation efforts are underway[124], the spatiotemporal resolution of genetic sampling currently available does not permit rigorous assessment of how humanity affects insect GD at a global scale. A concerted increase in sampling effort, especially in the data-poor regions we identify, will likely make this feasible in the not-too-distant future.

It is important to note that both metrics are calculated using mtDNA, which is a single, highly functional, non-recombining locus. Using mitochondrial DNA has many important advantages[12], yet relying on a single-locus marker has drawbacks. Notably, the genetic diversity that mtDNA captures is from a single draw from the stochastic coalescent process[63], and due to the lack of recombination, patterns of mitochondrial diversity can be impacted by selection[52,61–63].

Moreover, while a large fraction of insect taxa are infected with *Wolbachia* and other endosymbiotic bacterial parasites that can be in linkage disequilibrium with mitochondria and act as potent selective agents affecting mtDNA diversity[125–127], a survey of *Wolbachia* infections in BOLD find that *Wolbachia* COI is present in only 0.16 percent of cases[128]. Furthermore, the pattern of genetic diversity at any single locus is ultimately impacted by multiple processes (phylogeographic demographic history, mutation rate, taxon-specific mutation rates and linked selection) rather than any one process alone[64,129]. For example, despite the likelihood for linked selection and high coalescent variance to distort mtDNA diversity patterns, mtDNA data routinely retain key features of species-specific demographic and phylogeographic histories[130–133]. Lastly, emergent patterns of mitochondrial genetic diversity across assemblages demonstrably recapitulate well-known biodiversity patterns like the latitudinal biodiversity gradient, indicating meaningful biological signals[21,23,26].

While intraspecific mtDNA diversity has been shown to have a weak, non-significant correlation with whole genome genetic diversity in 38 European butterfly species[134] and is generally not sufficient for making detailed inference of demographic history or phylogenetic reconstruction[62], sampling the mitochondrial genetic diversity of thousands of taxa per locale with rapidly increasing data availability is a promising first step towards understanding an important component of the biodiversity of assemblages at global spatial scales[10,12,65,66], and this will be especially valuable until whole-genome data become more readily available through projects like the Earth BioGenome Project (Lewin et al.[135]) and GEOME (Riginos et al.[136]). While taking these important cautions into consideration, we view macrogenetics as a developing field and thus basic expectations regarding observed patterns are still not established. Our study is a step in the direction of establishing this foundational knowledge and suggests avenues for ways to test specific hypotheses.

By modeling relationships between environmental data and two measures of intraspecific genetic diversity, GDE and GDM, we make assemblage-level mitochondrial genetic diversity predictions for data-poor regions of the planet, while flagging and masking those with high uncertainty[137] (Fig. 2). These genetic diversity maps have the potential to fill knowledge gaps that far exceed the undersampling and taxonomic uncertainties underlying vertebrate and plant macroecological studies[138,139]. They can also highlight mitochondrial genetic diversity as an important biodiversity component that has yet been assessed for relatively few taxa[17], while focusing attention on a data-deficient group with evidence of global population declines and strong connections to ecosystem functions and services[140]. Taken together, GDM and GDE are promising biodiversity metrics for documenting and understanding "the little things that run the world"[141].

## Methods

### Aligning and filtering sequence data

The barcoding region of the mitochondrial locus *cytochrome c oxidase subunit I* (*COI*, sometimes abbreviated *cox1*) was selected as a genetic marker that can be deployed to study genetic diversity at the macrogenetic scale. We downloaded *COI* mitochondrial sequence data for insects directly from the BOLD website using the application programming interface (http://www.boldsystems.org/index.php/resources/api; downloaded 19 Nov 2019). Our initial database comprised 3,301,025 complete insect records before applying a series of quality filters. We used the BOLD database's OTU assignments (termed barcode identification numbers; BINs), which cluster similar sequences algorithmically and map them against the BOLD database[142]. After trimming end gaps from sequences, we removed exceptionally long sequences (>800 base pairs, bp) which contained a large proportion of gaps that negatively impacted alignments and the calculation of summary statistics. In addition, we removed shorter sequences (<400 bp) that the BOLD database uses for BIN

identification and may downwardly bias GD estimates. We only retained *COI* sequences from georeferenced specimens. Sequence alignments were independently performed for each OTU within single sampled geographic raster cells, i.e., grid cells. We used default settings in Clustal Omega (v1.2.3) to align the sequences and visually assessed both a random subset of alignments and alignments with genetic diversity values at the tails of the distribution to check for alignment errors[143].

To reduce the potential impact of invasive species on our analyses, we removed trans-continental invasive species from the dataset using a list of invasive insect species compiled from seven resources: Global Insect Species Database [http://www.issg.org/database; accessed 23 May 2020]; Invasive Species Compendium [https://www.cabi.org/isc/; accessed 24 May 2020]; Center for Invasive Species and Ecosystem Health [https://www.invasive.org/; accessed 24 May 2020]; Invasive Alien species in South-Southeast Asia[144]; Japan Ministry of the Environment [https://www.env.go.jp/en/nature/as.html; accessed 24 May 2020]; European Alien Species Information Network [https://easin.jrc.ec.europa.eu/easin/Home; accessed 24 May 2020]. We identified all species and OTUs present on multiple continents and removed those on the invasive species list from our dataset. While some invasive species may be restricted to single continents, removal of such taxa was not possible given the lack of information on changes in insect range boundaries and species assignments.

## Calculating the mean and evenness of genetic diversity (GDM and GDE)

Previous global macrogenetic studies focused on spatially defined metrics that summarize genetic diversities calculated across all species sampled from an area of arbitrary spatial resolution[19,145]. This is most commonly the average genetic diversity or, alternatively, a measure of the allelic richness derived from the total number of unique and/or common alleles of a genetic locus across all taxa within an area[146]. We used two distinct summaries of the former measure of genetic diversity (GD)—the mean (GDM) and evenness (GDE) of genetic diversity per unit of area. To obtain the GD for each OTU per grid cell, we calculated the average number of nucleotide differences across all pairwise sequence comparisons per OTU per base pair[93,147] (a.k.a. nucleotide diversity). Aggregated across OTUs within each grid cell, GDM is then defined as the average GD among OTUs in each grid cell, following[21]. Because the distribution of GDM at the grid cell scale was highly skewed towards zero, we performed a square-root transformation to achieve a more normal distribution, consistent with Theodoridis et al.'s[21] approach. All subsequent statistical analyses of GDM at the grid cell scale were based on the transformed GDM.

While GDM is a standard metric in the macrogenetic toolbox, GDE is derived from a set of metrics often used in ecological studies of biodiversity. Hill numbers permit direct comparisons of diversity across scales and data types[148–150]. GDE is then defined as the first-order Hill number of GD across OTUs per grid cell, corrected by sampled OTU richness[68]:

$$\frac{\exp\left(\sum_{i=0}^{N} -\pi_i \ln(\pi_i)\right)}{N} \quad (1)$$

Where $N$ is the number of OTUs in the assemblage and $\pi_i$ is the GD for a single OTU. Correcting for sampled OTU richness allows for comparison across assemblages of different numbers of OTUs. The numerator of this metric is the exponential of Shannon's diversity index, which is also referred to as Shannon's information measure or Shannon's entropy in the literature[151]. It is commonly used to describe evenness and variability of species abundances[152,153], and here we follow the approach of Overcast et al.[68] by adapting it to do the same for genetic diversities calculated from all species sampled from a particular area.

High values of GDE indicate areas where most OTUs have a similar GD (Fig. 1), whereas lower GDE arises when GD values across the community diverge considerably[152]. The distribution of GD values within an area of low GDE can take a variety of shapes, but the most common in our observed data is markedly L-shaped whereby most OTUs have low or zero GD along with a small number of OTUs with large GD (Fig. 1d).

## Spatial resolution and sampling decisions

To assess how the spatial scale and density of OTU sampling impacted our results and to establish a sampling strategy that maximizes the amount of information, we calculated both metrics at 1) three different spatial resolutions, and 2) six different thresholds of minimum OTU sample sizes per grid cell. The spatial resolutions include 96.5 km × 96.5 km, 193 km × 193 km, and 385.9 km × 385.9 km equal-area grid cells using a Behrmann cylindrical equal-area projection, which are 1°, 2°, and 4° longitude at 30°N. We considered a minimum of 10, 25, 50, 100, 150, or 200 unique OTUs per grid cell. We chose the spatial resolution that balanced the average number of OTUs per grid cell, the number of grid cells, the average number of taxonomic orders per grid cell, and variation in the number of OTUs across grid cells (Supplementary Fig. 2). After choosing an appropriate spatial resolution for our analysis, we performed the modeling procedure outlined below for all minimum OTU thresholds. While retaining results across the range of minimum OTUs per grid cell, we focus our analysis using the threshold that results in the least-biased and most precise estimates of GD when predicting a trained model onto withheld test data, where a slope near one with a y-intercept near zero indicates minimal prediction bias and high precision is indicated by a high $R^2$ with low root mean squared error. With respect to numbers of sampled allele copies per OTU, we used a minimum of three individuals per OTU per grid cell. This is a sensible approach to estimate GD while still maximizing data use because BOLD data submissions may omit duplicate alleles and coalescent theory suggests that using average pairwise distance from 5–10 samples per OTU provides estimates of genetic diversity that are as reliable as those obtained from hundreds of samples[93]. To explore this sampling dynamic explicitly, we conducted coalescent simulation experiments comparing how the calculation of GD varies given identical samples with and without duplicate alleles removed. These simulations showed that retaining only unique haplotypes resulted in a small, consistently upward bias in GD values. Additionally, increasing values of effective population size (*Ne*) decreased this bias, with estimates of GD with and without duplicate alleles converging for *Ne* values greater than ~10e5 (Supplementary Methods, Supplementary Fig. 15).

Because 97.2% of OTUs are represented by six taxonomic orders (Supplementary Fig. 10; Supplementary Table 4), with 84.2% represented by three (Diptera, Lepidoptera, and Hymenoptera), we investigated whether and to what degree over-represented orders might be driving the signal of GDM and GDE. We compared the global frequency distributions of per-cell GDM and GDE with these three orders removed with the distribution of these summary statistics for the entire data set. The distributions of per-cell GDM and GDE between these filtered data sets and the original data set were compared using Welch's unequal variance t-tests[154]. In addition, we mapped the observed distribution of GDM and GDE for the three orders to compare their geographical variation in GD with the full data sets.

Although coalescent theory predicts that the number of allele copies per OTU per grid cell will have a limited impact on the per OTU genetic diversity[93], we examined whether this assumption was met in the data by testing for Pearson's correlations between the per OTU GD and the number of individuals per OTU. The relationship was statistically significant, but extremely weak ($r_{100} = 0.029$, $p_{100} < 0.001$). Similarly, to investigate whether per grid cell sampling, i.e., total number of individuals, number of individuals per OTU, and number of OTUs per cell, had an effect on GDE or GDM, we tested for Pearson's correlations

between these quantities (no relationship, all $P > 0.20$, Supplementary Table 5; Supplementary Fig. 16). We also assessed sampling variation by taking the ten most sampled grid cells (2748 to 13,300 OTUs per grid cell) and obtaining sampling distributions of GDM and GDE for each by resampling with replacement 100 OTUs per sample ($N = 1000$ resamples) and calculating the summary statistics for each resample (Supplementary Fig. 17). Finally, we considered the spatial distribution of OTUs by visualizing the distribution of the number of grid cells occupied by each OTU.

## Environmental variable selection

We aggregated a total of 49 abiotic, biotic, and anthropogenic variables that potentially influence intraspecific genetic diversity in insect communities (Supplementary Table 1). We removed highly correlated variables ($r > 0.75$), prioritizing variables that represent climate extremes, climate variability, habitat variability, last glacial maximum (LGM) climate stability, and human influence on the environment. We prioritized these classes of variables because the influences of environmental extremes and variability across space and time are more likely to shape local communities than averages[155,156].

We retained a final data set of 11 ecologically relevant variables: five bioclimatic variables, habitat heterogeneity, global human modification, and four metrics of climate stability (temperature and precipitation) since the LGM (Supplementary Table 1). The five bioclimatic variables describe climate extremes and variability, and were obtained from the CHELSA database[157]. They include maximum temperature of the warmest month (MTWM), minimum temperature of the coldest month (MTCM), precipitation of the wettest month (PWM), precipitation of the driest month (PDM), temperature seasonality, and precipitation seasonality[157,158]. The habitat heterogeneity metric was calculated as the standard deviation of the Enhanced Vegetation Index, which was derived from the Moderate Resolution Imaging Spectroradiometer (MODIS) (2.5 arc-min; ref. 159). The human modification variable is a cumulative measure of human modification to terrestrial areas[160]. Measures of both the historical trend and variability of temperature and precipitation over the last 21,000 years were obtained from Theodoridis et al.[21]. The specific definitions of these derived metrics include "deep-time climate trend", the change in climate within each century, averaged across centuries, and "deep-time climate variability", meaning the standard deviation around the change in climate, averaged across centuries. Low deep time trend values indicate regions with long-term climate stability, while low variability values indicate regions with short-term climate stability. Each variable was aggregated from its original resolution (see Supplementary Table 1) to 193 km by 193 km resolution through bilinear interpolation.

In addition, we explored the relationship between GDE and GDM and a binary variable delineating the globe into areas that do or do not freeze, which are areas where the long-term minimum temperature of the coldest month (MTCM) is below 0 °C versus above 0 °C, respectively. These regions correspond with sharp community turnover in birds[72] and could correlate with critical life processes for insects. We tested this variable separately from the full predictive model because it is highly collinear with the deep-time temperature variability metric ($r > 0.75$) included in the model. Given this correlation, we acknowledge the redundancy in the statistical tests, but we believe that including two perspectives on the geographical structure of genetic diversity allows for direct comparison with existing patterns in the literature that are not apparent otherwise.

## Modeling approach

We applied a Bayesian modeling approach to identify the environmental conditions that best explain the global distribution of GDM and GDE in insects. We independently modeled the relationship between the set of 11 uncorrelated, ecologically relevant variables (see above) and per grid-cell GDM and GDE values. To assess the predictive ability of

the model, we split the data set into 75% training and 25% testing sets, stratifying the sampling by continent to maximize spatial representation of sampling in both data sets. All model selection and model fitting was performed on the training set, while predictive performance of the best fit model was assessed by predicting the withheld test data set.

We prioritized constructing a simple, interpretable linear model that predicts GDM and GDE across the globe by first reducing the number of potential variable combinations, followed by a Bayesian hierarchical generalized linear mixed model (GLMM) approach that accounts for spatial autocorrelation[73]. We reduced the number of potential predictor variable combinations from the set of 11 variables with low collinearity using Bayesian regression coupled with projective prediction feature selection. This approach minimizes the number of variables in a simple model while retaining comparable predictive power to a model that includes the full suite of predictor variables[161,162]. Computational and statistical limitations due to exploring such a wide variety of variable combinations in a Bayesian context prohibited accounting for spatial autocorrelation in this first step. For each model we used regularizing priors on all slope parameters (N(0, 0.1)) and the error term (N(0, 1)). We centered and scaled all variables to a standard deviation of 1 and mean of 0 prior to modeling.

If residual spatial autocorrelation (SAC) is present, the assumption of independent and identically distributed residuals would be violated, resulting in potentially biased overprecision of parameter estimates[163]. We tested for SAC in the residuals of the resulting simplified models using Moran's $I$ and 10,000 simulations implemented in the R package *spdep* v1.1-2[164]. We detected significant levels of SAC in the residuals of our GDE model (Moran's $I = 0.149$, $P = 0.008$) and our GDM model (Moran's $I = 0.306$, $P < 0.001$).

Given this presence of SAC, we used a Bayesian hierarchical GLMM implemented in the R package *glmmfields* v 0.1.4 to model the relationship between the variables and the two GD metrics (GDE and GDM) while accounting for SAC[73]. SAC is modeled as a random effect with a multivariate t-distribution determining the shape of the covariance matrix. Model parameters were estimated from the posterior distribution using a No U-Turn Sampler[165,166]. Further model specifications are provided in the Supplementary Methods. We again tested for SAC in the residuals of these models using the same approach as above. The proportion of variance explained by the models were assessed with Bayesian $R^2$[167], modified to account for spatial autocorrelation error. After selecting a model, we used the percentage of prior-posterior overlap to assess the identifiability of parameter estimates relative to the information provided by their prior distributions[168]. Low overlap between the prior and posterior distribution of a parameter indicates that there is sufficient information in the data to overcome the influence of the prior.

We tested for the statistical significance of a linear or quadratic relationship between latitude and GDM and GDE while accounting for spatial autocorrelation using a modified two-sided t-test of spatial association, implemented in the R package *SpatialPack* v0.3.0[169,170]. This was done for the full data set and independently for the three most sampled orders. We also independently tested the effect of whether an area freezes or not on the two GD metrics using the same modified t-test of spatial association.

## Global genetic diversity map generation

Using the final models of GDM and GDE, we created maps of the global distribution of insect GD. We used 1000 draws from the posterior distribution to predict terrestrial environments across the globe. We included all continents except Antarctica, which had no observed data and included environments far more extreme than the observed data. We created maps of the median predicted GDM and GDE, along with the upper and lower 95% HDI. In addition, we created bivariate color maps of these prediction intervals for combined GDM/GDE to highlight areas where GDM and GDE vary in similar and different directions.

To avoid making poor predictions into areas with environments that are non-analogous to the areas used to train the models, we used multivariate environmental similarity surface (MESS) maps (Supplementary Fig. 9). MESS maps visualize how environmentally similar or different areas across the globe are compared to the model training data[171]. Following best practices to avoid extrapolation[172], we used the MESS results to mask areas with non-analogous environmental space (values less than 0) on our global prediction maps, indicating areas with high prediction uncertainty.

## Reporting summary

Further information on research design is available in the Nature Portfolio Reporting Summary linked to this article.

## Data availability

The geographic and genetic sequence data, in addition to raw model output generated in this study, have been deposited in the figshare database at https://doi.org/10.6084/m9.figshare.c.6563836.v1[173]. All environmental data are publicly available and links are provided in Supplementary Table 1. Source data are provided with this paper.

## Code availability

All code used for data processing and analysis is available at https://doi.org/10.5281/zenodo.8125548.

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

## Acknowledgements

We thank Dr. Jason L Brown for his valuable feedback on early drafts of the manuscript, and the Anderson, Hickerson, and Carnaval labs at City College of New York for their feedback at every step of the project. Additionally, we thank Dr. Matthew Hahn and Dr. Tara Pelletier for their helpful feedback on the interpretation of mtDNA diversity patterns. C.M.F., M.J.H., A.C.C., I.O., and A.R. acknowledge support from NSF DBI 2104147 and the NSF RCN Cross-Scale Processes Impacting Biodiversity (DEB 1745562). D.J.L. was funded by NSF DEB-1541557. J.M.K. was supported by the Japan Society for the Promotion of Science Postdoctoral Fellowships for Foreign Researchers Program and JSPS Grants-In-Aid for Scientific Research (KAKENHI) (20F20774).

## Author contributions

C.M.F., L.D.B., and M.J.H. conceived of the study. C.M.F., L.D.B., J.M.K., K.A.M., I.O., A.R., P.P.A.S., A.C.C., and M.J.H. framed the study. C.M.F., L.D.B., and M.J.H. carried out the analyses. C.M.F. and M.J.H. led the writing. C.M.F., L.D.B., A.C.C., E.P.E., J.M.K., D.J.L., K.A.M., R.M., I.O., A.R., P.P.A.S., and M.J.H. contributed to interpretation of the results and to the writing, and all have approved the submission. A.R. and P.P.A.S. contributed to Fig. 1.

## Competing interests

The authors declare no competing interests.
