## [Peer Review File · Nature Communications]

Global determinants of insect mitochondrial genetic diversityREVIEWER COMMENTS

Reviewer #1 (Remarks to the Author):

The manuscript presents a very welcome analysis of global distribution of intraspecific genetic diversity in insects, for which the authors selected and aligned a large number of mtDNA sequences from the BOLD database. It is generally very well written.

My main perplexity concerns the analytical approach, which I found rather confusing. In particular, the authors recognize the importance of accounting for spatial autocorrelation (SAC) in their analyses. However, as far as I understood, they did not include terms for SAC in all of their analyses and they performed multiple steps of model selection, with and without accounting for SAC. I did not actually understand the logic of this approach and I don't see how accounting for SAC throughout the whole analysis could not be a better solution.

I also don't really understand how the "final" model selection (test of the current/past climate hypotheses) step was applied. First of all, it is not clear whether a "null" model (e.g. with an intercept and a SAC term only) was also tested. It was done elsewhere in this study and I think it should be done here too. Second, what is the actual meaning of first selecting a subset of predictors based on the importance of their effect on the response and, then, dividing the surviving predictors into groups based on a completely new criterion (current vs. past, in this case) and perform another round? Personally, if past climate variables were retained in the final models, I'd assume they have a significant effect on the response, and I'd like to know what their effects on the response are.

Another general issue is whether the geographic density of the cells that were considered in the analysis is sufficient for the global-scale extrapolation that was eventually performed.

As I wrote, although the details of the procedure remain rather unclear to me, it seems like spatial autocorrelation has been accounted for, which is reassuring. However, I would appreciate a more careful effort to explain how reliable the extrapolation may be. As the authors highlight the importance of being able to cast predictions over huge unsampled areas, some sort of cross-validation of predictions would be necessary. In this respect, I was perplexed by the choice of filtering the data to only include cells with at least 100 OTUs. On one side, I see that the choice is somehow conservative, and may be the best thing to do, but is really leaving only a handful of cells outside North America and Europe. As a reader, I would really like to see how things change if the threshold is lowered and, at any rate, a deep discussion concerning the dramatic lack of sampling throughout most of the world.

Very importantly, most of the discussion is devoted to explaining what a "bimodal" latitudinal trend in genetic diversity may mean. However, as far as I understand, the authors did not find evidence for a latitudinal trend in GDM. Indeed, after (somewhat inelegantly, in fact) removing part of the data, they detected a signal for GDE, but not GDM, but they discussed their results against predictions and previous findings which, all, concerned GDM, not GDE. Conversely, the interpretation of the (statistically significant) relationships of GDM/GDE with climate variables are barely discussed! I argue that this important portion of the discussion/study deserves deep rethinking.

A relatively minor point is that, while the addition of Genetic Diversity Evenness (GDE) to the more usual "mean" GDM is introduced as a very important step forward compared to previous similar studies, there is no mention in the Introduction about why exactly this is the case in terms of which patterns of GDE should be expected under different hypotheses. In fact, there is some (interesting) discussion about expectations concerning GDE, based on the results from Overcast et al. 2020 (now published in MER). However, 1) the importance of measuring GDE in terms of which hypotheses may be tested by using it should be clarified in the Introduction, so that the reader may immediately understand it; 2) by reading the paper by Overcast et al., I did not have the impression that the GDE computed here is actually the same as the qD in Overcast et al (as the latter contains a term for species abundance): I think this should be carefully explained.

Other aspects of the discussion were not very well explained or (to my advise) lacked a strict logical consistency. I have noted this in a commented version of the manuscript's .pdf, which I

attach and which also contains more sparse comments.

All in all, my suggestion for the authors is to 1) streamline and adjust the statistical analysis (especially wrt the treatment of SAC), providing thorough justification for each step and 2) rework the manuscript so that the discussion starts from well-defined expectations to be explained in the introduction, concerning both the GDM and GDE metrics.

Reviewer #2 (Remarks to the Author):

This is an ambitious series of analyses that sets out to map and predict the global distribution of the mean genetic diversity and the evenness of genetic diversity in insects globally. The authors analyze >2.3 million sequences mitochondrial CO1 gene from >95K species sampled across 187 grid cells (193km x 193km) grid cells. This is an impressive number of sequences and species and there have been no macrogenetic studies of insects to date. Questions about the broad spatial patterns in intraspecific genetic diversity are, as the authors note, interesting, timely, and important for conservation.

While I am supportive of and interested in the questions addressed in this paper and think analyses of these data can certainly be made publishable, I cannot support the publication of the paper in its current format. My primary concern is that the data analyzed are inappropriate for addressing the questions posed. mtDNA are not neutral, do not recombine, are not well correlated with demography, are not correlated with neutral genome-wide diversity or adaptive potential (e.g., Allio et al 2017 Mol Biol Evol; Edwards and Bensch 2009 Mol Ecol; Galtier et al. 2009 Biol Lett; Galtier et al 2009 Mol Ecol). For example, in *Drosophila*, the mt genome plays an outsized role in clinal adaptation and is suspected to be heavily involved with climatic adaptation (see Quantifying the relative contributions of the X chromosome, autosomes, and mitochondrial genome to local adaptation Lasne et al. 2019 Evolution). Such data are unsuited for testing demographic hypotheses should be analyzed and interpreted as if they represent adaptive genetic diversity (see Yiming et al. 2021 Ecography). The firm conclusion of the Galtier et al 2009 ME review and references therein are that mtDNA cannot be used as a general population genetic measure of the diversity of anything but mitochondrial DNA.

Text from the manuscript: "In contrast, species richness and GD may be decoupled due to the "cradle" hypothesis that predicts higher speciation rates, more population structure, and smaller range sizes leading to Rapoport's Rule, the tendency for species' range sizes to increase with increasing latitude. Second, we might predict that the influence of Late Quaternary climate fluctuations to have an impact on GD through population demographic processes influenced by cyclical variation in precipitation, temperature, and glaciation patterns where areas with more stable climatic histories are predicted to have increased GD. Finally, we consider the influence of human disturbance on patterns of assemblage-wide GD, which we expect to decrease in magnitude in areas of high human influence."

Each of these hypotheses acts through demographic processes acting on populations. mtDNA is not suited to test them.

The Galtier review is particularly compelling and comprehensive and those criticisms of using mtDNA as a measure of general genetic diversity have, to my knowledge, only grown stronger. If the authors disagree with this substantial literature and feel pragmatism is warranted those criticisms need to be directly confronted in text if the reader is to be able to evaluate the paper on its own. The strong conclusion of the paper can be summed up in this quote:

"For all these reasons, mtDNA is perhaps intrinsically the worst population genetic and phylogenetic molecular marker we can think of."

The authors of this manuscript correctly point out that their approach follows the methods of previous studies, some of which were published in this journal (e.g., Theodoridis et al. 2020). These papers too, to varying degrees, did not address the substantial shortcomings of using

mtDNA to estimate genetic diversity. Galtier also predicted the continued use of mtDNA as a population genetic marker:

"Despite the many concerns we express about the use of the mitochondrial marker in molecular ecology, we are confident that scientists will keep on analysing mtDNA variation in the wild. The reason why we think mtDNA will keep its popularity is its practical convenience".

Despite my harsh criticisms of mtDNA work, I do appreciate there is value in exploring global patterns of mtDNA diversity. I suspect that there are many interesting processes related to the biology of mtDNA underlying some of the patterns detected. However, the sentiment the impressive volumes of data for mtDNA allow these markers to overcome their shortcomings is, to the best of our knowledge, not well supported by data. If mtDNA cannot measure the quantities of interest increasing the volume of data will not help (Paz-Vinas et al 2021 *Ecol Let*; Galtier et al 2009 *ME*).

Putting aside the appropriateness of this measure of mtDNA as a population genetic marker there are also significant methodological issues with this work. Again the authors have followed previous studies but these studies share the same flaws. Schmidt and Garraway 2021 (*Con Gen*) note that mean genetic diversity of arbitrary numbers of species aggregated from within arbitrary sized grid cells is not a meaningful population genetic or conservation relevant quantity. This and the previous studies the authors followed take the mean genetic diversity of across all species sampled per cell regardless of the fact that the species to be averaged will have extremely different life and evolutionary histories, which will produce different equilibrium genetic diversities, and the fact that mtDNA mutation rates can vary extensively. Additionally this method of data aggregation means that different numbers of species and indeed different species are being compared. Millette et al grouped mtDNA sequences by species within grid cell and I suggest the authors explore this method. It too is not ideal but here nucleotide diversity is interpretable. The authors are careful to call their metric mean genetic diversity until the discussion where they slip into making inferences about intraspecific variation. Again the previous studies call mean genetic diversity intraspecific genetic diversity too. I cannot see how the mean of multiple species gives information about intraspecific variation. The Millette approach is a measure of intraspecific mtDNA genetic diversity—this does not alleviate concerns about the inappropriateness of mtDNA as a population genetic marker.

Paz-Vinas et al note that arbitrarily aggregating sequences across large spatial areas create implausible populations. This is true in the extreme. Populations are not defined genetically and contain multiple species. Most inferences in this paper refer to population level processes (evolutionary change, relationships to environments, drift etc). Paz-Vinas also showed that sequence deposition is often incomplete and idiosyncratic authors tending to deposit only new sequences in a region. This too will significantly affect their measures of genetic diversity.

The predictive maps look good. However, I am very leery about interpolating the globe from just 187 grid cells. The posterior checks look good but that just means the model can predict the data used to generate it, which is important! But predicting different continents with different species groups and geological histories with such a small sample seems fraught.

Finally, I am concerned with the conservation claims made in this paper given the use of mtDNA. Claims from previous papers (e.g., Theo) are already being misapplied. Their mtDNA is now included in UN Biodiversity planning despite the fact that it is not useful for conservation. In conservation we are interested conserving adaptive potential. Neutral genome-wide genetic diversity is not well correlated with adaptive potential and organellar genetic diversity is not correlated with either. Strong conservation claims made from analyses of this marker are thus dangerous.

Minor thoughts

L131: Why remove cells above 60? Shouldn't we expect evenness to be very high here because it's a specialized environment for an insect?

L148: What is the random effect?

L150 What is spatial habitat variation?

L237-238: Disagree—neutral theory can predict positive or negative SGDCs. See Laroche et al. 2015 AmNat A neutral theory for interpreting correlations between species and genetic diversity in communities

L310: Why should evenness be correlated with GD or SR. Also these regions appear to be hotspots for different taxa, which are a minority of this dataset

319-322: Environmental filtering/ niche differentiation is explicitly a non-neutral community assembly process. Higher GD due to stronger local adaptation is a stretch I think. Especially because the size of these grid cells precludes any "local" interpretation

L329-332: Or if you missed sampling populations

L364-5: Value of putatively neutral diversity is being debated but not the conservation value of using a single mitochondrial gene (The Peril of Gene-Targeted Conservation Kardos & Shafer 2018 TREE)

Reviewer #3 (Remarks to the Author):

This is an interesting study. It is hard to know whether the results would be robust to issues like (a) alignment errors, which could impact the estimates of diversity, and (b) the presence of NUMTS in the dataset (which are unfortunately common). Unfortunately, NUMTS can be more than 10% different from functional copies of COI, which can lead to overestimation of diversity. I really like this study but think the authors should address this---without these details it is hard to know how big of an impact this had on their other conclusions. As such, I would suggest this could be considered major revision.

Reviewer #4 (Remarks to the Author):

The "Global determinants of the distribution of insect genetic diversity" by French et al is an interesting, well-written study that explores and reviews the mechanisms underlying global patterns of intraspecific genetic diversity. The authors introduced the limitations of current knowledge of global patterns of intraspecific genetic diversity for insects compared to other charismatic lineages such as birds, mammals or reptilian, introduced their results and provided several mechanisms that may explain the patterns found.

I have enjoyed the manuscript and it can make an important contribution in the field. The objectives are clear and ambitious, the methodology used is novel and the models well explained, and the results discussed are clear and relevant for improving the approaches for understanding global patterns of diversity of insects. One of the main implications of the study is relevant, the factors driving species and intraspecific genetic diversity are different. Also the declining poleward pattern of intraspecific genetic diversity is unexpected, and the mechanism underlying this pattern are well discussed. The implication of the study for further global macrogenetic studies is high and this study is important for understanding macroecological patterns of GD. I have few main comments and a short list of minor comments.

Main comments:

(1) In the Discussion (lines 231-235) authors argued that patterns of intraspecific genetic diversity are different from those driving the classical negative latitudinal gradients in species richness, which is the pattern expected for the readers. I am wondering if authors tested the pattern of the diversity of OTUs that were used to estimate GDM and GDE, which should reveal inconsistencies at the two organizational level (species and populations). I guess that much more effort in sequencing has been done in temperate than tropical regions and the patterns of OTUs used can reveal these differences, but I did not find this results. Here, the criteria for selecting an OTUs was at least three specimens sequenced (447-448), which sounds fine, however in temperate regions I guess that a higher number specimens are sequenced than in the tropics and therefore values of GDM can reveal differences in sequencing effort because higher number of alleles may mean higher estimates of GDM. I am not sure how authors can distinguish the effects of sequencing effort among regions on the results. Also the authors can test SGDC for each grid cell for determining the relationship between species and intraspecific genetic diversity, which is

missed in the current version of the manuscript.

(2) Some sections of the Discussion sounds too speculative and based on results found in other studies, and I miss a clear discussion of the importance of current environmental variables in the study. For instance, Fig 4 shows how MTWM and precipitation seasonality are relevant for determining GDM, whereas these two variables together with PWM are critical for GDE. However, I miss a discussion of the implications of this result. In contrast, authors discussed the effects of Last Glacial Maximum, species' range size, habitat specializations, life history traits, insects diapause, but unfortunately these explanations were not related to current environmental conditions, despite its importance (lines 162-167). I suggest authors to explicitly discuss the importance of current environmental climatic variables in further versions of the study

(3) Taxon-specific patterns of GD are weakly discussed, I suggest authors add a paragraph in the Discussion in this aspect.

Minor comments:

- line 35: provide more details on "both GD metrics follow a bimodal latitudinal gradient"
- line 228: add "of the equator (fig. 3)"
- lines 229: change to "(LGM, <60° latitude)"
- line 249: change "consistent" to "opposite"
- line 434, add "Fig 1d)".
- line 903. Check the name of the journal

- figure 1: It is a great figure with great schemes for showing the meaning of GDM and GDE. However, I am not sure if "low GDM/ low GDE" is realistic, and I may remove this option in the figure or at least suggest how rare it should be. For clarity, in (d) I suggest increase values of GD (Y-axis) in High GD/HighGDE and I suggest reduce values of GD (Y-axis) in "low GDM/lowGDE".

- figure 2, in line 600, change to "(e, f)". To facilitate the understanding of figure 3, I suggest add the latitude values in the figures.

- figure 3, I guess the values of latitude (X-axis) are incorrect, I guess values should rank from 50 to 75°. I suggest also to add this values on the maps in figure 2. In line 615-616: "while GDM showed a qualitative positive quadratic trend (d)..." change to " trend (a)..."

I hope these comments help to improve the clarity of the manuscript.

Reviewers #1 and #4 were asked to comment on the issues raised by reviewer #2 using the following questions:

1. Has the other reviewer raised technical issues related to the use of mtDNA that you feel are important to address? Do you disagree with any of their criticisms?
2. Do you think the mtDNA data analyzed are appropriate for addressing the questions posed?
3. Does the other reviewer's comments alter your stance on the conceptual advance and/or novelty of the study?

Response from Reviewer #1:

Indeed, the remarks raised by my colleague are definitely grounded and cannot be dismissed. The recent literature cited by the referee is absolutely relevant.

I would still not subscribe that mtDNA diversity, in general, bear *no* information about demographic and evolutionary processes, and, in particular, I do not agree that neutral genetic variation is useless. The authors do not claim to measure adaptive potential, they (with shortcomings that the colleague and myself have highlighted) attempt to test for historical and demographic hypotheses.

In any case, I strongly agree that the limitations brought forward by the referee and the literature she/he cites should be much better acknowledged in the study, along with the highlighted technical issues entailed by the use of raw genbank/bold data (e.g. the bias due to the fact that many authors only deposited unique sequences in databanks).

I also completely agree on the suggestion to explore Millette et al's approach of considering species identity (which I should have mentioned myself), while it looks to me that the issue of aggregating populations into "arbitrary" units mentioned by Pas-Vinas is more a concern regarding other details of Millette's approach.

Already in my original report I made clear that the study requires substantial upgrade before being considered for publication. The concerns raised by the colleague certainly do not improve my judgement, but I would not say that, in principle, I am against publishing any such study just because it is based on mtDNA.

Response from Reviewer #4:

1. Has the other reviewer raised technical issues related to the use of mtDNA that you feel are important to address? Do you disagree with any of their criticisms?

In general, I partially agree with the other reviewer. In fact, there are other markers or genomic techniques that can be used to study population genetics as the other reviewer has listed, however the manuscript focuses on intraspecific genetic diversity across species at a global scale, which is recently called macrogenomics. In order to do it, the authors used available sequences of cox1 that covered the world, which is probably the only fragment that is massively sequenced and can be used in a global scale study of intraspecific genetic variation.

Although the other reviewer is right in "mtDNA are not neutral, do not recombine, are not well correlated with demography, are not correlated with neutral genome-wide diversity or adaptive potential", cox1 provide an accurate species delimitation because the genetic divergence clusters species separately and the cladogenesis is well captured. Moreover, intraspecific genetic diversity and structure of cox1 corresponds to biogeography and dispersion (e.g., Papadopoulou et al, ME, 2009; Baselga et al., Nat Comm, 2013; Múrria et al, ME, 2017) and this fragment is widely used in macrogenetics (Leigh et al, Nat Rev Gen, 2021). Unfortunately, data available for neutral markers such as microsat or the mt genome of *Drosophila* that the other reviewer suggested, is limited and currently it is available for a small number of species, but the data available is far for a global scale study. Moreover, I have my doubts if microsat data can be easily compared across species because the markers used are commonly specific for each species. Until genome information is not available for thousands of species at the population level, i.e., several individuals sequenced for each population, we cannot use mt genome sequencing in macrogenetics. Also the authors are not clearly interested in adaptive genetics or general genetic diversity, as the other reviewer indicated, instead authors want to test macroecological patterns of intraspecific genetic variation for as much number of species as possible covering the widest area. For this reason, authors specifically analyzed the intraspecific DNA barcode diversity, which is clear in the manuscript.

On another hand, I agree with the other reviewer that one main problem is the number of available sequences that limits the analyses and conclusion that authors can perform. As the other reviewer suggested, I also agree the authors should introduce the limitations of the study:

- the "poor" availability of sequences at the global scale increases the grid cells at a 193 km x 193 km equal-area resolution, which is apparently a large size for a population genetic study. Also there are "only" 187 regions accumulated mainly in Europe, North-America and East-Australia, which should introduce some bias to the results. These limitations should influence the results found.

- I suggest authors increase the explanations of the limitations of using DNA barcoding data in population genetics studies that the authors briefly introduced in lines 79-82. As the other reviewer have pointed, I suggest include some of the limitations pointed by Paz-Vinas et al 2021

Ecol Let; Galtier et al 2009 ME.

2. Do you think the mtDNA data analyzed are appropriate for addressing the questions posed?

Despite the mtDNA data analyzed has limitations, I think the data used can address the questions posted and the study is well-performed. However, assuming the limitations of cox1 data, I suggest authors rephrase the hypothesis based on the comments by the other reviewer. In fact, some of the hypothesis based on demographic processes cannot be directly tested using the available data. Improving this section is critical. As the other reviewer, I agree in "I suspect that there are many interesting processes related to the biology of mtDNA underlying some of the patterns detected." I may focus the new hypothesis on these processes.

Another interesting criticisms by the other reviewer is the metric used: mean GD in each grid. Authors must to argue extensively why this metric is appropriate, and I think it is. Moreover, authors can explain if they used the abundance of each haplotype in a grid or only the data of each unique sequences, which can introduce bias as the other reviewer suggested, specially when authors used the mean GD. This is also critical.

3. Does the other reviewer's comments alter your stance on the conceptual advance and/or novelty of the study?

No, I still think of the high contribution of the manuscript. However, the authors should clearly explain the limitations of the data used.

REVIEWER COMMENTS

We thank the editor for considering a revised version of our manuscript with updated title, “Global distribution of insect genetic diversity.” We have implemented all of the suggestions of the four reviewers, which we believe has led to a greatly improved manuscript. In summary, we have:

- Clarified the description of the modeling workflow to make it easier to reproduce.
- Added sensitivity analyses for five additional minimum OTU thresholds (>10, >25, >50, >100, >150, >200), which demonstrate qualitatively similar results.
- Added simulation experiments which show that any potential bias due to missing duplicate alleles is negligible across the relevant parameter space of our models.
- Simplified our modeling approach by removing an extra step of model selection, which extends our initial results based on contemporary climate variables to show that the genetic diversity of insect assemblages has also been significantly influenced by historical climate change.
- Revised our previous discussion to place findings in the context of the eco-evolutionary processes underlying Rapoport’s rule, varying range sizes, and late Pleistocene population histories.
- Added a discussion on the advantages and limitations of using mtDNA for global macrogenetics studies.

We appreciate the attention dedicated to our work and thank the reviewers for their time and willingness to provide extensive and constructive feedback. Below, we give point-by-point responses to each set of reviewer comments.

Reviewer #1 (Remarks to the Author):

The manuscript presents a very welcome analysis of global distribution of intraspecific genetic diversity in insects, for which the authors selected and aligned a large number of mtDNA sequences from the BOLD database. It is generally very well written.

Author response: We thank you for the interest and giving this a careful read with many helpful comments. We’ve implemented all suggestions, notably we simplified the modeling approach and communicated uncertainty/extrapolation in model predictions and data filtering choices more clearly. In addition, we revised our approach to assessing the latitudinal GD gradient. The manuscript was greatly improved with your suggestions.

My main perplexity concerns the analytical approach, which I found rather confusing. In particular, the authors recognize the importance of accounting for spatial autocorrelation (SAC) in their analyses. However, as far as I understood, they did not include terms for SAC in all of their analyses and they performed multiple steps of model selection, with and without accounting for SAC. I did not actually understand the logic of this approach and I don’t see how accounting for SAC throughout the whole analysis could not be a better solution.

I also don’t really understand how the “final” model selection (test of the current/past climate hypotheses) step was applied. First of all, it is not clear whether a “null” model (e.g. with an

intercept and a SAC term only) was also tested. It was done elsewhere in this study and I think it should be done here too. Second, what is the actual meaning of first selecting a subset of predictors based on the importance of their effect on the response and, then, dividing the surviving predictors into groups based on a completely new criterion (current vs. past, in this case) and perform another round? Personally, if past climate variables were retained in the final models, I'd assume they have a significant effect on the response, and I'd like to know what their effects on the response are.

Author response: Thank you for pointing out this confusion. In retrospect, we agree that we should use a more consistent and clear modeling workflow with only one round of model selection. While integrating spatial autocorrelation in the analysis from the beginning would be ideal, existing approaches are inappropriate and/or unfeasible for the present analytical context.

To present a more clear and consistent modeling workflow (as suggested) we omitted the second round of model selection from the modeling pipeline and only performed projection predictive selection followed by a t-distributed random fields GLMM regression. This is warranted because we used regularizing priors on the beta coefficients, which will shrink any non-informative predictors to zero. Finally, this allowed us to use a final set of predictive models that accommodated spatial autocorrelation.

Another general issue is whether the geographic density of the cells that were considered in the analysis is sufficient for the global-scale extrapolation that was eventually performed.

Author response: Thank you for bringing this up. We assessed modeling and prediction error in multiple ways to explore the impacts of extrapolating to geographic areas beyond our sampling (see par 1 of the Methods section "Modeling approach", par 2 of the Methods section "Global genetic diversity map generation", Fig. 2 (b,d,f), Supp. Fig. 2, Supp. Fig. 8, Supp. Fig. 9, text reproduced below). In addition, in response to the comment below, we explored additional OTU filtering thresholds, which impacted the geographic density of cells sampled (Supp. Fig. 1). Our responses to the comments below will expand upon these statements. In the end we found that using a threshold of 100 OTUs per cell (relaxed from 150) resulted in better performing models and a large increase in geographical sampling, especially noticeable in parts of Africa. Additionally, we found that this threshold balanced per-cell and global sampling magnitude with per-cell and global sampling variance.

Par 1 of "Modeling approach": "To assess the predictive ability of the model, we split the data set into 75% training and 25% testing sets, stratifying the sampling by continent to maximize spatial representation of sampling in both data sets. All model selection and model fitting was performed on the training set, while predictive performance of the best fit model was assessed by predicting the withheld test data set."

Par 2 of "Global genetic diversity map generation": "To avoid making poor predictions into areas with environments that are non-analogous to the areas used to train the models, we used multivariate environmental similarity surface (MESS) maps (Supplementary Fig. 9). MESS maps

visualize how environmentally similar or different areas across the globe are compared to the model training data 167. Subsequently, we used the MESS results to mask areas with non-analogous environmental space (values less than 0) on our global prediction maps, indicating areas with high prediction uncertainty.”

As I wrote, although the details of the procedure remain rather unclear to me, it seems like spatial autocorrelation has been accounted for, which is reassuring. However, I would appreciate a more careful effort to explain how reliable the extrapolation may be. As the authors highlight the importance of being able to cast predictions over huge unsampled areas, some sort of cross-validation of predictions would be necessary.

Author response: This is an excellent point and we agree with the need to communicate the impacts of extrapolation more clearly. As such, we have now added a model validation step, where we split our data into 75% training and 25% testing sets, where training and testing samples were stratified by continent to ensure the most even geographic sampling possible. The test data was withheld until a final model was selected, then the test data was predicted with the model. We include multiple ways of assessing the impact of extrapolation:

- *Observed vs predicted scatterplots of the test data show little evidence of spatial bias in predictions, especially at the presented minimum OTU threshold (100) (Supp. Fig. 3),*
- *Maps of the predicted values are masked, removing areas where the environmental space is outside the range of variation used to train the models (Fig. 2, Supp. Fig. 8),*
- *Maps of model residuals show no spatial bias or autocorrelation, which are corroborated with Mantel’s I tests (Supp. Fig. 6, Table 2)*
- *Multivariate environmental similarity surfaces (MESS) show regions where the environmental space of the predicted area is outside the range of the data used to train the models (Supp. Fig. 9),*
- *Maps of model 95% highest density intervals (model error) across the globe (Supp. Fig. 8), indicate areas with potential geographic bias in prediction error, although areas with high error are relatively localized*

In this respect, I was perplexed by the choice of filtering the data to only include cells with at least 100 OTUs. On one side, I see that the choice is somehow conservative, and may be the best thing to do, but is really leaving only a handful of cells outside North America and Europe. As a reader, I would really like to see how things change if the threshold is lowered and, at any rate, a deep discussion concerning the dramatic lack of sampling throughout most of the world.

Author response: Thank you for this comment. We agree that we were being overly conservative and that it would be useful to explore the sensitivity of the results across different thresholds. Therefore, we performed the above modeling workflow for different minimum OTU thresholds- 10, 25, 50, 100, 150, and 200. The overall conclusions of our analysis were not significantly impacted by OTU threshold, with the most dramatic departures at the extreme

filtering scenarios, 10 and 200, with both cases yielding much poorer model fits and performance. For instance, similar environmental variables were selected to be predictive of GDE. For each threshold, all global predictions were highly correlated ($r > 0.80$), and performance on test data was generally high.

While the GDM results were a little more sensitive to the minimum OTU threshold, the results were still qualitatively similar across most OTU thresholds. As suggested by the reviewer, another important benefit of lowering the minimum OTU threshold was that global sampling dramatically increased from 193 cells to 245 cells given the minimum threshold of 100 OTUs. For the sake of narrative cohesiveness, we opted to highlight in the main text the threshold with the model that performed best on withheld test data (minimum of 100 OTUs) and referred readers to supplementary material for the results given all of the other OTU thresholds (10, 25, 50, 150, and 200).

Very importantly, most of the discussion is devoted to explaining what a “bimodal” latitudinal trend in genetic diversity may mean. However, as far as I understand, the authors did not find evidence for a latitudinal trend in GDM. Indeed, after (somewhat inelegantly, in fact) removing part of the data, they detected a signal for GDE, but not GDM, but they discussed their results against predictions and previous findings which, all, concerned GDM, not GDE. Conversely, the interpretation of the (statistically significant) relationships of GDM/GDE with climate variables are barely discussed! I argue that this important portion of the discussion/study deserves deep rethinking.

-Author response: Thank you, on reflection we agree we may have over interpreted the latitudinal gradient. We opted for an alternative, simpler, approach to determining the relationship between GD and latitude. Following the reviewer’s excellent point, we chose not to remove data from higher latitudes. Using spatially modified correlations, we now find a negative quadratic relationship between GDE and latitude, but no relationship between GDM and latitude (Table 1). In our revised manuscript, we interpret this in light of arthropod, vertebrate, and plant biodiversity patterns, and remove discussion of a “bimodal” gradient within the text (see Discussion, multiple sections)

-We also appreciate the reviewer's suggestion to include more discussion on the statistically significant relationships of GDM/GDE with environmental variables), which we now do in the Discussion:

“In contrast to the declines of GDE in previously glaciated areas, it could be that the peak of GDE in the subtropics, and more generally the correspondence of GDM and GDE with seasonally high temperatures, is the result of climates which have remained stable since the LGM (Fig. 4). Moreover, GDM and GDE have contrasting relationships with seasonally high precipitation (PWM), where GDM increases with PWM, while GDE decreases, corresponding with inflated GDE in arid, hot regions rather than the wet tropics (Fig. 2, 4).”

“One possible explanation is that these areas are more climatically stable and hence may harbor more uniform demographics that manifest as elevated GDE values.”

“...increasing GDE with latitude may be driven by uniformly larger range sizes that result from greater physiological tolerances in harsher environments. While this is consistent with both GDM and GDE being most strongly associated with the seasonally hot conditions (MWTM) found in the subtropics (Fig. 4), this relationship drastically breaks down in the colder temperate and subarctic regions (Fig. 2) that frequently freeze, especially for GDE (Fig. 4).”

“We are hypothesizing that both larger ranges and climatic stability could have a positive effect on GD, yet they are different processes and do not necessarily co-occur. For example, a climatically stable area might also coincide with small and variable range sizes, and conversely an area that experiences cycles of glaciation may also coincide with many large ranged species. But in both cases, GD is predicted to be deflated and it is only in the “sweet spot” of areas with many large ranged species occupying relatively stable climates does one predict elevated GD consistently across species, to yield higher values of both GDE and GDM.”

A relatively minor point is that, while the addition of Genetic Diversity Evenness (GDE) to the more usual “mean” GDM is introduced as a very important step forward compared to previous similar studies, there is no mention in the Introduction about why exactly this is the case in terms of which patterns of GDE should be expected under different hypotheses. In fact, there is some (interesting) discussion about expectations concerning GDE, based on the results from Overcast et al. 2020 (now published in MER). However, 1) the importance of measuring GDE in terms of which hypotheses may be tested by using it should be clarified in the Introduction, so that the reader may immediately understand it; 2) by reading the paper by Overcast et al., I did not have the impression that the GDE computed here is actually the same as the qD in Overcast et al (as the latter contains a term for species abundance): I think this should be carefully explained.

Author response: The GDE we employ is based on the commonly used metric in macroecology and community ecology, and our use for genetic data here is identical to the measure introduced by Overcast et al. 2021 (in that study, evenness was calculated from abundances, trait measures, and genetic diversities). However, we thank the reviewers overall point here and we agree that we should provide a more comprehensive justification for the use of GDE for macro-genetics beyond the fact that both measures yield more information from the data than GDM alone. Our expectations of GDE given different processes draws on classic macroecology, comparative phylogeography as well as process-explicit simulation-based studies by Overcast, and we now expand upon this in the discussion:

“Areas with higher levels of GDE could also partially emerge from different levels along the continuum of fundamental community assembly processes 120. Although the spatial scale is not always in line with the grid-cells we employ, Overcast et al. 75 found in simulated and empirical

(arthropod, annelid, and trees) communities to have elevated GDE under ecologically neutral conditions in contrast to non-neutral, i.e., niche-structured conditions. In these specific model-dependent cases, the lower GDE in communities assembled via environmental filtering is likely caused by increased genetic diversity in hyper-dominant species with stronger local ecological adaptation. In line with the process-explicit modeling, our observation of GDE increasing from the tropics to the subtropics shows that equatorial insect communities may have stronger local niche-structured mechanisms (i.e., less ecologically neutral conditions), than subtropical temperate insect communities. This would be consistent with the idea of stronger niche conservatism in the tropics. 121”

Other aspects of the discussion were not very well explained or (to my advise) lacked a strict logical consistency. I have noted this in a commented version of the manuscript's .pdf, which I attach and which also contains more sparse comments.

Author response: We thank the reviewer for this additional effort and have copied the comments to the bottom of this document, adding our responses. It is under the heading “Reviewer 1 commented manuscript response”.

All in all, my suggestion for the authors is to 1) streamline and adjust the statistical analysis (especially wrt the treatment of SAC), providing thorough justification for each step and 2) rework the manuscript so that the discussion starts from well-defined expectations to be explained in the introduction, concerning both the GDM and GDE metrics.

Reviewer #2 (Remarks to the Author):

This is an ambitious series of analyses that sets out to map and predict the global distribution of the mean genetic diversity and the evenness of genetic diversity in insects globally. The authors analyze >2.3 million sequences mitochondrial CO1 gene from >95K species sampled across 187 grid cells (193km x 193km) grid cells. This is an impressive number of sequences and species and there have been no macrogenetic studies of insects to date. Questions about the broad spatial patterns in intraspecific genetic diversity are, as the authors note, interesting, timely, and important for conservation.

Author response: We thank the reviewer and agree that this was a worthwhile endeavor that will be generally relevant and an important starting point for looking at insect genetic diversity at the macro scale.

While I am supportive of and interested in the questions addressed in this paper and think analyses of these data can certainly be made publishable, I cannot support the publication of the paper in its current format. My primary concern is that the data analyzed are inappropriate for addressing the questions posed. mtDNA are not neutral, do not recombine, are not well correlated with demography, are not correlated with neutral genome-wide diversity or adaptive potential (e.g., Allio et al 2017 Mol Biol Evol; Edwards and Bensch 2009 Mol Ecol; Galtier et al. 2009 Biol Lett; Galtier et al 2009 Mol Ecol). For example, in *Drosophila*, the mt genome plays an outsized role in clinal adaptation and is suspected to be heavily involved with climatic adaptation (see Quantifying the relative contributions of the X chromosome, autosomes, and mitochondrial genome to local adaptation Lasne et al. 2019 Evolution). Such data are unsuited for testing demographic hypotheses should be analyzed and

interpreted as if they represent adaptive genetic diversity (see Yiming et al. 2021 Ecography). The firm conclusion of the Galtier et al 2009 ME review and references therein are that mtDNA cannot be used as a general population genetic measure of the diversity of anything but mitochondrial DNA.

Author response: While we acknowledge the reviewer's point about mtDNA, we think that this opinion is overstated in the context of a macrogenetics study of this scale. The stated shortcomings were currently acknowledged in the previous version of the manuscript, but they do not outweigh the practical advantages of being able to use mtDNA on such a large global scale and are not in line with evidence that mtDNA patterns do generally relate to important processes beyond "anything but mitochondrial DNA", (with a healthy dose of probabilistic uncertainty).

The Edwards and Bensch 2009 paper was part of a vigorous debate about the merits of mtDNA data for the inference of detailed complex demographic histories used for phylogenetic/phylogeographic reconstruction used at that time. That particular debate has long been settled, but we are asking much less of this locus in calculated genetic diversities rather than reconstructing the detailed demographic history of each OTU. Rather, we argue that aggregating large numbers of OTUs per grid cell at large geographical scales reflects patterns of mtDNA that could relate to coarse-level processes that span demography, selection, and selectively neutral genetic mechanisms.

For a good example of connecting intraspecific mtDNA diversity to coarse-level population histories, a very recent macrogenetic study of ~15,000 European butterfly mtDNA sequences from ~300 species (Dapporto et al. 2019, <https://doi.org/10.1111/1755-0998.13059>) found a pattern of depleted diversity in previously glaciated areas, a classic prediction of founder effect expansion after the last glacial maximum (i.e., the classic idea championed by the late Godfrey Hewitt). Rather than this observation only being related to the mitochondrial DNA of butterflies as the reviewer here suggests, it is more likely that the community history of post-Pleistocene expansion had a detectable impact on mtDNA diversity patterns. One could counter that we are merely seeing the effect of linked positive selection occurring in the previously glaciated areas, yet these two processes (demographic founder effect and positive linked selection) are not mutually exclusive and are both worthy of discussion.

Another example of mtDNA data reflecting important features of evolutionary history is from humans. While mtDNA data from humans are unable to detect such things and admixture from human/Neanderthal interbreeding, the genetic diversity and divergence patterns of human mtDNA have always reflected origins in Africa with subsequent expansion into the rest of the planet, something that has largely been corroborated with whole genome data.

While we strongly acknowledge the study by Galtier et al. 2009, observed mtDNA patterns reflect combined demographic and selection processes in addition to a large stochastic component (i.e., the coalescent). This is consistent with the general observation of a large amount of noise in macrogenetic studies (not to mention variation in mutation rates and generation times etc).

The other papers the reviewer cites are interesting studies that we now cite:

“These single-locus markers have drawbacks: notably, the genetic diversity that mtDNA captures is from a single draw from the stochastic coalescent process operating within a lineage and may be subject to selection in addition to neutral demographic processes 62–67.”

Allio et al. 2017 shows that mtDNA rates are variable and faster than nuclear DNA rates, which in itself does not contradict any specific inferences or conclusions we make. However, variable rates could have impacted the observed patterns and we now cite this paper (see above) to highlight this potential confounder. The results of Lasne et al. 2019 are only based on a single taxon and therefore should not be used to make generalizations across all insects. The study of Yiming et al. is especially interesting, but the data is MHC, a locus that is thought to be impacted by balancing selection (which would not specifically impact haploid mtDNA).

Text from the manuscript: “In contrast, species richness and GD may be decoupled due to the “cradle” hypothesis that predicts higher speciation rates, more population structure, and smaller range sizes leading to Rapoport’s Rule, the tendency for species’ range sizes to increase with increasing latitude. Second, we might predict that the influence of Late Quaternary climate fluctuations to have an impact on GD through population demographic processes influenced by cyclical variation in precipitation, temperature, and glaciation patterns where areas with more stable climatic histories are predicted to have increased GD. Finally, we consider the influence of human disturbance on patterns of assemblage-wide GD, which we expect to decrease in magnitude in areas of high human influence.”

Each of these hypotheses acts through demographic processes acting on populations. mtDNA is not suited to test them.

Author response: We agree that mtDNA is ill suited for detailed inference of complex demographic and/or phylogenetic history. As laid out in our previous response (and examples) above, basic coarse-level demographic processes are completely decoupled from mtDNA diversity patterns.

The Galtier review is particularly compelling and comprehensive and those criticisms of using mtDNA as a measure of general genetic diversity have, to my knowledge, only grown stronger. If the authors disagree with this substantial literature and feel pragmatism is warranted those criticisms need to be directly confronted in text if the reader is to be able to evaluate the paper on its own. The strong conclusion of the paper can be summed up in this quote: "For all these reasons, mtDNA is perhaps intrinsically the worst population genetic and phylogenetic molecular marker we can think of."

Author response: We thank the reviewer for pointing out that we should clarify the limitations of connecting mtDNA with demographic history. We acknowledge that it is a noisy relationship, yet not an empty one"

"These single-locus markers have drawbacks: notably, the genetic diversity that mtDNA captures is from a single draw from the stochastic coalescent process operating within a lineage and may be subject to selection in addition to neutral demographic processes 62–67."

"While intraspecific mtDNA diversity data is insufficient for making detailed inference of demographic history or phylogenetic reconstruction 67, we instead treat it as an important variable to study on the macroscale, rather than a population genetic marker. Macrogenetics is a relatively new field and thus basic patterns are still not established. Our study is a step in this direction and suggests avenues for ways to test specific hypotheses."

The authors of this manuscript correctly point out that their approach follows the methods of previous studies, some of which were published in this journal (e.g., Theodoridis et al. 2020). These papers too, to varying degrees, did not address the substantial shortcomings of using mtDNA to estimate genetic diversity. Galtier also predicted the continued use of mtDNA as a population genetic marker:

"Despite the many concerns we express about the use of the mitochondrial marker in molecular ecology, we are confident that scientists will keep on analysing mtDNA variation in the wild. The reason why we think mtDNA will keep its popularity is its practical convenience".

Author response: The paper by Galtier et al. made some important findings, but was specifically referring to the use of mtDNA for single species population genetic studies and/or phylogenetic studies. In contrast, we are not using mtDNA as a population genetic marker in the traditional way Galtier et al. are referring to. We are using available mtDNA for a macrogenetic study aggregating intraspecific diversity patterns from large numbers of co-distributed OTUs and correlating this with a suite of environmental variables. Using these observed correlations to infer eco-evolutionary and demographic processes is beyond the direct scope of our study or any other macrogenetic study to date (with the possible exception of Exposito-Alonso et al. 2022, DOI: 10.1126/science.abn5642). Instead, we are using the results to discuss possible explanations of these correlations (i.e., we are not inferring them with a process explicit model; see DOI: 10.1126/sciadv.abj2271 for review). Therefore, our study is a valuable and important first step that generates hypotheses that will inspire and lead to future studies.

Despite my harsh criticisms of mtDNA work, I do appreciate there is value in exploring global patterns of mtDNA diversity. I suspect that there are many interesting processes related to the biology of mtDNA underlying some of the patterns detected. However, the sentiment the impressive volumes of data for mtDNA allow these markers to overcome their shortcomings is, to the best of our knowledge, not well supported by data. If mtDNA cannot measure the quantities of interest increasing the volume of data will not help (Paz-Vinas et al 2021 Ecol Let; Galtier et al 2009 ME).

Author response: We are glad that the reviewer found value in our study that uncovers statistical correlations between mtDNA GD and environmental variables. Following the logic of Pilowski et al. 2022, our phenomenological model finds some unexpected correlations that can suggest the mechanisms that produce them. Actual causation (be that of regional demographic histories or histories of selection) are points of discussion and ultimately hypotheses that are testable by process-based modeling in future studies, not points of direct inference or conclusion.

Putting aside the appropriateness of this measure of mDNA as a population genetic marker there are also significant methodological issues with this work. Again the authors have followed previous studies but these studies share the same flaws. Schmidt and Garroway 2021 (Con Gen) note that mean genetic diversity of arbitrary numbers of species aggregated from within arbitrary sized grid cells is not a meaningful population genetic or conservation relevant quantity. This and the previous studies the authors followed take the mean genetic diversity of across all species sampled per cell regardless of the fact that the species to be averaged will have extremely different life and evolutionary histories, which will produce different equilibrium genetic diversities, and the fact that mtDNA mutation rates can vary extensively.

Author response: We appreciate the argument from Schmidt and Garroway 2021, but note that we do at least restrict our study to insects and that our second summary statistic is actually a way to measure the variance in GD that can come from variation in life and evolutionary histories (GDE).

Additionally this method of data aggregation means that different numbers of species and indeed different species are being compared. Millette et al grouped mtDNA sequences by species within grid cell and I suggest the authors explore this method. It too is not ideal but here nucleotide diversity is interpretable. The authors are careful to call their metric mean genetic diversity until the discussion where they slip into making inferences about intraspecific variation. Again the previous studies call mean genetic diversity intraspecific genetic diversity too. I cannot see how the mean of multiple species gives information about intraspecific variation. The Millette approach is a measure of intraspecific mtDNA genetic diversity—this does not alleviate concerns about the inappropriateness of mtDNA as a population genetic marker.

Author response: Our GDM and GDE are both derived from the classic average pairwise distances (per OTU), which is sometimes referred to as nucleotide diversity and is essentially a useful measure of intraspecific or intrapopulation or intra-location per species variation in genetic diversity (depending on what unit one chooses to sample). GDM is the average of this per grid cell and GDE is the evenness. It has desirable and robust sampling properties (as

opposed to allelic richness). We clarified this by saying “mean intraspecific variation” instead of just saying “intraspecific variation” alone.

Pas-Vinas et al note that arbitrarily aggregating sequences across large spatial areas create implausible populations. This is true in the extreme. Populations are not defined genetically and contain multiple species. Most inferences in this paper refer to population level processes (evolutionary change, relationships to environments, drift etc).

Author response: We agree that summarizing genetic variation across OTUs within arbitrary grid cells is a compromise out of pragmatism. We agree with the reviewer’s point brought up by Pas-Vinas et al is an interesting dynamic to consider for macrogenetic studies. Macrogenetic studies are inherently different from single species population genetic studies with different goals and approaches but there is a shared inherent abstraction of what a “population” is in relation to a geographically defined sample. They are rarely congruent, yet methods based on population genetic theory can still yield useful results despite that populations are rarely homogenous or well defined (Barton et al., 2019; <https://doi.org/10.7554/eLife.45380>).

Following this interesting point suggested by the reviewer, we now quantify and report that the vast majority (~75%) of the sampled OTUs are only found in single grid-cells (Supp. Fig. 11). Coalescent theory with migration (i.e., the structured coalescent; Nordborg and Krone 2001 and Nordborg 1997) predicts that local sampling can capture variation associated with unsampled areas and that this can lead to higher average pairwise distances (i.e., higher GD). We don’t see this as a bug, but as a feature that enables us to potentially capture information about species with larger and/or more isolated ranges. In relation to this important dynamic brought up by the reviewer, we now cite a recent macrogenetics paper that directly looks at links between range size and GD empirically and theoretically (Alonso-Espisito et al. 2022).

Paz-Vinas also showed that sequence deposition is often incomplete and idiosyncratic authors tending to deposit only new sequences in a region. This too will significantly affect their measures of genetic diversity.

Author response: We are well aware that authors (and participants in the BOLD initiative) neglect to deposit duplicate allele copies, and this is one reason why we chose our metric of GD to be average pairwise distances. This metric has several desirable statistical sampling properties (Tajima 1983) including approaching its expectation with only 5-10 samples. Therefore, missing duplicate samples, while positively biasing this summary metric will likely have a negligible effect. As mentioned to reviewer #1, we explore and verify this explicitly with simulations given a range of sample sizes and have included it in the new version of our manuscript, in the Supplementary Materials, and main text:

“To explore this sampling dynamic explicitly, we conducted coalescent simulation experiments comparing how the calculation of GD varies given identical samples with and without duplicate alleles removed. These simulations showed that retaining only unique haplotypes resulted in a small, consistently upward bias in GD values. Additionally, increasing values of effective population size (N_e) decreased this bias, with estimates of GD with and without duplicate alleles

converging for Ne values greater than $\sim 10e5$ (Supplementary Materials, Supplementary Fig. 15)”

The predictive maps look good. However, I am very leery about interpolating the globe from just 187 grid cells. The posterior checks look good but that just means the model can predict the data used to generate it, which is important! But predicting different continents with different species groups and geological histories with such a small sample seems fraught.

Author response: We have new results based on a more clear modeling procedure as well as a range of minimum threshold for number of OTUs power per grid cell. As with the original manuscript, we still do not extrapolate to environments beyond what are used to train the model. In addition, we added a model validation step to our modeling process, where we train the model and predict it to test data that was withheld. The data are stratified according to continent to enforce the most even spatial sampling possible. We highlight potential spatial biases in prediction by coloring observed versus predicted plots by continent and find no obvious spatial patterns in the residuals (Supp. Fig. 3 & 6). We do agree that our study has a biased focus on North America and Europe, and we now mention this in the Discussion:

“Sampling biases, including the overrepresented North American and European sampling we find, may also influence this relationship, although we find no evidence of an influence in our data (Supplementary Fig. 14, Supplementary Table 5).”

“Additionally, we find that OTUs occupy more grid cells between 40° and 60° latitude (Supplementary Fig. 14), matching our expectation for larger ranges at higher latitudes, although we acknowledge that sampling bias may contribute to this pattern as well.”

Finally, I am concerned with the conservation claims made in this paper given the use of mtDNA. Claims from previous papers (e.g., Theo) are already being misapplied. Their mtDNA is now included in UN Biodiversity planning despite the fact that it is not useful for conservation. In conservation we are interested conserving adaptive potential. Neutral genome-wide genetic diversity is not well correlated with adaptive potential and organellar genetic diversity is not correlated with either. Strong conservation claims made from analyses of this marker are thus dangerous.

Author response: We agree that conservation claims should be made with caution and therefore we do not make any strong claims regarding how our study should directly inform conservation decisions. While anything gained from our study should be done with care, macrogenetic studies like ours are a potential first step to help inform conservation goals (along with other information), which is why we cited DeWoody et al. 2021 and and Reed & Frankham 2003. Likewise, studies like ours could aid in monitoring efforts.

Minor thoughts

L131: Why remove cells above 60? Shouldn't we expect evenness to be very high here because it's a specialized environment for an insect?

Author response: *We agree and have now opted to include cells above 60.*

L148: What is the random effect?

Author response: *We removed this sentence and provide a sentence that briefly describes what the model does:*

"...we used Bayesian generalized mixed models (GLMMs) that account for spatial autocorrelation in the residuals to explain environmental relationships and make predictions"

We provide a more detailed explanation of the random effect/covariance matrix in the methods:

"SAC is modeled as a random effect with a multivariate t-distribution determining the shape of the covariance matrix."

And in the Supplementary Materials:

"Rather than estimating random effects at each location, which can be computationally intensive, a spatial field of correlated random effects at a subset of locations or "knots" was modeled. The choice of the number of knots is somewhat subjective, so we fit models using 5, 10, 15, 20, 25, and 30 knots. Models with 20 knots had the lowest residual SAC for GDE, while 30 knots had the lowest residual SAC for GDM, so we chose 20 knots for the GDE models and 30 knots for the GDM models. We used regularizing priors on all slope parameters ($N(0, 0.1)$) and sigma ($N(0, 1)$), as well as on the Gaussian process θ parameter ($N(0, 5)$), which controls how steeply the correlation between knots declines, and the Gaussian process sigma parameter ($N(0, 1)$), which controls the amplitude of spatial deviations."

L150 What is spatial habitat variation?

Author response: *We changed the language to reflect what we use in the Methods- habitat heterogeneity. It is the standard deviation of the Enhanced Vegetation Index across the area, a derived metric from MODIS. We provide this explanation in the methods:*

"The habitat heterogeneity metric was calculated as the standard deviation of the Enhanced Vegetation Index, which was derived from the Moderate Resolution Imaging Spectroradiometer (MODIS) (2.5 arc-min; 154)."

L237-238: Disagree—neutral theory can predict positive or negative SGDCs. See Laroche et al. 2015 AmNat A neutral theory for interpreting correlations between species and genetic diversity in communities

Author response: *This is an interesting point, yet the Laroche et al. 2015 model is not built for the macrogenetic scale we are in and instead incorporates community richness with the GD of a*

single focal species. Instead, we look to predictions from the local community-level population genetic model of Overcast et al. 2021 to help relate our observed deviations from SGDC expectations with the neutral and non-neutral processes that are explicitly incorporated in Overcast et al. 2021.

L310: Why should evenness be correlated with GD or SR. Also these regions appear to be hotspots for different taxa, which are a minority of this dataset

Author response: We have rephrased this text to further discuss how this relationship might be predicted with respect to relatively more climatic stability. Our new analysis also shows the latitudinal gradient of GDE holds significantly for two of the three orders that dominate our data (Diptera and Lepidoptera) and we now clarify that broad generalizations across all insects should be avoided in this context.

319-322: Environmental filtering/ niche differentiation is explicitly a non-neutral community assembly process. Higher GD due to stronger local adaption is a stretch I think. Especially because the size of these grid cells precludes any “local” interpretation

Author response: We agree and have rephrased to acknowledge this possible mismatch of scale:

“Areas with higher levels of GDE could also partially emerge from different levels along the continuum of fundamental community assembly processes 120. Although the spatial scale is not always in line with the grid-cells we employ, Overcast et al. 75 found in simulated and empirical (arthropod, annelid, and trees) communities to have elevated GDE under ecologically neutral conditions in contrast to non-neutral, i.e., niche-structured conditions.”

L329-332: Or if you missed sampling populations

Author response: This may be true, but this paragraph attempts to explain potential patterns in a biological context, rather than in the context for sampling errors. We discuss sampling issues in the last paragraph of the discussion as well as extensively in the methods.

L364-5: Value of putatively neutral diversity is being debated but not the conservation value of using a single mitochondrial gene (The Peril of Gene-Targeted Conservation Kardos & Shafer 2018 TREE)

Author response: We now acknowledge this important distinction and have now added this citation.

Reviewer #3 (Remarks to the Author):

This is an interesting study. It is hard to know whether the results would be robust to issues like (a) alignment errors, which could impact the estimates of diversity, and (b) the presence of NUMTS in the dataset (which are unfortunately common). Unfortunately, NUMTS can be more than 10% different from functional copies of COI, which can lead to overestimation of diversity. I really like this study but think the authors should address this---without these details it is hard to know how big of an impact this had on their other conclusions. As such, I would suggest this could be considered major revision.

Author response: Thank you for the positive words about our study as well as voicing both of your concerns with our genetic data (which we address below).

In response to a), we did not find concerning patterns in alignment statistics and visually inspected a random sample of alignments and found no issues. After filtering, we also inspected alignments with pairwise nucleotide difference values at the tails of the distribution and found no evidence of errors:

"We used default settings in Clustal Omega (v1.2.3) to align the sequences and visually assessed both a random subset of alignments and alignments with genetic diversity values at the tails of the distribution to check for alignment errors 140."

In response to b), BoLD flags stop codons and prevents submission of sequences with them. BoLD has rigorous procedures to weed out sequences with stop codons. Therefore, we can safely assume that NUMTS are not a serious problem with the data.

Reviewer #4 (Remarks to the Author):

The "Global determinants of the distribution of insect genetic diversity" by French et al is an interesting, well-written study that explores and reviews the mechanisms underlying global patterns of intraspecific genetic diversity. The authors introduced the limitations of current knowledge of global patterns of intraspecific genetic diversity for insects compared to other charismatic lineages such as birds, mammals or reptilian, introduced their results and provided several mechanisms that may explaining the patterns found.

I have enjoyed the manuscript and it can make an important contribution in the field. The objective are clear and ambitious, the methodology used is novel and the models well explained, and the results discussed are clear and relevant for improving the approaches for understanding global patterns of diversity of insects. One of the main implications of the study is

relevant, the factors driving species and intraspecific genetic diversity are different. Also the declining poleward pattern of intraspecific genetic diversity is unexpected, and the mechanism underlying this pattern are well discussed. The implication of the study for further global macrogenetic studies is high and this study is important for understanding macroecological patterns of GD. I have few main comments and a short list of minor comments.

Author response: We thank the reviewer for the enthusiasm in stating that our study is interesting, well written and makes an important and ambitious contribution to the field. We also thank the reviewer for all the valuable suggestions for improvement that we have implemented thereby leading to a much improved manuscript.

Main comments:

(1) In the Discussion (lines 231-235) authors argued that patterns of intraspecific genetic diversity are different from those driving the classical negative latitudinal gradients in species richness, which is the pattern expected for the readers. I am wondering if authors tested the pattern of the diversity of OTUs that were used to estimate GDM and GDE, which should reveal inconsistencies at the two organizational level (species and populations). I guess that much more effort in sequencing has being done in temperate than tropical regions and the patterns of OTUs used can reveal these differences, but I did not find this results. Here, the criteria for selecting an OTUs was at least three specimens sequenced (447-448), which sounds fine, however in temperate regions I guess that a higher number specimens are sequenced than in the tropics and therefore values of GDM can reveal differences in sequencing effort because higher number of alleles may mean higher estimates of GDM. I am not sure how authors can distinguish the effects of sequencing effort among regions on the results. Also the authors can test SGDC for each grid cell for determining the relationship between species and intraspecific genetic diversity, which is missed in the current version of the manuscript.

Author response: We thank reviewer 4 here for bringing up some critical issues. First, our best understanding is that the number of sampled OTUs has no bearing on actual species richness, but instead it is really more reflective of sequencing efforts. Second, although counterintuitive, sampling effort has little impact on GD because of the summary statistic we used. As opposed to allelic richness (which certainly does reflect sampling effort), our GD is based on coalescent theory which was used by Tajima (1983) to show that this metric approaches its expectation after only 5-10 samples. However, while this metric converges on the expectation quickly, the variance of GD across OTUs within a grid cell should be affected by sampling effort (i.e., lower variance with greater sampling effort). To more directly explore this, we looked at how the number of OTUs per grid-cell and individuals per grid-cell relate to GDM and GDE (Supp. Fig. 16). We found there to be no bias in these metrics with greater sampling effort and that variance decreases with greater sampling (also predicted). We also tested these correlations formally using Pearson's method and found no significant correlation (Supp. Table 5):

“Similarly, to investigate whether per grid cell sampling, i.e., total number of individuals, number of individuals per OTU, and number of OTUs per cell, had an effect on GDE or GDM, we tested

for Pearson's correlations between these quantities (no relationship, all $P > 0.20$, Supplementary Table 5; Supplementary Fig. 16)."

So, while lower variance in GDM and GDE does come with more sampling effort, there is no detectable sampling bias driving global patterns in these two metrics.

(2) Some sections of the Discussion sounds too speculative and based on results found in other studies, and I miss a clear discussion of the importance of current environmental variables in the study. For instance, Fig 4 shows how MTWM and precipitation seasonality are relevant for determining GDM, whereas these two variables together with PWM are critical for GDE. However, I miss a discussion of the implications of this result. In contrast, authors discussed the effects of Last Glacial Maximum, species' range size, habitat specializations, life history traits, insects diapause, but unfortunately these explanations were not related to current environmental conditions, despite its importance (lines 162-167). I suggest authors to explicitly discuss the importance of current environmental climatic variables in further versions of the study

Author response: We agree there needs to be better discussion directly about these key variables and GDM and GDE. Because the signal was stronger for GDE with respect to predictors and latitude, we now focus more of our discussion on this measure of evenness in GD, and most specifically for Lepidoptera and Diptera (the two orders that stood out independently with respect to the latitudinal gradient). A peak of both GDM and GDE around the desert areas that hug 30 degrees latitude could putatively relate to higher environmental tolerances that lead to larger and more uniform ranges which then could relate to larger and more uniform effective population sizes and GD values. Aside from latitude, this is also manifested by the model variables that came out as important drivers of the global correlations consistently across sampling schemes (MTWM, Temp Trend, and PWM). Conversely, the drop off of GDM and GDE is abrupt as sampling occurs in colder regions that are in the higher latitudes. While this is also manifested in the same aforementioned predictor variables, a putative mechanism could be:

1. founder effect expansion into areas that were de-glaciated after the Last Glacial Maximum and/or

2. recent selective sweeps on the tightly linked mtDNA genomes that could relate to adaptation to colder and/or more seasonal conditions in these higher latitudes.

However, we are clear to point out that these discussions are not direct inferences of processes but rather discussion of the results derived by our correlative models.

(3) Taxon-specific patterns of GD are weakly discussed, I suggest authors add a paragraph in the Discussion in this aspect.

Author response: We agree there needs to be more exploration and discussion of taxon-specific patterns, and we now include more interesting results related to the three largest orders of

insects that dominate the data we use in this study. Specifically, we find both Lepidoptera and Diptera data are consistent with the quadratic latitudinal gradients of GDE that we find whereas there does not look to be any meaningful patterns in Hymenoptera (if we treat them as a single group lumping together solitary and eusocial).

Minor comments:

- line 35: provide more details on “both GD metrics follow a bimodal latitudinal gradient”

Author response: *We updated our results regarding the latitudinal gradient and added much more discussion, with detail provided in other responses.*

- line 228: add “of the equator (fig. 3)”

Author response: *Agreed, although we removed this sentence in response to other revisions.*

- lines 229: change to “(LGM, <60° latitude)”

Author response: *We ended up considering the full range of latitude values, rather than filtering those above 60°.*

- line 249: change “consistent” to “opposite”

Author response: *We intentionally use “consistent” in this line to mean the lower genetic diversity is consistent with lower haplotype richness*

- line 434, add “Fig 1d”).

Author response: *Thank you, changed*

- line 903. Check the name of the journal

Author response: *Thank you, updated*

- figure 1: It is a great figure with great schemes for showing the meaning of GDM and GDE. However, I am not sure if “low GDM/ low GDE” is realistic, and I may remove this option in the figure or at least suggest how rare it should be. For clarity, in (d) I suggest increase values of GD (Y-axis) in High GD/HighGDE and I suggest reduce values of GD (Y-axis) in “low GDM/lowGDE”.

Author response: *We appreciate the compliment. In fact, we do find low GDM/lowGDE empirically in the data (in a few regions). We have modified our figure according to the reviewer’s suggestion for (d).*

- figure 2, in line 600, change to “(e, f)”. To facilitate the understanding of figure 3, I suggest add the latitude values in the figures.

Author response: Thank you for catching this. We changed (E, F) to (e, f). We also added latitude values to all maps.

- figure 3, I guess the values of latitude (X-axis) are incorrect, I guess values should rank from 50 to 75°. I suggest also to add this values on the maps in figure 2. In line 615-616: “while GDM showed a qualitative positive quadratic trend (d)...” change to “ trend (a)...”

Author response: We integrated this figure into figure 3 and made corrections- the coordinates were transformed from projected coordinates to lat/long and we updated GDM and GDE latitudinal trends with new results.

I hope these comments help to improve the clarity of the manuscript.

Author response: We thank the reviewer for their thorough and thoughtful comments that greatly improved the quality of the manuscript.

Reviewers #1 and #4 were asked to comment on the issues raised by reviewer #2 using the following questions:

1. Has the other reviewer raised technical issues related to the use of mtDNA that you feel are important to address? Do you disagree with any of their criticisms?
2. Do you think the mtDNA data analyzed are appropriate for addressing the questions posed?
3. Does the other reviewer's comments alter your stance on the conceptual advance and/or novelty of the study?

Response from Reviewer #1:

Indeed, the remarks raised by my colleague are definitely grounded and cannot be dismissed. The recent literature cited by the referee is absolutely relevant.

I would still not subscribe that mtDNA diversity, in general, bear *no* information about demographic and evolutionary processes, and, in particular, I do not agree that neutral genetic variation is useless. The authors do not claim to measure adaptive potential, they (with shortcomings that the colleague and myself have highlighted) attempt to test for historical and demographic hypotheses.

Author response: We would further say that we don't take it this far. Our study uses correlative methods, but we do discuss putative historical, evolutionary and demographic hypotheses . We treat genetic diversity as an interesting variable to study on the macroscale, likely shaped by historical processes impacting demography (as well as by selection, and neutral mutational mechanisms). But we do not view it as a population genetic marker for directly testing demographic hypotheses. This is a relatively new field (macrogenetics) and thus basic patterns are still not established. Our study is a step in this direction and suggests avenues for ways to test specific hypotheses.

In any case, we strongly agree that the limitations brought forward by the referee and the literature she/he cites should be better acknowledged in the study, along with the highlighted technical issues entailed by the use of raw genbank/bold data (e.g. the bias due to the fact that many authors only deposited unique sequences in databanks).

Author response: We have added further discussion of limitations and have now included simulations showing that omission of duplicate alleles has a minor impact on the bias (Supp. Materials; Supp. Fig. 15).

I also completely agree on the suggestion to explore Millette et al's approach of considering species identity (which I should have mentioned myself), while it looks to me that the issue of aggregating populations into "arbitrary" units mentioned by Pas-Vinas is more a concern regarding other details of Millette's approach.

Author response: We now show that the vast majority of OTUs are restricted to one grid cell, such that the Millette approach wouldn't work well. We also explain that the grouping into grid cells of arbitrary size is a useful compromise and less of a concern than stated with respect to prediction under the coalescence with migration/isolation or metapopulation. If ranges exceed the grid cell, local sampling is predicted to yield higher GD under a wide range of circumstances (Nordborg and Krone 2001; Charlesworth and Charlesworth 20210, chapter 7 for details), while absolute isolation leading to local coalescence is also not problematic.

Already in my original report I made clear that the study requires substantial upgrade before being considered for publication. The concerns raised by the colleague certainly do not improve my judgement, but I would not say that, in principle, I am against publishing any such study just because it is based on mtDNA.

Author response: A point-by-point response to the reviewer's original comments can be found above.

Response from Reviewer #4:

1. Has the other reviewer raised technical issues related to the use of mtDNA that you feel are important to address? Do you disagree with any of their criticisms?

In general, I partially agree with the other reviewer. In fact, there are other markers or genomic techniques that can be used to study population genetics as the other reviewer has listed, however the manuscript focuses on intraspecific genetic diversity across species at a global scale, which is recently called macrogenomics. In order to do it, the authors used available sequences of *cox1* that covered the world, which is probably the only fragment that is massively sequenced and can be used in a global scale study of intraspecific genetic variation.

Author response: We agree that our use of the potentially problematic mtDNA is a useful compromise that enables the scale of sampling to be done at the macrogenetic scale. Using something like microsatellites mined from the literature would be dwarfed by our study in scale and scope, and also come with its own baggage with regards to relation to neutral or adaptive processes and or rate variation/saturation.

Although the other reviewer is right in “mtDNA are not neutral, do not recombine, are not well correlated with demography, are not correlated with neutral genome-wide diversity or adaptive potential”, *cox1* provide an accurate species delimitation because the genetic divergence clusters species separately and the cladogenesis is well captured. Moreover, intraspecific genetic diversity and structure of *cox1* corresponds to biogeography and dispersion (e.g., Papadopoulou et al, ME, 2009; Baselga et al., Nat Comm, 2013; Múrria et al, ME, 2017) and this fragment is widely used in macrogenetics (Leigh et al, Nat Rev Gen, 2021). Unfortunately, data available for neutral markers such as microsat or the mt genome of *Drosophila* that the other reviewer suggested, is limited and currently it is available for a small number of species, but the data available is far for a global scale study. Moreover, I have my doubts if microsat data can be easily compared across species because the markers

used are commonly specific for each species. Until genome information is not available for thousands of species at the population level, i.e., several individuals sequenced for each population, we cannot use mt genome sequencing in macrogenetics. Also the authors are not clearly interested in adaptive genetics or general genetic diversity, as the other reviewer indicated, instead authors want to test macroecological patterns of intraspecific genetic variation for as much number of species as possible covering the widest area. For this reason, authors specifically analyzed the intraspecific DNA barcode diversity, which is clear in the manuscript.

Author response: As mentioned above, we largely agree with all of this. As the reviewer describes, we use the only data source with the geographic and taxonomic scale available to answer this question. While not perfect, is the standard approach in macrogenetics thus far in other taxa and provides a baseline for future work.

On another hand, I agree with the other reviewer that one main problem is the number of available sequences that limits the analyses and conclusions that authors can perform. As the other reviewer suggested, I also agree the authors should introduce the limitations of the study:

- the "poor" availability of sequences at the global scale increases the grid cells at a 193 km x 193 km equal-area resolution, which is apparently a large size for a population genetic study. Also there are "only" 187 regions accumulated mainly in Europe, North-America and East-Australia, which should introduce some bias to the results. These limitations should influence the results found.

Author response: We now explore a range of sampling densities (minimum threshold of 25, 50, 100, 150, and 200 OTUs per grid cell), and find the overall results to be largely consistent. Fortunately, the threshold leading to the best model fit (minimum of 100) also led us to use a larger number of grid cells (245). We masked out regions outside the model we used and only made predictions in the remaining areas. We also agree and acknowledge the obvious sampling bias in North America and Europe. These changes were made in response to the helpful comments made by reviewer #1.

- I suggest authors increase the explanations of the limitations of using DNA barcoding data in population genetics studies that the authors briefly introduced in lines 79-82. As the other reviewer have pointed, I suggest include some of the limitations pointed by Paz-Vinas et al 2021 Ecol Let; Galtier et al 2009 ME.

Author response: We agree and have now expanded discussion of limitations in the Introduction and Discussion. The overall point of view that we have is that while intraspecific mtDNA diversity data is insufficient for making detailed inference of demographic history or phylogenetic reconstruction (Galtier et al. 2009), we instead treat it as an important variable to study on the macroscale, rather than a population genetic marker.

2. Do you think the mtDNA data analyzed are appropriate for addressing the questions posed?

Despite the mtDNA data analyzed has limitations, I think the data used can address the questions posted and the study is well-performed. However, assuming the limitations of cox1 data, I suggest authors rephrase the hypothesis based on the comments by the other reviewer. In fact, some of the hypothesis based on demographic processes cannot be directly tested using the available data. Improving this section is critical. As the other reviewer, I agree in "I suspect that there are many interesting processes related to the biology of mtDNA underlying some of the patterns detected." I may focus the new hypothesis on these processes.

Author response: As previously mentioned, we don't actually test these hypotheses via inferences of underlying processes, but instead we conduct a correlative study using a phenomenological model and discuss possible interpretation and processes that could be more directly tested via process-explicit models in future studies. In other words, our study does not really test hypotheses, but rather it generates them (Pilowsky et al. 2022 for a discussion of the distinction between these two types of modeling approaches).

Another interesting criticism by the other reviewer is the metric used: mean GD in each grid. Authors must argue extensively why this metric is appropriate, and I think it is. Moreover, authors can explain if they used the abundance of each haplotype in a grid or only the data of each unique sequence, which can introduce bias as the other reviewer suggested, especially when authors used the mean GD. This is also critical.

Author response: In the manuscript we have explained that we use a classic metric (average pairwise differences) that is robust to small sample sizes (unlike allelic richness) and now provide simulation experiments showing that this metric is also not strongly biased by the removal of duplicate alleles (Supp. Materials, Supp. Fig. 15) We have also addressed these two issues directly with other reviewers.

3. Does the other reviewer's comments alter your stance on the conceptual advance and/or novelty of the study?

No, I still think of the high contribution of the manuscript. However, the authors should clearly explain the limitations of the data used.

Author response: We thank the reviewer for noting the contribution we are making and now have expanded text explaining the limitations of mtDNA and the sampling compromises our and other macrogenetic studies make:

Introduction:

“Most macrogenetic studies of animal taxa are based on mitochondrial DNA (mtDNA) sequence data, the most abundant type sequence data in public repositories 15. These single-locus markers have drawbacks: notably, the genetic diversity that mtDNA captures is from a single draw from the stochastic coalescent process operating within a lineage and may be subject to selection in addition to neutral demographic processes 62–67. However, the ability to sample the genetic diversity of thousands of taxa per locale outweighs these theoretical considerations 68,69.”

Discussion:

“Moreover, both metrics are calculated using mtDNA, which is a single, non-recombining locus that is likely shaped by demographic history and selection 67,101. While intraspecific mtDNA diversity data is insufficient for making detailed inference of demographic history or phylogenetic

reconstruction 67, we instead treat it as an important variable to study on the macroscale, rather than a population genetic marker. Macrogenetics is a relatively new field and thus basic patterns are still not established. Our study is a step in this direction and suggests avenues for ways to test specific hypotheses.”

Reviewer 1 commented manuscript response

Author response: *We thank reviewer 1 for their additional comments on the manuscript. The thoroughness of their review greatly improved the quality of the manuscript. We reproduced the comments below and acknowledge our use of brief responses for small, grammatical changes.*

Abstract

L29 maybe "to diversity of life"? Otherwise, one may misunderstand it for the origin of life from non-living matter..

Author response: *Agreed, changed*

Introduction

L45 Shouldn't it be "barcoding" in general, including databases that were built without high-throughput sequencing techniques?

Author response: *Agreed, changed*

L72 "is largely lacking for"?

Author response: *Agreed, changed*

L78 It is not entirely clear to me how this paragraph connects with the flow of the introduction. Is the present study addressing the issue of species identification?

Author response: *Thank you for pointing this out. The paragraph is meant to highlight the difficulty in quantifying insect diversity, and the promise of DNA barcoding for closing the sampling gap. We think the specific mention of "...global quantification of insect species diversity..." focused the paragraph too narrowly on species diversity, when we want to highlight that other aspects of diversity can be quantified from barcoding data. We removed "species" from the sentence to make the paragraph fit in with the rest of the introduction.*

L94 Indeed. However, this should be less of an issue for studies where the identity of species has been accounted for (see Millette et al., ref. 25). Isn't it?

Author response: *We think we were misunderstood on this point and have now clarified the text. We are merely stating that the average across species GDs has little information on the variation in GDs across species.*

L99 Remove "Explanations for"?

Author response: *Agreed, removed.*

L101 I feel like this paragraph may be rephrased to be more explicit.

As far as I get it, the "museum" hypothesis predicts higher GD in tropics and the "cradle" hypothesis predicts lower GD in tropics *only if* there is a positive relationship between range size and *local* GD. In other words, *overall* GD may be higher in large-range species just because there are many and far-away individuals, but, taken two gene copies *within* a given distance, they may have the same probability to be different alleles than two gene copies taken from the same distance in a small-ranged species. In this case, we would not expect any latitudinal gradient based on the size of the ranges..

Author response: These are interesting points but we are basing our general predictions based on the power law derived in Exposito-Alonso et al. 2022 as well as population genetic theory recently reviewed by Charlesworth and Jenson 2022 (and Nordborg and Krone 2002). Pairs of individuals sampled at the same distance in a small range and large range species are not predicted to have the same probability of being different. Unless there is complete isolation, there is some probability that one of the coalescent lineages in the larger ranged species "escapes" into unsampled areas (before a coalescent event, going back in time) thereby prolonging the time to final coalescence between the pair of lineages (and increasing the GD on average). These details are also reviewed in Chapter 7 of Charlesworth and Charlesworth 2010.

Charlesworth, B., and D. Charlesworth. 2010. "Elements of Evolutionary Genetics."

<https://www.research.ed.ac.uk/en/publications/elements-of-evolutionary-genetics>.

Charlesworth, Brian, and Jeffrey D. Jensen. 2022. "How Can We Resolve Lewontin's Paradox?" *Genome Biology and Evolution* 14 (7). <https://doi.org/10.1093/gbe/evac096>.

Nordborg, M., and S. M. Krone. 2002. "Separation of Time Scales and Convergence to the Coalescent in Structured Populations." In , edited by M. Slatkin. and M. Veuille, 130–64. University Press.

L110 I would like to see explained how this is actually different from the "museum" hypothesis. I see that the authors described the latter as being connected to range size, rather than stability per se. But what, if not stability, causes tropical species to be older? Is it lower speciation rate? What is there in the tropics to lower the speciation rate?

Author response: We are hypothesizing that both larger ranges and climatic stability could have a positive effect on GD yet they are different processes and do not necessarily co-occur. For example, a climatically stable area might also coincide with small and variable range sizes, and conversely an area that experiences cycles of glaciation may also coincide with many large ranged species. But in both cases, GD is predicted to be deflated and it is only in the "sweet spot" of areas with many large ranged species with relatively stable climates does one predict elevated GD consistently across species (to yield higher values of GDE).

L 113 It is not explained how all of the above predictions are expected to affect GDM and GDE specifically. Are the same predictions valid for both? Why?

Author response: Our use of both metrics is based on the idea that using both extracts more information about aggregate GDs from many OTUs instead of using GDM alone. It does not so much stem from specific predictions for each except that both being high could be predicted in areas of stability and the evolution of larger ranges in a large number of codistributed species. We have elaborated in the introduction to give a better justification:

“Existing macrogenetic studies, which are inherently correlative, have only described average intraspecific genetic diversity (GDM), and are therefore unable to determine whether high average genetic diversities are due to high diversity within most community members or to the effects of a few taxa with extremely high diversity (Fig. 1; see Methods). The two metrics present complimentary information when considered together, with demonstrated utility in discriminating among possible processes generating natural community assemblages 73.”

“...we consider the influence of human disturbance on patterns of assemblage-wide GD, which we expect to decrease the magnitude and evenness of GD in areas of high human influence due to general reductions in abundance with the potential proliferation of disturbance-adapted species 48.”

L114 It may be useful to explain why and how human disturbance is expected to affect GD (GDE or GDM?). Probably it is just a control, in this context, but it would not hurt to explain it here.

Author response: We agree that further explanation is warranted. We added text to the previous sentence, so the sentence reads as:

“Finally, we consider the influence of human disturbance on patterns of assemblage-wide GD, which we expect to decrease in magnitude and evenness in areas of high human influence due to general reductions in abundance with disturbance-adapted species potentially proliferating”. Also, there have been a number of recently published macrogenetic studies looking at aggregated GDs and human impact.

Results

L131 Having a look at the data and then removing the portion that does not fit a model does not look like the very best statistical practice, to me. Is it really sensible to speak about the "statistical significance" of a parameter after one has modified the data? Maybe, a parameter-free approach like a GAM would have been a better way to explore these data? (of course, in principle, one should note go for a new approach just because the first one did not reveal significance, but..)

Author response: *Thank you and we agree that removing cells from specific latitudes was not the best statistical practice. We revised and simplified our assessment of the relationship between GD and latitude, and we expanded upon specifics in the main reviewer response text.*

L134 Are these estimated and HDI from the model accounting for spatial autocorrelation (SAC)? (I don't think that models not accounting for SAC are actually useful, in this context)

Author response: *These were from models accounting for SAC, but we revised our approach and are more clear about the modeling choices.*

L135 What are the numbers on the x axis of Fig. 3? I think that reporting the latitude would greatly help.

Author response: *The numbers were projected coordinates, but we agree that this was unclear and we transformed them back to the WGS84 projection for interpretability. They are now inset plots in Fig. 2.*

L137 "in that" rather than "where"?

Author response: *Agreed, although the text was modified in response to other comments.*

L137 I would say this is almost centered on zero... it does not exhibit *any* trend.

Author response: *Agreed, although the text was modified in response to other comments and we are more clear about the lack of a trend.*

L145 Why a t-test? This test assumes independent observations. This is obviously not the case here, because of spatial autocorrelation (in principle, there could also be phylogenetic non-independence, but I may hope this is not a huge issue and it has been very rarely accounted for in this kind of studies...)

Author response: *We agree that we overlooked this component and opted for a spatially modified t-test that accounts for spatial autocorrelation. Thank you for bringing this to our attention.*

L148 Wasn't it the same method for latitude?

Author response: *We simplified our approach for assessing the relationship between latitude and GD and outline our approach in the main response text.*

L152 The phrasing is not very clear. Is it in the raw data or in the prediction? Why not just say that they were found to correlate positively with MTWM?

Author response: *Thanks for catching this. We updated the language to be more precise:*

“We found that GDM and GDE covary significantly with current and historical climate, and that both are positively correlated with maximum temperature of the warmest month (MTWM) and climate stability variables”

L154 Is it really meaningful to *first* select variables and then partition them again to further test submodels? The rationale for this analysis deserves better justification..

Author response: We agree with your assessment and opted to revert to a single round of variable selection, which we detail further in the main response text.

L159: Does it mean that all results listed above were obtained without accounting for SAC? This is not clear to me, even after reading the Methods section, If so, are they reliable/meaningful?

Author response: Yes, we reduced the variable space prior to accounting for SAC out of necessity. While this may bias variable selection towards those variables that have matching SAC patterns with the GD metric, we include regularizing priors for the GLMM that accounts for SAC so this bias is minimized in interpretation of the final model. In our updated analysis, we refrain from splitting model selection into two components to make this modeling choice clearer.

L163 As far as I understand, there was no test for a null hypothesis (that a model with the intercept and spatial autocorrelation term only might have performed better than any other).

Author response: By using regularizing priors, we expect the null hypothesis to result in response variable coefficients that overlap zero, which we do not find.

L167 I noticed that, in several cases, SE for ELPD are rather high compared to the differences between ELPD values themselves. Does it mean that the difference between the models is not "significant"? How is the SE on "ELPD difference" computed? It could be explained (at least in Table 2) for those readers that (like me) are not familiar with this approach.

Author response: Yes, thanks for bringing up this point of confusion. The standard errors of the two metrics are actually calculated in different manners, where the SE_{ELPD} is sensitive to sampling variation (e.g. low N or correlation among posterior samples), while $SE_{ELPD-DIF}$ is not as sensitive. We highlight this difference and our decision making in an updated table caption, reproduced here:

“The ELPD is the expected log (pointwise) predictive density, where a higher ELPD indicates more support for the model. The predictor column is additive, where the model with the predictor listed contains all of the predictors listed before. In addition to ELPD, we report the difference in ELPD (ELPD-DIF), which is the difference in ELPD from the reference model. The $SE_{ELPD-DIF}$ is the standard error of the difference in ELPD from the reference model. SE_{ELPD} is a coarse summary of model support variation that is sensitive to sampling quality, and we take the advice

of Vehtari et al. (2017) to use $ELPD-DIF$ and $SE_{ELPD-DIF}$ when comparing models. We chose the simplest model whose $ELPD$ overlapped with the reference model's $ELPD$ ($ELPD-DIF + SE_{ELPD-DIF}$ overlaps zero) and corroborated our selection with the heuristic method of selection employed by the *projpred* R package function `suggest_size()`. The chosen model is indicated with gray shading.”

L168 R^2 values are pretty high. It would be nice to see scatterplots like those in Fig. 2 for these effects. Not being familiar with the *glmmfields* package, I wonder whether these R^2 reflect the variance explained by the fixed effects only or they include the contribution of the SAC term. It could be clarified for the reader.

Author response: We agree that further clarification is warranted. We modified the original Bayesian R^2 method so it properly accounts for the variance explained by the SAC term. Bayesian R^2 looks like this: $var_{fit} / (var_{fit} + var_{error})$, where var_{error} = variance in the residuals (sigma parameter for the fixed effects) (Gelman et al. 2017, <https://doi.org/10.1080/00031305.2018.1549100>). We expanded var_{error} to include both variance in the residuals and variance in the gaussian process error used to fit the spatial autocorrelation covariance matrix, so $var_{error} = \sigma_{fixed\ effects} + \sigma_{GP}$. We added a line in the methods to acknowledge that we accounted for spatial error:

“The proportion of variance explained by the models were assessed with Bayesian R^2 (Gelman et al. 2019), modified to account for spatial autocorrelation error.”

In addition, we added R^2 density plots as insets to our posterior panels (a, c) on Figure 4.

L169 "the most support" sounds wrong to me. "Highest support"? (see also L173).

Author response: Agreed. We changed accordingly.

L169 Is it because of SE on $ELPD$? Why may H2 be confidently discarded, instead? Again, since this is not the most common approach to hypothesis testing, it could definitely be explained. Also. Reproducing the hypothesis labels (H1, H2, H3) in table 2 would help (if these are to be maintained in the final version of the study, which I am not sure is the best way to go [see general comment]).

Author response: We agree with your comment in the general reviews and omitted this stage of model selection/hypothesis testing.

L190 This is not really visible in the map

Author response: We have updated our maps and resulting interpretations of our maps.

L214 I do not understand. Does it mean that the relationships between predictors and GD were tested with a reduced data set containing only 2.8% of the OTUs? If this has not been done,

how relevant is the test described here? Did the authors consider using phylogenetic regression to account for differences in baseline GD across taxa? Wouldn't that be the best way to do it?

Author response: *We agree that this wording was not precise enough. We changed the text to:*

“Using Welch’s unequal variance t-tests, we found no significant difference in GDE estimates between the full dataset and the dataset with the most prevalent outliers removed“

We did consider phylogenetic regression, but there is no available insect-wide phylogeny to use, and inferring a phylogeny for these purposes would not be ideal, given the single locus mtDNA dataset. The coarseness and imbalance of the per-order sampling precluded the use of an ANOVA or similar non-parametric method that takes into account the grouping, but not the precise evolutionary relationships, so we opted for a simple assessment of the impact of removing highly sampled taxa or leaving them in with Welch’s t-tests.

Discussion

L226 It was not significant, if I remember well!

Author response: *Agreed, and we updated our discussion to reflect new results.*

L228 This is not clear. "areas that were unglaciated" would include the tropics.

Author response: *This sentence was removed in light of new results.*

L230 Where does this distinction between "wet" and "dry" (?) tropics come from in the analysis?

Author response: *This sentence was removed in light of new results.*

L244 The whole discussion at L231-244 is based on non significant latitudinal trends in GDM, significant latitudinal trends in GDE for which no prediction has been proposed yet, or indirect geographical projections of predicted GDM based on the observed relation with climate variables. To me, it is not useful to spend so much time to explain such a weak signal. By the way, if the relationship between species richness and GD (or lack thereof) was to be discussed, why not analyzing species richness as a predictor of GD?

Author response: *We agree that dedicating too much discussion to GDM in this context was not warranted. We opted for a qualitative comparison to existing biodiversity patterns in general, rather than attempting to explain the mechanistic underpinnings/relationships among the biodiversity patterns listed. We add text to qualify our discussion, which is repeated here:*

“Although our approach is correlative and not as suitable for direct inference of processes as simulation models 89, these results suggest that forces underlying intraspecific genetic diversity are inherently different from those driving the classical negative latitudinal gradients in species

richness and phylogenetic diversity found in most taxa, including ants, butterflies, and spiders 90,91 92–94, as well as plants 95, which are expected to be strongly linked to insect biogeographic patterns. Bees (order Hymenoptera) are a notable exception with a latitudinal gradient similar to GDE with highest richness at mid-latitudes 96. However...

L245 None of these studies mentioned a trend in GDE, as far as I know, and not significant trend in GDM was observed here!

Author response: This is true, and we now specify that we are comparing the GDE trend to average GD in the other big studies. The text now reads as:

“The latitudinal gradient of GDE peaking in the subtropics also contrasts with recent macrogenetic studies of vertebrates, all of which find a negative latitudinal gradient of average genetic diversity peaking in the tropics and declining poleward, including mammals²⁰, amphibians²², and fishes²¹.”

L260 Yes. If we assume that the genetic diversity within each species is mixed up across its whole range, which is rarely the case. A wealth of Phylogeographic studies show that large swaths of each species range can be dominated by a few lineages, as a result of recent range expansions. The plausibility of this explanation should be discussed. However, as I said earlier, I see no reason to discuss this, as there is no evidence for lower GD in the tropics compared to mid latitudes.

Author response: We appreciate this phylogeographic pattern, but we are coming from the perspective of coalescent theory with migration (i.e., the structured coalescent; Nordborg and Krone 2002 and Nordborg 1997), which we outline in our response to a previous comment of yours from the introduction (L101). We also discuss range expansions being a likely explanation for lower GDE in areas that were recently deglaciated and recolonized.

Additionally, we have also added discussion on the subtlety of the relationship between GDE, GD, and range size, acknowledging the correlation between GDE and GDM and providing reasons why they may differ:

“Assuming that Rapoport’s Rule is a general tendency in the two large insect orders, Diptera and Lepidoptera, sampled coalescent times among sampled individuals within larger ranged subtropical species are expected to be older and yield consistently larger values of GD that aggregate to larger GDE within grid cells 107, especially if species ranges are larger than the 193x193 km grid-cells we chose and there is subdivision within the range of a species 108,109. The inflation of GD from large-ranged species in the assemblage results in more even and generally higher genetic diversities and is reflected in the positive correlation between GDE and GDM (Fig. 2d,e). However, we do not find a relationship between GDM and latitude. We are hypothesizing that both larger ranges and climatic stability could have a positive effect on GD, yet they are different processes and do not necessarily co-occur. For example, a climatically stable area might also coincide with small and variable range sizes, and conversely

an area that experiences cycles of glaciation may also coincide with many large ranged species. But in both cases, GD is predicted to be deflated and it is only in the “sweet spot” of areas with many large ranged species occupying relatively stable climates does one predict elevated GD consistently across species, to yield higher values of both GDE and GDM. "

L271 I do not think that the resolution is coarse enough to avoid this. Phylogeographic studies show that large swaths of each species range can be dominated by a few lineages, as a result of recent range expansions. This counterargument to the range-size explanation should be discussed immediately after that. However, as I said earlier, I wonder if it is worth to discuss the whole latitudinal trend, as there is no evidence for lower GD in the tropics compared to mid latitudes.

Author response: *We removed this sentence in favor of more nuanced discussion addressed in other comments.*

L277 What does "uniform" mean here? Less structured? Why?

Author response: *Thanks for bringing up this lack of clarity. We made ourselves more clear by changing the language to:*

“Insect diapause is thought to provide adaptive tolerance to wider abiotic conditions and may result in larger and more uniform range sizes across an assemblage”

L279 Here, "uniform" means "less different among species". It seems like the authors are suggesting a logical connection with the "uniform" at L 277, but I don't see any. Am I wrong?

Author response: *We make the connection more explicit and clear:*

“The positive relationship between range size and GD 65,89 provides a possible mechanistic relationship that connects Rapoport’s Rule and the more uniformly high genetic diversities found in higher latitude regions with pronounced seasonality that do not freeze”

We think this correction in combination with our correction in response to your previous response makes the meaning of “uniform” more consistent across applications.

L279 Precipitation seasonality was found to be *negatively* correlated with GDM. How can it be that regions with pronounced seasonality have higher GD?!

Author response: *Thank you, we agree that our discussion became confusing and misleading. We have omitted this erroneous connect and have stuck with only the broad patterns and putative processes.*

L287 I completely failed to follow the logic of L280-287.

Author response: As we say above, we agree that our discussion became confusing and misleading. We have omitted this erroneous connection and have stuck with only the broad patterns and putative processes.

L309 This explanation makes sense. However, it would be worth mentioning that the poleward decline in GD was only assessed "by eye" after plotting the data, and does not result from any statistical analysis. Moreover, isn't it in complete contrast with the dismissal of any effect of past climate?!

Author response: Thanks for pointing this contradiction out. After our updated analysis, we do find an effect of past climate, which makes this observation consistent with our results. Statistically speaking, this was best captured by GDE whereby lower values are driven by large numbers of OTUs with zero GD within a grid-cell. We have now described these dynamics.

L311 So far, the authors have discussed about a *negative* relation between species richness and GD. This sentence seem to imply the exact opposite!!

*Author response: We removed the suggestion of a possible negative SGDC to emphasize that we were discussing the *lack* of a broad relationship between species richness and GD. We also removed this sentence in light of the dominance of other insect orders in our sampling (i.e., the hotspots we were discussing were really just minor hotspots at a different taxonomic scale than "all insects").*

L315 As far as I understood, odonata are a tiny fraction of the data. Is this relevant?

Author response: Agreed, we removed this sentence.

L322 This is to say that high-latitude communities are assembled via environmental filtering more than low-latitudes? As a reader, I would like it to be explicit.

Author response: Thank you for pointing this out. We added our expectations more explicitly, copied below:

"Areas with higher levels of GDE could also partially emerge from different levels along the continuum of fundamental community assembly processes 120. Although the spatial scale is not always in line with the grid-cells we employ, Overcast et al. 75 found in simulated and empirical (arthropod, annelid, and trees) communities to have elevated GDE under ecologically neutral conditions in contrast to non-neutral, i.e., niche-structured conditions. In these specific model-dependent cases, the lower GDE in communities assembled via environmental filtering is likely caused by increased genetic diversity in hyper-dominant species with stronger local ecological adaptation. In line with the process-explicit modeling, our observation of GDE increasing from the tropics to the subtropics shows that equatorial insect communities may have stronger local niche-structured mechanisms (i.e., less ecologically neutral conditions), than subtropical

temperate insect communities. This would be consistent with the idea of stronger niche conservatism in the tropics. 121”

L326 I suspect that this may be made more explicit, but I sort of I follow this. Overcast et al. found that "neutral" communities should have higher GDE, and this is consistent with temperate communities being less structured than tropical ones. However, about the high latitudes, the idea is that, there, environmental filtering prevails in that a few species thrive better than others... Maybe it can be clarified whether environmental filtering and neutral assembly processes are completely different and independent phenomena?

Author response: *Yes, you hit the nail on the head. We added clarifying text to directly relate our expectations regarding neutrality (clarifying text bolded for emphasis):*

*“ In line with the process-explicit modeling, our observation of GDE increasing from the tropics to the subtropics shows that equatorial insect communities may have stronger local niche-structured mechanisms **i.e., less ecologically neutral conditions**, than subtropical temperate insect communities. This would be consistent with the idea of stronger niche conservatism in the tropics. 121”*

L330 What if they are just not sampled? What if they are sampled more just because they are threatened?

Author response: *We agree that this opens up a can of worms that should remain closed. We removed it from discussion*

L359 How reliable the extrapolation is? Can it be quantified by cross-validation?

Author response: *While we didn't do a full cross-validation approach, we did revise our methods to include splitting the data into 75% training and 25% testing so we could test how reliable the extrapolation is. In addition, we added multiple methods of assessing extrapolation error and masked areas with environmental space that fell outside the range of environmental conditions we used to train our models. This is outlined in our response to a general comment.*

Methods

L387 "for which"?

Author response: *After review, replacing “but which” with “and” better conveys our meaning.*

L416 Being essentially a proportion, GDM is not expected to show a normal distribution, and it has often been modelled using a beta regression (see, e.g., Miraldo et al. 2016). Why not following this path here? Could it be better justified?

Author response: *A beta regression would be reasonable, but we opted to use the square-root transform approach used by Theodoridis et al. (<https://doi.org/10.1038/s41467-020-16449-5>) for*

comparison's sake and now cite them. The transform resulted in a reasonable distribution and was easy to interpret, so we went with it. Here is the modified text:

“Because the distribution of GDM at the grid cell scale was highly skewed towards zero, we performed a square-root transformation to achieve a more normal distribution, consistent with Theodoridis et al. (2020)’s approach.”

L442 It looks a lot to me! How does it compare to similar studies? Where all other studies that employed fewer species per cell so much wrong? I think this needs to be explained!

Author response: We agree that a wider range of minimum OTU thresholds needed to be explored. We explored 10, 25, 50, 100, 150, and 200 as possible minimum OTU thresholds and found the 100 was the most appropriate threshold for our data, although predictions were highly correlated with each other from the other thresholds. We outline our selection choice further in a response to the general review comment made by the reviewer.

L448 I do not understand why this is "conservative". If BOLD data submissions omit duplicate alleles, any choice would overestimate diversity. The impact of this feature of the data should be acknowledged and, possibly, estimated. I reckon that it may be very heavy, in fact, and even affect geographic pattern. In fact, it will almost certainly “flatten out” any geographic pattern (if duplicate haplotypes are systematically omitted, the overestimation of diversity will be stronger in less diverse areas).

If 5-10 samples provide almost the same information as hundreds, choosing 3 as a threshold may be considered sensible, but I don't see how it is “conservative”.

Author response: Thank you, we agree that this was a poor word choice. We replaced “conservative” with “sensible”, which is more in line with what we were trying to convey. As explained elsewhere, we also chose average pairwise differences because of its robustness to sample size as well as robustness to omission of duplicate alleles. To confirm this, we now provide the results of coalescent simulation experiments, and matching the reviewer’s prediction, the bias does decrease in more diverse areas (higher effective population size).

L525 Is it really *that* weak? Can it be explained?

Author response: We realize that “weakly informative” was not the correct terminology. “Regularizing” was more appropriate because we set the priors centered around zero, which will force predictors with weak signal to have a slope at or near zero. We made the appropriate change.

L541 If I get it right, this means that the selection process could have resulted in a different "candidate" model if it had been directly started with SAC accounted for. Wouldn't this be a very good reason to directly start with SAC accounted for?

Author response: Although this part of the model selection process was omitted, we still chose to perform an initial round of variable selection that does not account for SAC. Exploring a large candidate set of models was unfortunately not feasible with our spatial GLMM modeling approach. We explored other modeling approaches that consider spatial autocorrelation from the beginning (machine learning with spatial cross-validation and spatial/conditional autoregressive models with multiple model selection approaches), but they were not appropriate for our data set and modeling goals. With the machine learning approach, spatial autocorrelation was still present in model residuals in the highest performing models, and variable selection was ambiguous with difficult to interpret results. The spatial/conditional autoregressive models were unable to properly account for spatial autocorrelation, and are not reliable for extrapolating to areas outside of the neighborhood used to train the model.

L543 Typically, hypotheses are set in the Introduction. I suggest to do so.

Author response: We agree and decided to refrain from formalizing hypotheses in the manner described here and in the results.

L550 It means that SAC was accounted for? It did not look like that in the Results. I would make it explicit since the start.

Author response: We ended up using a simpler approach to assess the GD~latitude relationships that accounts for spatial autocorrelation as well (spatially modified t-test). The relevant methods text is copied below:

“We tested for the statistical significance of a linear or quadratic relationship between latitude and GDM and GDE while accounting for spatial autocorrelation using a modified t-test of spatial association, implemented in the R package SpatialPack v0.3 165,166. This was done for the full data set and independently for the three most sampled orders. We also independently tested the effect of whether an area freezes or not on the two GD metrics using the same modified t-test of spatial association.”

L551 There seems to be something wrong in the way this sentence is constructed. Isn't it that the models had a linear and a quadratic term for latitude as a predictor of GDM or GDE?

Author response: This is fixed in the new text copied in the previous response

L552 It is not clear whether an intercept model only was also considered when evaluating the hypotheses for current and past climate.

Author response: We considered the possibility of an intercept-only model by using regularizing priors on the fixed-effects.

L554 Why independently? Isn't it just another climate variable? I think it might be explained.

Author response: *We chose to use the same statistical approach that we used to assess the GD~latitude relationships for its simplicity. Although we use the language of the “freezeline”, it is inherently a spatial binary variable that would require adding a level of hierarchy to our already complex environmental GLMM, which leads to poor model convergence.*

L555 Why not GLMMs here?

Author response: *See above response*

REVIEWER COMMENTS

Reviewer #1 (Remarks to the Author):

My impression is that the current version represents a big improvement from the original manuscript, especially in terms of the statistical analyses. I am glad to see that my suggestions have been taken into account and, at occasions, appreciated.

Nonetheless, I would still like to suggest some streamlining to the description of the structure of the study, which, I find, is still not well defined in the introduction.

The final paragraphs of the introduction, as it is customary, present the main hypotheses and predictions to be tested in the study.

One reads about "museum" vs. "cradle" hypothesis, with the first predicting GD(M?) to be higher in tropics *because* of older species (associated with larger ranges) and the second predicting GD(M?) to be lower in tropics (associated with smaller ranges).

Then, the authors mention the effect of climate stability, which is predicted to be positively associated with GD(M?). The authors do not state this explicitly, but, as far as I am concerned, this is another (more process-based) way of stating the "museum" hypothesis.

The last idea mentioned in this section is that human activities may have reduced GDM *and* GDE " due to general reductions in abundance with the potential proliferation of disturbance-adapted species".

After this, one would expect that the authors tested for the effect of species range, climate stability and human disturbance, which is not what one reads next.

I would *really* like to see this section modified so that the reader gets a clear set of hypothesis associated with a set of predictions, each of them stated in terms of statistical effects that can actually be tested in the subsequent analyses (which should include *the* relevant predictors [or their proxies] and, if the case, some important controls, and nothing else, all duly explained and justified). It would be a huge improvement and it would have the further benefit of reducing the verbosity of the discussion.

By the way, although I did not mention this in my comments the original version of the manuscript, I now believe that the hypothesis that the geographic distribution of intraspecific genetic diversity may be (also) influenced by the effect of *temperature* on mutation rates should be also discussed, as it seemed to be suggested as an important factor by results by, at least, De Kort et al. 2021 (<https://www.nature.com/articles/s41467-021-20958-2>) and Manel et al., 2020 (<https://www.nature.com/articles/s41467-020-14409-7>).

Also, as far as I understand, the complete list of the initial 49 variables, as well as their correlations and the rationale for the inclusion of the final set of 11 variables rather than another set of uncorrelated predictors, is not provided. I think it would be a necessary addition.

I am not particularly convinced by the idea of having the "freeze" variable tested separately from the other predictors. If it is a matter of it being strongly collinear with some others, it should be clarified. In any case, as I see it, it is preferable to pick up a set of predictors based on clearly-defined hypotheses and predictions and test them all at once.

I would comment in the main Results section about the efficiency of retaining cells based on different thresholds for the number of OTUs in terms of R^2 and RMSE (wouldn't the latter be a more useful index to report than the slope or intercept?).

I am not 100% convinced of the idea of testing the importance of the variables without accounting for SAC and then run the final model with SAC, but I understand it may have been forced by technical limitations, and it seems justified. If this is the case, however, I think it should be disclosed in the main text.

A small detail: L101. Replace "complimentary" with "complementary"

Reviewer #2 (Remarks to the Author):

This is my second review of this paper. I remain excited about its potential but was disappointed by the responses to my comments. In my last review, I had hoped to convey my support for this work so long as the appropriateness of the data and analyses for the questions addressed were appropriately, directly, and clearly caveated. In my view the authors have in many important instances, dug in on critical issues while weakly defending their position. Many of the noted shortcomings are acknowledged in the response to reviewers. Some critical papers are now cited but criticisms are minimally discussed if they are discussed at all. Given the generality of the claims made and the weakness of the data and analyses any reader who does not read the paper very carefully, or who does not have direct expertise in the area will be led astray. If this paper was published in its current form it would contribute to growing confusion in the field together with other similar papers misusing and misinterpreting similar data (e.g., Manel et al Nat Comms; Theodoridis et al Nat Comms; Miraldo et al Science).

I will provide a brief summary of my thoughts here as the responses below got long and are very repetitive. First please imagine a study of another functional gene. One might be interested in EPAS1 which helps people use lower amounts of oxygen efficiently (mtDNA too is important for metabolism). If we could easily access global sequence data we could come up with many interesting hypotheses about how it might vary through space in relation to environments. This would be a very interesting study of an interesting gene and how it varies within a species.

Now imagine that the study took the average diversity and evenness of this gene for hundreds of species that have had millions of years of divergent evolutionary trajectories. This study would be very comparable to the study under review. I now ask the editors, reviewers, and authors to reread the opening two paragraphs and think of the above-described study. I suspect that if you received such a paper you would think that diversity for this single adaptive gene with a very specific function would seemingly obviously not be useful for almost any of the broad biodiversity, conservation, and adaptive potential claims made. Not useful for hypothesis generating or testing. These opening paragraphs make direct reference to the utility of a single adaptive gene (barcoding) in all these instances. Claims by the authors that they are not directly testing or are generating coarse hypotheses seem misleading given the tone of this opening and indeed much of the paper.

Apologies for my lack of editing. This process has run long. I hope the points are clear despite any errors.

Brief review of current draft paper with many more details in my response to responses below.

Line 39-56: We have established and agreed (I think) with references that strong selection, absence of recombination, erratic mutation rates that vary 100 fold across that mtDNA diversity is highly unpredictable and it is not correlated with genome-wide nuclear diversity (Allio et al., 2017; Galtier et al., 2009; Nabholz et al., 2009; Romiguier et al., 2014). I thus think statements about how mtDNA can inform about biodiversity at any level let alone “adaptive potential” and “ecosystem resilience” are overstated. Given that the measure in the current study are averages of hundreds of species with very disparate levels of diversity this study can in no way inform about intraspecific diversity (textbook defined as genetic diversity within a single species but used differently in the present study). I suggest these two paragraphs be recrafted.

Line 47: These references refer to genome wide variation which mtDNA is not a measure of. They cannot be interpreted as supporting the use of barcoding diversity in the context used.

Line 52: Conservation genetics is interested in adaptive potential. mtDNA is not correlated with genome-wide nDNA diversity or adaptive potential (Galtier et al 2009 ME).

Line 53: I’m not sure of the relevance of statements about neutral genetic variation are to this study because mtDNA has highly functional critical genes and undergoes strong selection. I suggest deleting this line

Line 81: It is good to note this drawback. I suggest the authors note the many issues with averaging measures of single within species draws of the coalescent process across highly diverged species

Line 83: This is an agreement that lots of data outweighs the significant drawbacks of this marker and the analytical approach. I obviously disagree and can only see the approach aggravating issues. Nevertheless the papers cited here do not support the statement. They refer to eDNA studies, utility in species delimitation, and early steps at calculating abundance from the number of sequences in samples. They say nothing about estimating and genetic diversity metric which indeed is a very non-standard use of barcodes.

Line 157: Dichotomizing continuous variable is generally bad practice

Line 165: Including highly correlated variables as the authors have done is fine for predictive modelling, which the mapping exercise is. But it removes the biological interpretability of individual effects due to violating the assumption of independence among variables in regressions.

Line 147: It is incorrect and misleading to call this “intraspecific genetic diversity” here and throughout. The textbook definition of intraspecific variation is variation among individuals of the same species.

Line 263: should state something like “which highly functional and is under strong selection in most species”

Line 275: I cannot see what average evenness and richness can tell us about single species ranges. No range information is included in the study. I suggest that all reference to relationships between GD and GDE and ranges be removed as speculative and outside of analyses and data.

While I am supportive of and interested in the questions addressed in this paper and think analyses of these data can certainly be made publishable, I cannot support the publication of the paper in its current format. My primary concern is that the data analyzed are inappropriate for addressing the questions posed. mtDNA are not neutral, do not recombine, are not well correlated with demography, are not correlated with neutral genome-wide diversity or adaptive potential (e.g., Allio et al 2017 Mol Biol Evol; Edwards and Bensch 2009 Mol Ecol; Galtier et al. 2009 Biol Lett; Galtier et al 2009 Mol Ecol). For example, in *Drosophila*, the mt genome plays an outsized role in clinal adaptation and is suspected to be heavily involved with climatic adaptation (see Quantifying the relative contributions of the X chromosome, autosomes, and mitochondrial genome to local adaptation Lasne et al. 2019 Evolution). Such data are unsuited for testing demographic hypotheses should be analyzed and interpreted as if they represent adaptive genetic diversity (see Yiming et al. 2021 Ecography). The firm conclusion of the Galtier et al 2009 ME review and references therein are that mtDNA cannot be used as a general population genetic measure of the diversity of anything but mitochondrial DNA.

Author response: While we acknowledge the reviewer's point about mtDNA, we think that this opinion is overstated in the context of a macrogenetics study of this scale. The stated shortcomings were currently acknowledged in the previous version of the manuscript, but they do not outweigh the practical advantages of being able to use mtDNA on such a large global scale and are not in line with evidence that mtDNA patterns do generally relate to important processes beyond "anything but mitochondrial DNA", (with a healthy dose of probabilistic uncertainty).

I am happy we agree on the critical issues of non neutral data, no recombination, little correlation with demography, no correlation with neutral genome-wide diversity or adaptive potential (both of which are critical for conservation and management. I disagree that the opinion is overstated. This is a significant list of major issues that the availability of data does not make go away. Indeed the aggregation methods used exacerbate these issues as described below. I also disagree that the issues were well addressed in this draft or the previous draft. I have read this draft a number of times, and while additional citations are given they are not well discussed. This is well a established literature with strong consensus and many more references than I gave. I encourage the authors to fully engage with this literature.

The Edwards and Bensch 2009 paper was part of a vigorous debate about the merits of mtDNA data for the inference of detailed complex demographic histories used for phylogenetic/phylogeographic reconstruction used at that time. That particular debate has long been settled, but we are asking much less of this locus in calculated genetic diversities rather than reconstructing the detailed demographic history of each OTU. Rather, we argue that aggregating large numbers of OTUs per grid cell at large geographical scales reflects patterns of mtDNA that could relate to coarse-level processes that span demography, selection, and selectively neutral genetic mechanisms.

The Edwards and Bensch debate was in the context of phylogenetics but was fundamentally about whether mtDNA can be used to infer evolutionary processes. The consensus on both sides was that in many cases it was useful for describing pattern but it was not suitable for inferring process. The entire paper is framed around evolutionary processes and their relation to mtDNA diversity. The process vs patterns consensus clearly holds for pop gen and the current study. Regarding averaging sequence from many species mtDNA and nDNA evolution are uncorrelated (as we agreed above) so mtDNA can only be used to draw inferences about mt. Second the level of evolution is

the population (both adaptively and neutrally). No matter grid cell size variation the average diversity and evenness of hundreds of species cannot reflect population level processes. If these coarse level process (demography, neutral and adaptive evolution) are the focus of the paper and the authors think their data reflects them it authors must explain how with support. I cannot see an easy way to evolutionary interpretability.

For a good example of connecting intraspecific mtDNA diversity to coarse-level population histories, a very recent macrogenetic study of ~15,000 European butterfly mtDNA sequences from ~300 species (Dapporto et al. 2019, <https://doi.org/10.1111/1755-0998.13059>) found a pattern of depleted diversity in previously glaciated areas, a classic prediction of founder effect expansion after the last glacial maximum (i.e., the classic idea championed by the late Godfrey Hewitt). Rather than this observation only being related to the mitochondrial DNA of butterflies as the reviewer here suggests, it is more likely that the community history of post-Pleistocene expansion had a detectable impact on mtDNA diversity patterns. One could counter that we are merely seeing the effect of linked positive selection occurring in the previously glaciated areas, yet these two processes (demographic founder effect and positive linked selection) are not mutually exclusive and are both worthy of discussion.

I am aware of this paper. I agree that there are interesting spatial patterns in the data. However, the authors analyzed species separately (I believe) and did not average values across species. This paper does not support the authors approach.

The authors first prediction is:

“First, since high selection, absence of recombination, erratic mutation rate and stochastic variation in population size make overall mtDNA divergence highly unpredictable (Allio et al., 2017; Galtier et al., 2009; Nabholz et al., 2009; Romiguier et al., 2014), we expect to find no correlations between mtDNA diversity (haplotype diversity) and species traits related to population size, dispersal capability, number of generations and climatic tolerance (prediction 1).”

As I’ve hoped to communicate my concerns are well known and established and clearly expressed in this paper.

Further the authors found:

“Despite the theoretical expectation that levels of neutral genetic variation should increase with effective population size, no relationships between range size (a probably good proxy for population size) and mitochondrial DNA diversity has emerged in our extensive data set”

"Moreover, the fall of the neutrality assumption for mtDNA implies the importance of mtDNA variation in influencing functional traits. Recent evidence proved that these mitochondrial-derived traits are involved, among others, in local adaptation to climatic conditions (Toews et al., 2014)."

This paper found spatial variation in their data.

Another example of mtDNA data reflecting important features of evolutionary history is from humans. While mtDNA data from humans are unable to detect such things and admixture from human/Neanderthal interbreeding, the genetic diversity and divergence patterns of human mtDNA have always reflected origins in Africa with subsequent expansion into the rest of the planet, something that has largely been corroborated with whole genome data.

I apologize if I am missing something here. I do not see the relevance of this for the current study. They almost certainly used haplotype diversity and not nucleotide diversity (as in the above reference) and this is a very different question in a single species. As noted above mtDNA can inform pattern, sometimes, for some species, in some situations. This work is about spatial variation and not conservation, biodiversity etc.

While we strongly acknowledge the study by Galtier et al. 2009, observed mtDNA patterns reflect combined demographic and selection processes in addition to a large stochastic component (i.e., the coalescent). This is consistent with the general observation of a large amount of noise in macrogenetic studies (not to mention variation in mutation rates and generation times etc).

I disagree that the Galtier paper is strongly acknowledged. It is cited only twice and only in the context of mtDNA sequences being a single locus. This critique was much more substantial with a whole section entitled “the worst marker ever”. The opening 2 paragraphs and the closing 2 paragraphs do not seem to me to acknowledge Galtier’s critique.

The other papers the reviewer cites are interesting studies that we now cite: “These single-locus markers have drawbacks: notably, the genetic diversity that mtDNA captures is from a single draw from the stochastic coalescent process operating within a lineage and may be subject to selection in addition to neutral demographic processes 62–67.” Allio et al. 2017 shows that mtDNA rates are variable and faster than nuclear DNA rates, which in itself does not contradict any specific inferences or conclusions we make.

I disagree. You average across species with very different mtDNA mutation rates (sometimes as much as 100 fold variation). This variability directly means these quantities should not be averaged. They are also uncorrelated with nDNA evolution. The therefore cannot be used to make the same inferences that neutral nDNA can make.

However, variable rates could have impacted the observed patterns and we now cite this paper (see above) to highlight this potential confounder. The results of Lasne et al. 2019 are only based on a single taxon and therefore should not be used to make generalizations across all insects. The study of Yiming et al. is especially interesting, but the data is MHC, a locus that is thought to be impacted by balancing selection (which would not specifically impact haploid mtDNA).

Lasne was just an example and I hope the authors will search further. For example please see also Hurst and Jiggins Proc B

“We review here the evidence for indirect selection on mtDNA in arthropods arising from linkage disequilibrium with maternally inherited symbionts. **We note first that these symbionts are very common in arthropods** and then review studies that reveal the extent to which they shape mtDNA evolution. **mtDNA diversity patterns are compatible with neutral expectations for an uninfected population in only 2 of 19 cases.** ... We therefore conclude that **these elements often confound the inference of an organism's evolutionary history from mtDNA data and that mtDNA on its own is an unsuitable marker for the study of recent historical events in arthropods.**”

Text from the manuscript: “In contrast, species richness and GD may be decoupled due to the “cradle” hypothesis that predicts higher speciation rates, more population structure, and smaller range sizes leading to Rapoport’s Rule, the tendency for species’ range sizes to increase with increasing latitude. Second, we might predict that the influence of Late Quaternary climate fluctuations to have an impact on GD through population demographic processes influenced by cyclical variation in precipitation, temperature, and glaciation patterns where areas with more stable climatic histories are predicted to have increased GD. Finally, we consider the influence of human disturbance on patterns of assemblage-wide GD, which we expect to decrease in magnitude in areas of high human influence.”

Each of these hypotheses acts through demographic processes acting on populations. mtDNA is not suited to test them.

Author response: We agree that mtDNA is ill suited for detailed inference of complex demographic and/or phylogenetic history. As laid out in our previous response (and examples) above, basic coarse-level demographic processes are completely decoupled from mtDNA diversity patterns.

I suspect that the final sentence is a typo, but I agree 😊. However the authors have not, at least to my satisfaction, demonstrated that arbitrarily sampled sequences averaged across species disregarding populations can say much if anything about evolutionary processes, coarse or not. Much of the writing does not suggest coarseness or tentativeness in with conclusions.

The Galtier review is particularly compelling and comprehensive and those criticisms of using mtDNA as a measure of general genetic diversity have, to my knowledge, only grown stronger. If the authors disagree with this substantial literature and feel pragmatism is warranted those criticisms need to be directly confronted in text if the reader is to be able to evaluate the paper on its own. The strong conclusion of the paper can be summed up in this quote: “For all these reasons, mtDNA is perhaps intrinsically the worst population genetic and phylogenetic molecular marker we can think of.”

Author response: We thank the reviewer for pointing out that we should clarify the limitations of connecting mtDNA with demographic history. We acknowledge that it is a noisy relationship, yet not an empty one”

“These single-locus markers have drawbacks: notably, the genetic diversity that mtDNA captures is from a single draw from the stochastic coalescent process operating within a lineage and may be subject to selection in addition to neutral demographic processes 62–67.”
“While intraspecific mtDNA diversity data is insufficient for making detailed inference of demographic history or phylogenetic reconstruction 67, we instead treat it as an important variable to study on the macroscale, rather than a population genetic marker. Macrogenetics is

a relatively new field and thus basic patterns are still not established. Our study is a step in this direction and suggests avenues for ways to test specific hypotheses.”

I admit that do not understand what the authors are trying to say here. The manuscript until this point has been fully about population genetics. I can't imagine a scenario where a paper about nucleotide diversity and spatial distribution of a population genetic marker isn't about population genetics. Given the agreed upon significant weaknesses of the marker the authors need to make the case that they can get a coarse patterns in aggregate rather than assert that they can. I detail specific problems with their metric below.

The authors of this manuscript correctly point out that their approach follows the methods of previous studies, some of which were published in this journal (e.g., Theodoridis et al. 2020). These papers too, to varying degrees, did not address the substantial shortcomings of using mtDNA to estimate genetic diversity. Galtier also predicted the continued use of mtDNA as a population genetic marker:

"Despite the many concerns we express about the use of the mitochondrial marker in molecular ecology, we are confident that scientists will keep on analysing mtDNA variation in the wild. The reason why we think mtDNA will keep its popularity is its practical convenience".

Author response: The paper by Galtier et al. made some important findings, but was specifically referring to the use of mtDNA for single species population genetic studies and/or phylogenetic studies. In contrast, we are not using mtDNA as a population genetic marker in the traditional way Galtier et al. are referring to. We are using available mtDNA for a macrogenetic study aggregating intraspecific diversity patterns from large numbers of co-distributed OTUs and correlating this with a suite of environmental variables. Using these observed correlations to infer eco-evolutionary and demographic processes is beyond the direct scope of our study or any other macrogenetic study to date (with the possible exception of Exposito-Alonso et al. 2022, DOI: 10.1126/science.abn5642). Instead, we are using the results to discuss possible explanations of these correlations (i.e., we are not inferring them with a process explicit model ; see DOI: 10.1126/sciadv.abj2271 for review). Therefore, our study is a valuable and important first step that generates hypotheses that will inspire and lead to future studies.

I think the major point of disagreement has not been well communicated. mtDNA cannot capture demography etc at the within species level as we clearly agree. The authors assert that taking the average of a number across species that does not do the job they wish it would in a single species is somehow better or different. My contention is that averaging does not generate the ability for mtDNA capture these processes even in a coarse way and the authors have not cited data to support their contention that it does. These are population level process. Given millions of years of independent evolution mtDNA with very different mutation rates I think it is clear that averaging only makes it worse. The authors do not present the context of their findings as 'possible explanation' (see for example the opening two paragraphs and the closing two paragraphs)

Despite my harsh criticisms of mtDNA work, I do appreciate there is value in exploring global patterns of mtDNA diversity. I suspect that there are many interesting processes related to the biology of mtDNA underlying some of the patterns detected. However, the sentiment the impressive volumes of data for mtDNA allow these markers to overcome their shortcomings is,

to the best of our knowledge, not well supported by data. If mtDNA cannot measure the quantities of interest increasing the volume of data will not help (Paz-Vinas et al 2021 Ecol Let; Galtier et al 2009 ME).

Author response: We are glad that the reviewer found value in our study that uncovers statistical correlations between mtDNA GD and environmental variables. Following the logic of Pilowski et al. 2022, our phenomenological model finds some unexpected correlations that can suggest the mechanisms that produce them. Actual causation (be that of regional demographic histories or histories of selection) are points of discussion and ultimately hypotheses that are testable by process-based modeling in future studies, not points of direct inference or conclusion.

Again the authors very generally do not present their findings as tentative.

Putting aside the appropriateness of this measure of mDNA as a population genetic marker there are also significant methodological issues with this work. Again the authors have followed previous studies but these studies share the same flaws. Schmidt and Garroway 2021 (Con Gen) note that mean genetic diversity of arbitrary numbers of species aggregated from within arbitrary sized grid cells is not a meaningful population genetic or conservation relevant quantity. This and the previous studies the authors followed take the mean genetic diversity of across all species sampled per cell regardless of the fact that the species to be averaged will have extremely different life and evolutionary histories, which will produce different equilibrium genetic diversities, and the fact that mtDNA mutation rates can vary extensively.

Author response: We appreciate the argument from Schmidt and Garroway 2021, but note that we do at least restrict our study to insects and that our second summary statistic is actually a way to measure the variance in GD that can come from variation in life and evolutionary histories (GDE).

I'm afraid this response does not help. I can think of no instance where averaging any sort of genetic diversity across species would be appropriate. No study of genetic diversity using SNPs or microsats would average any genetic metric across species. I cannot see such a paper getting out to review. As noted below and by others there is an approach that would enable not averaging. The authors could treat species as a random effect and allow slopes and intercepts to vary. This is easy in the brms package that they use. You could then calculate the relationship between genetic diversity of each species and any other quantity and ask whether there are overall consistent general relationships. This avoids averaging and because relationships are calculated within species things like variable diversity, histories, and mutation rates would not matter.

Additionally this method of data aggregation means that different numbers of species and indeed different species are being compared. Millette et al grouped mtDNA sequences by species within grid cell and I suggest the authors explore this method. It too is not ideal but here nucleotide diversity is interpretable. The authors are careful to call their metric mean genetic diversity until the discussion where they slip into making inferences about intraspecific variation. Again the previous studies call mean genetic diversity intraspecific genetic diversity too. I cannot see how the mean of multiple species gives information about intraspecific variation. The Millette approach is a measure of intraspecific mtDNA genetic diversity—this does not alleviate concerns about the inappropriateness of mtDNA as a population genetic marker.

Author response: Our GDM and GDE are both derived from the classic average pairwise distances (per OTU), which is sometimes referred to as nucleotide diversity and is essentially a useful measure of intraspecific or intrapopulation or intra-location per species variation in genetic diversity (depending on what unit one chooses to sample). GDM is the average of this per grid cell and GDE is the evenness. It has desirable and robust sampling properties (as opposed to allelic richness). We clarified this by saying “mean intraspecific variation” instead of just saying “intraspecific variation” alone.

Again the authors appeal to previous studies with little argumentative back up. These GDM and GDE metrics are not classic having only been used in the handful for recent macrogenetic studies.

I appreciate the intention behind using genetic diversity and evenness to partition diversity patterns into different components that together are more informative than a single metric. This is a widely used method in ecology to attribute biodiversity differences due to changes in total abundance across an assemblage and changes in the species abundance distribution.

In population genetics we take similar approaches comparing evenness metrics (e.g., expected heterozygosity) to richness metrics (allelic richness) to infer, for example, population size changes.

However, the genetic diversity mean and evenness metrics used here are not completely analogous to their use in macroecology and in my view they don't measure what they purport to measure. I am open to being corrected, but my reasoning for this view is below:

L97-101: Existing macrogenetic studies, which are inherently correlative, have only described average intraspecific genetic diversity (GDM), and are therefore unable to determine whether high average genetic diversities are due to high diversity within most community members or to the effects of a few taxa with extremely high diversity (Fig. 1; see Methods)."

As the authors note, because the mean cannot distinguish a high value from many species having relatively high GD or from few species having extremely high GD, it therefore explicitly does not measure intraspecific genetic diversity. It cannot be used to test questions related to patterns of intraspecific genetic diversity. It is also uninformative about evolutionary processes acting across species because similar patterns across cells can arise for completely different reasons. A cell might have high diversity because there are several species whose abundances (and therefore genetic diversity) increase with temperature, whereas in a neighboring cell genetic diversity might increase because one species originated in this region and has extremely high diversity there relative to other species.

Regarding the mean as an assemblage level summary: The distribution of genetic diversity is right skewed (lines 464-466: "The distribution of GD values within an area of low GDE can take a variety of shapes, but the most common in our observed data is markedly L-shaped whereby most OTUs have low or zero GD along with a small number of OTUs with large GD (Fig. 1d)."

The mean is not an appropriate summary statistic for skewed distributions. The authors nevertheless use the mean and use this fact as a rationale for using evenness.

Coming to evenness: this is estimated taking the Hill number approach, where evenness is the Shannon index divided by the number of OTUs for a cell (line 453). This metric mathematically depends on the number of sampled species in the cell. The authors have done sensitivity analyses based on different minimum numbers of OTUs (between 10-200 OTUs/cell), but these tests cannot test sensitivity for evenness because evenness necessarily decreases with increasing numbers of OTUs in a cell. And, there is still a lot of wiggle room for sample size effects if the maximum sample size is over 1000 OTUs with a 200 OTU minimum. It therefore makes absolute sense to me that evenness is significantly correlated with latitude (lower at higher absolute latitude) and with freezing (higher evenness in areas that rarely freeze). These are where more taxa are sampled. I would be interested in seeing whether these relationships disappear if the number of taxa is modelled as another covariate to control for species sample size. Besides these issues with metrics, the authors' statements in the rebuttal do not match the revised manuscript. The authors say that they do not test hypotheses, but generate them. This contradicts the clearly worded hypotheses to be tested on lines 104-119, and statements made throughout the manuscript (e.g. L289-290: "We are hypothesizing that both larger ranges and climatic stability could have a positive effect on GD, yet they are different processes and do not necessarily co-occur"). Again, neither of the demographic hypotheses proposed here can be appropriately tested with only mitochondrial data.

Pas-Vinas et al note that arbitrarily aggregating sequences across large spatial areas create implausible populations. This is true in the extreme. Populations are not defined genetically and contain multiple species. Most inferences in this paper refer to population level processes (evolutionary change, relationships to environments, drift etc).

Author response: We agree that summarizing genetic variation across OTUs within arbitrary grid cells is a compromise out of pragmatism. We agree with the reviewer's point brought up by Pas-Vinas et al is an interesting dynamic to consider for macrogenetic studies. Macrogenetic studies are inherently different from single species population genetic studies with different goals and approaches but there is a shared inherent abstraction of what a "population" is in relation to a geographically defined sample. They are rarely congruent, yet methods based on population genetic theory can still yield useful results despite that populations are rarely homogenous or well defined (Barton et al., 2019; <https://doi.org/10.7554/eLife.45380>). Following this interesting point suggested by the reviewer, we now quantify and report that the vast majority (~75%) of the sampled OTUs are only found in single grid-cells (Supp. Fig. 11). Coalescent theory with migration (i.e., the structured coalescent; Nordborg and Krone 2001 and Nordborg 1997) predicts that local sampling can capture variation associated with unsampled areas and that this can lead to higher average pairwise distances (i.e., higher GD). We don't see this as a bug, but as a feature that enables us to potentially capture information about species with larger and/or more isolated ranges. In relation to this important dynamic brought up by the reviewer, we now cite a recent macrogenetics paper that directly looks at links between range size and GD empirically and theoretically (Alonso-Espisito et al. 2022).

This helps for sure. Thank you.

Paz-Vinas also showed that sequence deposition is often incomplete and idiosyncratic authors tending to deposit only new sequences in a region. This too will significantly affect their

measures of genetic diversity.

Author response: We are well aware that authors (and participants in the BOLD initiative) neglect to deposit duplicate allele copies, and this is one reason why we chose our metric of GD to be average pairwise distances. This metric has several desirable statistical sampling properties (Tajima 1983) including approaching its expectation with only 5-10 samples.

Therefore, missing duplicate samples, while positively biasing this summary metric will likely have a negligible effect. As mentioned to reviewer #1, we explore and verify this explicitly with simulations given a range of sample sizes and have included it in the new version of our manuscript, in the Supplementary Materials, and main text:

“To explore this sampling dynamic explicitly, we conducted coalescent simulation experiments comparing how the calculation of GD varies given identical samples with and without duplicate alleles removed. These simulations showed that retaining only unique haplotypes resulted in a small, consistently upward bias in GD values. Additionally, increasing values of effective population size (N_e) decreased this bias, with estimates of GD with and without duplicate alleles converging for N_e values greater than $\sim 10^5$ (Supplementary Materials, Supplementary Fig. 15)”

This also helps. Thank you.

The predictive maps look good. However, I am very leery about interpolating the globe from just 187 grid cells. The posterior checks look good but that just means the model can predict the data used to generate it, which is important! But predicting different continents with different species groups and geological histories with such a small sample seems fraught.

Author response: We have new results based on a more clear modeling procedure as well as a range of minimum threshold for number of OTUs power per grid cell. As with the original manuscript, we still do not extrapolate to environments beyond what are used to train the model. In addition, we added a model validation step to our modeling process, where we train the model and predict it to test data that was withheld. The data are stratified according to continent to enforce the most even spatial sampling possible. We highlight potential spatial biases in prediction by coloring observed versus predicted plots by continent and find no obvious spatial patterns in the residuals (Supp. Fig. 3 & 6). We do agree that our study has a biased focus on North America and Europe, and we now mention this in the Discussion:

“Sampling biases, including the overrepresented North American and European sampling we find, may also influence this relationship, although we find no evidence of an influence in our data (Supplementary Fig. 14, Supplementary Table 5).”

“Additionally, we find that OTUs occupy more grid cells between 40° and 60° latitude (Supplementary Fig. 14), matching our expectation for larger ranges at higher latitudes, although we acknowledge that sampling bias may contribute to this pattern as well.”

Great, thank you. This also helps. Sample size remains low for global prediction but this check is useful.

Finally, I am concerned with the conservation claims made in this paper given the use of mtDNA. Claims from previous papers (e.g., Theo) are already being misapplied. Their mtDNA is now included in UN Biodiversity planning despite the fact that it is not useful for conservation. In conservation we are interested conserving adaptive potential. Neutral genome-wide genetic

diversity is not well correlated with adaptive potential and organellar genetic diversity is not correlated with either. Strong conservation claims made from analyses of this marker are thus dangerous.

Author response: We agree that conservation claims should be made with caution and therefore we do not make any strong claims regarding how our study should directly inform conservation decisions. While anything gained from our study should be done with care, macrogenetic studies like ours are a potential first step to help inform conservation goals (along with other information), which is why we cited DeWoody et al. 2021 and and Reed & Frankham 2003. Likewise, studies like ours could aid in monitoring efforts.

I disagree that studies like this should aid in monitoring. That in fact is my explicit concern. They should not. mtDNA is simply not correlated with adaptive genetic diversity, genome wide diversity, or demography in a way that is useful for conservation. If the authors disagree I request that they argue their point with references. This is clearly laid out in the references provided.

It is also incorrect to state that the authors do not make strong conservation claims. There are uncaveated claims throughout.

For example the below claims are provided as context for their results:

Line 369: Genetic diversity is critical to the survival of insects and their complex interactions with other organisms. High genetic diversity may facilitate adaptation to changing climates, 123–125 emerging diseases, and pollutants: three (of many) potential drivers of the “insect apocalypse” 50. In addition, genetic diversity contributes to the diversity and stability of species interaction networks by affecting niche space and competition 126, community structure 127, and network complexity 128. At larger ecological scales, insect genetic diversity may reflect ecosystem function and structure as reliably as other traditional macroecological metrics such as species richness 129. It can also augment the resilience of ecosystems that provide continuing services for humankind 14, such as disease management, curbing the spread of invasive plants, aiding sustainable agriculture, pollinating food crops, and controlling pests 13.

Line 388: “By modeling relationships between environmental data and two complementary measures of intraspecific genetic diversity, GDE and GDM, we can make assemblage-level genetic diversity predictions for data-poor regions of the planet, while flagging and masking those with high uncertainty ¹³⁴ (Fig. 2).”

Reviews of reviews

I would still not subscribe that mtDNA diversity, in general, bear *no* information about demographic and evolutionary processes, and, in particular, I do not agree that neutral genetic variation is useless. The authors do not claim to measure adaptive potential, they (with shortcomings that the colleague and myself have highlighted) attempt to test for historical and demographic hypotheses.

Author response: We would further say that we don't take it this far. Our study uses correlative methods, but we do discuss putative historical, evolutionary and demographic hypotheses. We treat genetic diversity as an interesting variable to study on the macroscale, likely shaped by historical processes impacting demography (as well as by selection, and neutral mutational mechanisms). But we do not view it as a population genetic marker for directly testing demographic hypotheses. This is a relatively new field (macrogenetics) and thus basic patterns are still not established. Our study is a step in this direction and suggests avenues for ways to test specific hypotheses.

We agree that it doesn't bear "no information" however the references we supplied show that it bears vary little information in most contexts and is clearly not systematically related to demography or evolutionary processes. It is very clearly not related to genome wide diversity which all claims in the paper hinge on.

I certainly think that neutral genetic diversity IS useful and my primary criticism and the primary reason it does not relate to demography and evolutionary past well is that mtDNA is NOT neutral. I would be much happier if it was although averaging across species would remain incorrect. mtDNA genes underlie very important respiratory processes. Hints demography and evolutionary pasts will be eradicated when it is averaged across hundred of species that have diverged for millions of years.

The authors do make many claims about adaptive potential. For example:

Line 369: Genetic diversity is critical to the survival of insects and their complex interactions with other organisms. High genetic diversity may facilitate adaptation to changing climates, 123–125 emerging diseases, and pollutants: three (of many) potential drivers of the "insect apocalypse" 50. In addition, genetic diversity contributes to the diversity and stability of species interaction networks by affecting niche space and competition 126, community structure 127, and network complexity 128. At larger ecological scales, insect genetic diversity may reflect ecosystem function and structure as reliably as other traditional macroecological metrics such as species richness 129. It can also augment the resilience of ecosystems that provide continuing services for humankind 14, such as disease management, curbing the spread of invasive plants, aiding sustainable agriculture, pollinating food crops, and controlling pests 13.

These are strong claims that cannot be tested with mtDNA. No study of any single non-neutral gene from any part of the genome would acceptably make this claim. I believe Taking the average of such a gene's across hundreds of species makes even less sense.

In any case, we strongly agree that the limitations brought forward by the referee and the literature she/he cites should be better acknowledged in the study, along with the highlighted technical issues entailed by the use of raw genbank/bold data (e.g. the bias due to the fact that many authors only deposited unique sequences in databanks).

Author response: We have added further discussion of limitations and have now included simulations showing that omission of duplicate alleles has a minor impact on the bias (Supp. Materials; Supp. Fig. 15).

I think this is better done although the too strong claims remain

I also completely agree on the suggestion to explore Millette et al's approach of considering species identity (which I should have mentioned myself), while it looks to me that the issue of aggregating populations into "arbitrary" units mentioned by Pas-Vinas is more a concern regarding other details of Millette's approach.

Author response: We now show that the vast majority of OTUs are restricted to one grid cell, such that the Millette approach wouldn't work well. We also explain that the grouping into grid cells of arbitrary size is a useful compromise and less of a concern than stated with respect to prediction under the coalescence with migration/isolation or metapopulation. If ranges exceed the grid cell, local sampling is predicted to yield higher GD under a wide range of circumstances (Nordborg and Krone 2001; Charlesworth and Charlesworth 20210, chapter 7 for details), while absolute isolation leading to local coalescence is also not problematic.

The Millette approach would be straightforward to implement and would make the paper acceptable for publication in my opinion.

Already in my original report I made clear that the study requires substantial upgrade before being considered for publication. The concerns raised by the colleague certainly do not improve my judgement, but I would not say that, in principle, I am against publishing any such study just because it is based on mtDNA.

Author response: A point-by-point response to the reviewer's original comments can be found above.

I agree that mtDNA work merits publication. I did before and continue now to encourage the authors to develop hypothesis related to the biology of mtDNA

In general, I partially agree with the other reviewer. In fact, there are other markers or genomic techniques that can be used to study population genetics as the other reviewer has listed, however the manuscript focuses on intraspecific genetic diversity across species at a global scale, which is recently called macrogenomics. In order to do it, the authors used available sequences of cox1 that covered the world, which is probably the only fragment that is massively sequenced and can be used in a global scale study of intraspecific genetic variation.

Author response: We agree that our use of the potentially problematic mtDNA is a useful compromise that enables the scale of sampling to be done at the macrogenetic scale. Using something like microsatellites mined from the literature would be dwarfed by our study in scale and scope, and also come with its own baggage with regards to relation to neutral or adaptive processes and or rate variation/saturation.

I (with apologies) reiterate that the volume of data does not overcome the significant limitations with the analytical approach (averaging across species), the marker (which the reviewer notes they agree upon below), and the much too strong inferences made.

Although the other reviewer is right in “mtDNA are not neutral, do not recombine, are not well correlated with demography, are not correlated with neutral genome-wide diversity or adaptive potential”, cox1 provide an accurate species delimitation because the genetic divergence clusters species separately and the cladogenesis is well captured.

We are happy the reviewer agrees but do not see a way to the strong claims made in the paper from this agreement. We agree mtDNA is often good for species delimitation but that is not what the authors of the paper use it for. They use it for a measure of genetic diversity which is highly non standard outside of recent macrogenetics work

Moreover, intraspecific genetic diversity and structure of cox1 corresponds to biogeography and dispersion (e.g., Papadopoulou et al, ME, 2009; Baselga et al., Nat Comm, 2013; Múrria et al, ME, 2017) and this fragment is widely used in macrogenetics (Leigh et al, Nat Rev Gen, 2021). Unfortunately,

data available for neutral markers such as microsat or the mt genome of *Drosophila* that the other reviewer suggested, is limited and currently it is available for a small number of species, but the data available is far for a global scale study. Moreover, I have my doubts if microsat data can be easily compared across species because the markers used are commonly specific for each species. Until genome information is not available for thousands of species at the population level, i.e., several individuals sequenced for each population, we cannot use mt genome sequencing in macrogenetics. Also the authors are not clearly interested in adaptive genetics or general genetic diversity, as the other reviewer indicated, instead authors want to test macroecological patterns of intraspecific genetic variation for as much number of species as possible covering the widest area. For this reason, authors specifically analyzed the intraspecific DNA barcode diversity, which is clear in the manuscript. *Author response: As mentioned above, we largely agree with all of this. As the reviewer describes, we use the only data source with the geographic and taxonomic scale available to answer this question. While not perfect, is the standard approach in macrogenetics thus far in other taxa and provides a baseline for future work.*

I maintain that mtDNA is interesting but it's treatment in this study and the inferences drawn (as with previous studies) are inappropriate. The authors very clearly make claims about adaptive potential directly and indirectly when they cite the conservation relevance of their marker.

2. Do you think the mtDNA data analyzed are appropriate for addressing the questions posed? Despite the mtDNA data analyzed has limitations, I think the data used can address the questions posted and the study is well-performed. However, assuming the limitations of cox1 data, I suggest authors rephrase the hypothesis based on the comments by the other reviewer. In fact, some of the hypothesis based on demographic processes cannot be directly tested using the available data. Improving this section is critical. As the other reviewer, I agree in “I suspect that there are many interesting processes related to the biology of mtDNA underlying some of the patterns detected.” I may focus the new hypothesis on these processes.

Author response: As previously mentioned, we don't actually test these hypotheses via inferences of underlying processes, but instead we conduct a correlative study using a phenomenological model and discuss possible interpretation and processes that could be more directly tested via process-explicit models in future studies. In other words, our study does not really test hypotheses, but rather it generates them (Pilowsky et al. 2022 for a discussion of the

distinction between these two types of modeling approaches).

I just don't see this distinction as clearly communicated in the paper or doing the argumentative work that the authors think it does. If the data is unsuited to testing the hypotheses why would it be suited to generating them. They are poor measures of all the quantities of interest at best as is agreed upon by all reviewers and the authors. I have argued strongly with references that they are worse than poor measures. The support for the approach seems to be related to data availability and not appropriateness.

Another interesting criticisms by the other reviewer is the metric used: mean GD in each grid. Authors must to argue extensively why this metric is appropriate, and I think it is. Moreover, authors can explain if they used the abundance of each haplotype in a grid or only the data of each unique sequences, which can introduce bias as the other reviewer suggested, specially when authors used the mean GD. This is also critical.

Author response: In the manuscript we have explained that we use a classic metric (average pairwise differences) that is robust to small sample sizes (unlike allelic richness) and now provide simulation experiments showing that this metric is also not strongly biased by the removal of duplicate alleles (Supp. Materials, Supp. Fig. 15) We have also addressed these two issues directly with other reviewers.

As noted above the average of this metric taken across species is not classic and not demonstrated to be robust.

Reviewer #4 (Remarks to the Author):

After carefully reading the main text and the revision, I can see an important improvement of the manuscript. Now it is clearer and well addressed and it can be an important contribute to understand large scale pattern of intraspecific genetic diversity of insects. The patterns found are relevant for advancing this discipline and understanding global patterns of diversity.

As required for the reviewers, authors have managed to improve the description of the methodology, which now is clearer than previous versions. Specially, criteria for sequence selection is much clearer than the previous version and the statistical analyses used for validating their datasets are robust and well explained.

Minor revision:

- Lines 254-256: I cannot see the relations between the lack of correlation with GDE and GDM, and the departure from the expectations of SGDC, because the number of OTUs are not considered in the current study.

Lines 267-270: I may remove these two sentences from "Macrogenetics is a relatively new....". There are several expectations well established, however we are still dealing how to test them.

- Legend in figure 1 looks too llong, I suggest cut here

- Check reference 11.

Reviewer Responses

Reviewer 1

Reviewer #1 (Remarks to the Author):

My impression is that the current version represents a big improvement from the original manuscript, especially in terms of the statistical analyses. I am glad to see that my suggestions have been taken into account and, at occasions, appreciated.

Authors' response: *We thank Reviewer #1 for the positive feedback and helpful suggestions that have improved the manuscript.*

Nonetheless, I would still like to suggest some streamlining to the description of the structure of the study, which, I find, is still not well defined in the introduction.

The final paragraphs of the introduction, as it is customary, present the main hypotheses and predictions to be tested in the study.

One reads about “museum” vs. “cradle” hypothesis, with the first predicting GD(M?) to be higher in tropics *because* of older species (associated with larger ranges) and the second predicting GD(M?) to be lower in tropics (associated with smaller ranges).

Then, the authors mention the effect of climate stability, which is predicted to be positively associated with GD(M?). The authors do not state this explicitly, but, as far as I am concerned, this is another (more process-based) way of stating the “museum” hypothesis.

The last idea mentioned in this section is that human activities may have reduced GDM *and* GDE “ due to general reductions in abundance with the potential proliferation of disturbance-adapted species”.

After this, one would expect that the authors tested for the effect of species range, climate stability and human disturbance, which is not what one reads next.

I would *really* like to see this section modified so that the reader gets a clear set of hypothesis associated with a set of predictions, each of them stated in terms of statistical effects that can actually be tested in the subsequent analyses (which should include *the* relevant predictors [or their proxies] and, if the case, some important controls, and nothing else, all duly explained and justified). It would be a huge improvement and it would have the further benefit of reducing the verbosity of the discussion.

Authors' response: *We thank the reviewer for these constructive suggestions, and we have now rewritten the second half of the Introduction to reflect them. We also acknowledge that this*

study (and others like it) ultimately rely on correlative models and therefore cannot really test predictive hypotheses in the sense of causative relationships to observations. Thus, we moved discussion of biogeographic hypotheses to the Discussion and use the Introduction to discuss hypotheses and predictions in a correlative manner (L109-118):

“After calculating these two metrics across all OTUs for each cell, we explore how they correlate with global environmental and geographic variables. If GDM and GDE follow classical global biodiversity trends, we expect GDM and GDE to decrease with latitude⁶⁹ and human disturbance⁷⁰, and increase with climate stability⁷¹. Once these environmental correlates of intraspecific insect mitochondrial genetic diversity are found based on information extracted from insect sampling efforts across the globe, we use them to predict patterns of insect GDM and GDE in undersampled regions, which includes most of the planet. We discuss and interpret our results in the context of classic biogeographic patterns such as the latitudinal diversity gradients in insect species richness, average and variability in range sizes, and late Pleistocene climate cycles.”

By the way, although I did not mention this in my comments the original version of the manuscript, I now believe that the hypothesis that the geographic distribution of intraspecific genetic diversity may be (also) influenced by the effect of *temperature* on mutation rates should be also discussed, as it seemed to be suggested as an important factor by results by, at least, De Kort et al. 2021 (<https://www.nature.com/articles/s41467-021-20958-2>) and Manel et al., 2020 (<https://www.nature.com/articles/s41467-020-14409-7>).

Authors' response: *Yes, it is a good point that temperature could be an important factor. Word limit restrictions made us remove this part of the Introduction and Discussion, but we now make mention of this in the Discussion (L308-312). Thank you for the added references.*

“Likewise, declines in GDE in previously glaciated areas, and the positive correlations between GDM and GDE with MTWM are also consistent with the evolutionary speed hypothesis, where warm temperatures are posited to decrease generation times and speed up mutation rates, leading to higher genetic diversity¹⁰⁰. Similar patterns are found in macrogenetic studies of fish²² and animals and plants in general¹⁰¹.”

Also, as far as I understand, the complete list of the initial 49 variables, as well as their correlations and the rationale for the inclusion of the final set of 11 variables rather than another set of uncorrelated predictors, is not provided. I think it would be a necessary addition.

Authors' response: *Thank you for pointing this out. The variables are listed in Supplementary Table 1 and we added the correlation matrices to the Supplementary Materials. Additionally, we added the following text with our justification (L530-535):*

“We removed highly correlated variables ($r > 0.75$), prioritizing variables that represent climate extremes, climate variability, habitat variability, last glacial maximum (LGM) climate stability, and human influence on the environment. We prioritized these classes of variables because the

influences of environmental extremes and variability across space and time are more likely to shape local communities than averages. (Germain and Lutz 2020; Perez-Navarro et al. 2021).

Citations:

Germain, S.J., Lutz, J.A., 2020. Climate extremes may be more important than climate means when predicting species range shifts. Clim. Change 163, 579–598. <https://doi.org/10.1007/s10584-020-02868-2>

Perez-Navarro, M.A., Broennimann, O., Esteve, M.A., Moya-Perez, J.M., Carreño, M.F., Guisan, A., Lloret, F., 2021. Temporal variability is key to modelling the climatic niche. Divers. Distrib. 27, 473–484. <https://doi.org/10.1111/ddi.13207>

I am not particularly convinced by the idea of having the “freeze” variable tested separately from the other predictors. If it is a matter of it being strongly collinear with some others, it should be clarified. In any case, as I see it, it is preferable to pick up a set of predictors based on clearly-defined hypotheses and predictions and test them all at once.

Authors’ response: *We agree that the logic of testing the “freeze” variable separately was not made clear. It was highly correlated with other climate variables in the predictor set of the full model and we now state this clearly (L559-561):*

“We tested this variable separately from the full predictive model because it is highly collinear with the deep-time temperature variability metric ($r > 0.75$) included in the model.”

I would comment in the main Results section about the efficiency of retaining cells based on different thresholds for the number of OTUs in terms of R^2 and RMSE (wouldn’t the latter be a more useful index to report than the slope or intercept?).

Authors’ response: *We report the slope and intercept in addition to R^2 and RMSE because they indicate whether predictions are biased, such that a slope near 1 and a y-intercept near 0 indicate low bias in predictions. We clarify this in the Methods (L488-493):*

“While retaining results across the range of minimum OTUs per grid cell, we focus our analysis using the threshold that results in the least-biased and most precise estimates of GD when predicting a trained model onto withheld test data, where a slope near one with a y-intercept near zero indicates minimal prediction bias and a high R^2 with low root mean squared error indicates high precision.”

And we clarify this in the Results here (L128-131):

“This threshold resulted in the lowest bias (slope near 1 and y-intercept near zero) and highest accuracy (high R^2 , low root mean squared error) when predicting trained models to withheld test data (Supplementary Fig. 3).”

I am not 100% convinced of the idea of testing the importance of the variables without accounting for SAC and then run the final model with SAC, but I understand it may have been forced by technical limitations, and it seems justified. If this is the case, however, I think it should be disclosed in the main text.

Authors' response: *We agree and now place this in the main text of the Methods section (L578-580):*

“Computational and statistical limitations due to exploring such a wide variety of variable combinations in a Bayesian context prohibited accounting for spatial autocorrelation in this first step.”

A small detail: L101. Replace “complimentary” with “complementary”

Authors' response: *Fixed, thank you.*

Reviewer 2

Reviewer #2 (Remarks to the Author):

My review is attached as a pdf. I want to note I appreciate the effort by the authors. The paper is well written and I'm certain was an immense amount of work. Unfortunately, I remain unconvinced that the data and analyses support the conclusions. The clear path to publication in my opinion is to analyze the data as Millette et al do, or perhaps better use a mixed model. You could allow slopes and intercepts to vary for each species and summarize the strength and direction of effects across species. This avoids the 'lumping' of so many species. This is another significant reanalysis. I have not searched the lead author, but leads are often students on tight deadlines. Thus I note that I might also be convinced to recommend acceptance if the very broad general significance was toned down. The conclusions are oversold, and as you will see in my attached review the opening two paragraphs and the closing two paragraphs exemplify this in mind. They seem to ignore any other stated caveat.

I remain convinced that there are interesting patterns in here and that they will be related to the capricious evolutionary biology of mitochondria. I do not think those patterns bear on conservation or evolutionary change for any genome but the mitochondria's. Despite my review, I wish you the best of luck with the paper whether it is published here or elsewhere.

Authors' response: *We thank the reviewer for their thorough review and careful scrutiny of our claims. We feel that the language of the manuscript greatly improved in response to their reviews.*

The following comments were copied from the reviewer's attached pdf

This is my second review of this paper. I remain excited about its potential but was disappointed by the responses to my comments. In my last review, I had hoped to convey my support for this work so long as the appropriateness of the data and analyses for the questions addressed were appropriately, directly, and clearly caveated. In my view the authors have in many important instances, dug in on critical issues while weakly defending their position. Many of the noted shortcomings are acknowledged in the response to reviewers. Some critical papers are now cited but criticisms are minimally discussed if they are discussed at all. Given the generality of the claims made and the weakness of the data and analyses any reader who does not read the paper very carefully, or who does not have direct expertise in the area will be led astray. If this paper was published in its current form it would contribute to growing confusion in the field together with other similar papers misusing and misinterpreting similar data (e.g., Manel et al Nat Comms; Theodoridis et al Nat Comms; Miraldo et al Science).

I will provide a brief summary of my thoughts here as the responses below got long and are very repetitive. First please imagine a study of another functional gene. One might be interested in EPAS1 which helps people use lower amounts of oxygen efficiently (mtDNA too is important for metabolism). If we could easily access global sequence data we could come up with many interesting hypotheses about how it might vary through space in relation to environments. This would be a very interesting study of an interesting gene and how it varies within a species.

Authors' response: *EPAS1 is an excellent example of how any locus (including mtDNA) contains demographic information while also being impacted by selection. EPAS1 contains information about the out-of-Africa human history as well as adaptive introgression from Denisovans (Zhang et al. 2021, PNAS, <https://doi.org/10.1073/pnas.2020803118>). This is why researchers had to model the "neutral" demographic history of humans and Denisovans to detect selection via introgression at the EPAS1 genomic region (i.e. the EPAS1 gene genealogy contained the main features of the human demographic history). If one purely looked at EPAS1 genetic diversity patterns, one would in fact get useful coarse information about the history of humans. There is nothing controversial about this: anywhere in the genome is going to reflect the demographic history of the species to some degree and will also be distorted and/or transformed by selection. It is therefore no surprise that even in well studied species such as Drosophila and humans, mtDNA data contain the major features of each species' demographic histories (for Drosophila, one can compare Hale 1991; doi.org/10.1093/genetics/129.1.103 to Arguello et al. 2019; doi.org/10.1093/gbe/evab046). Figure 2 in Ellegren and Galtier (2016) does a great job illustrating the multiple processes underlying observed genetic variation.*

Now imagine that the study took the average diversity and evenness of this gene for hundreds of species that have had millions of years of divergent evolutionary trajectories. This study would be very comparable to the study under review. I now ask the editors, reviewers, and authors to reread the opening two paragraphs and think of the above-described study. I suspect that if you received such a paper you would think that diversity for this single adaptive gene with a very specific function would seemingly obviously not be useful for almost any of the broad biodiversity, conservation, and adaptive potential claims made. Not useful for hypothesis generating or testing. These opening paragraphs make direct reference to the

utility of a single adaptive gene (barcoding) in all these instances. Claims by the authors that they are not directly testing or are generating coarse hypotheses seem misleading given the tone of this opening and indeed much of the paper.

Authors' response: *We believe that if EPAS1 was available at the same scale as mtDNA is, it would be useful for a class III macrogenetics study (i.e. large-scale which re-analyzes data from public repositories, Leigh et al. 2021). As stated above, the fact that genetic diversity (nuclear included, Ellegren and Galtier 2016, Nat. Rev., doi:10.1038/nrg.2016.58) is affected by demography, selection, and mutation does not nullify conclusions about any one of these processes, as long as they are made with appropriate qualifications, which we make and expand upon in response to the reviewers' helpful specific comments. The above-described study using EPAS1 would still be potentially useful to understand biodiversity patterns and even potentially provide an early indicator of conservation concern, as mtDNA diversity across animals corresponds with their IUCN status (Petit-Marty et al. 2020, Diversity, <https://doi.org/10.1111/cons.12756>). We agree that the impact of multiple evolutionary forces on genetic diversity in all genes, not just mitochondrial, influences both the patterns we see and the appropriate interpretation of those patterns. However, generating hypotheses about which of any of these forces (i.e. demography, selection, mutation) influences the observed patterns is not rendered useless because of their interaction. Given our statement that this study is meant to suggest hypotheses, we removed discussion of biogeographic hypotheses in the last two paragraphs of the Introduction and instead opted to discuss them in the Discussion (L109-118).*

Apologies for my lack of editing. This process has run long. I hope the points are clear despite any errors. Brief review of current draft paper with many more details in my response to responses below.

Line 39-56: We have established and agreed (I think) with references that strong selection, absence of recombination, erratic mutation rates that vary 100 fold across that mtDNA diversity is highly unpredictable and it is not correlated with genome-wide nuclear diversity (Allio et al., 2017; Galtier et al., 2009; Nabholz et al., 2009; Romiguier et al., 2014). I thus think statements about how mtDNA can inform about biodiversity at any level let alone "adaptive potential" and "ecosystem resilience" are overstated. Given that the measure in the current study are averages of hundreds of species with very disparate levels of diversity this study can in no way inform about intraspecific diversity (textbook defined as genetic diversity within a single species but used differently in the present study). I suggest these two paragraphs be recrafted.

Authors' response: *We have substantially revised the text in this regard (L42-64).*

Line 47: These references refer to genome wide variation which mtDNA is not a measure of. They cannot be interpreted as supporting the use of barcoding diversity in the context used.

Authors' response: *Thank you for pointing this out. As mtDNA variation is literally nested within genome wide variation, we consider it to be one measure of genome-wide variation (albeit imperfect). But we have removed this statement and citations and replaced them with those that specifically refer to how mtDNA relates to these objectives.*

Line 52: Conservation genetics is interested in adaptive potential. mtDNA is not correlated with genome wide nDNA diversity or adaptive potential (Galtier et al 2009 ME).

Authors' response: *Part of adaptive potential is standing genetic variation, which mtDNA is an imperfect index of. It has successfully demonstrated its utility as an early indicator of IUCN status in animals (Petit-Marty et al. 2021, Diversity, <https://doi.org/10.1111/conl.12756>), which we now cite (see below). However, we acknowledge that it is an imperfect index, so in response to this comment and your following comment we toned down our statement about the conservation potential and removed statements about the value of neutral genetic diversity (L48-64).*

“Large-scale georeferenced DNA barcode surveys have great potential beyond their original use as biodiversity assessment tools and are increasingly utilized in global studies of biodiversity 8,9. In addition to accelerating taxonomy 10, DNA barcoding has become a tool in diagnosing conservation status and more generally understanding ecology, evolution and biogeography 11–13, but see ref. 14. More recently, DNA barcode surveys provide data for the emerging field of “macrogenetics” 15,16, which has the potential to become one of the many approaches used to monitor and protect genetic diversity 17,18. Class III macrogenetic studies, as defined by ref. 19, summarize the geographic distribution of intraspecific genetic variation of a taxonomic group across broad geographic scales and use data from large numbers of species aggregated from public repositories. Typically used to correlate environmental variables that predict this critical component of biodiversity 19,20, previous class III macrogenetic studies have focused on vertebrate groups, uncovering links between global patterns in intraspecific mitochondrial genetic diversity, species richness, and phylogenetic diversity 21,22, while documenting latitudinal gradients 23–25. They have provided mixed support for the influence of human disturbance on mitochondrial genetic diversity 21,23,26,27, and suggest that climate stability 21,27 and species’ range sizes 20,28 affect intraspecific mitochondrial genetic diversity on a global scale.”

Line 53: I’m not sure of the relevance of statements about neutral genetic variation are to this study because mtDNA has highly functional critical genes and undergoes strong selection. I suggest deleting this line

Authors' response: *We have removed this line.*

Line 81: It is good to note this drawback. I suggest the authors note the many issues with averaging measures of single within species draws of the coalescent process across highly diverged species

Authors' response: *This is a good point, we have also added a point about variation in COI mutation rates across insects (yet the cited study of Allio et al. has a limited sample for insect taxa) (L85-88):*

“Therefore, it is important to note that while mtDNA has useful properties 12, mtDNA diversity patterns are impacted by multiple processes that distort the patterns shaped by phylogeographic demographic histories, including selection, high coalescent variance, and mutation rate variation across taxa 52,61–64. ”

Line 83: This is an agreement that lots of data outweighs the significant drawbacks of this

marker and the analytical approach. I obviously disagree and can only see the approach aggravating issues. Nevertheless the papers cited here do not support the statement. They refer to eDNA studies, utility in species delimitation, and early steps at calculating abundance from the number of sequences in samples. They say nothing about estimating and genetic diversity metric which indeed is a very non standard use of barcodes.

Authors' response: *Thank you, we removed the first citation, which uses estimating richness as an application. However, the second cited paper does focus on use in estimating genetic diversity (and population genetic applications in general). We therefore chose to retain the second citation and other citations more specific to mtDNA barcodes. Additionally, we refined the text to not suggest that more data outweighs drawback (L88-91):*

“While taking these complex dynamics into consideration, sampling the mitochondrial genetic diversity of thousands of taxa per locale is a promising initial step towards understanding an important component of biodiversity at global spatial scales^{10,12,65,66}.”

Line 157: Dichotomizing continuous variable is generally bad practice

Authors' response: *While we generally agree with this statement, the dichotomization is based on a climatically and biologically relevant division, where 0° C indicates freezing potential and results in a division of the globe into areas that do freeze and do not freeze annually. Existing classifications of glacial/post-glacial areas do not capture this demonstrably biologically relevant division (White et al. 2019, “Regional Influences on Community Structure across the Tropical-Temperate Divide.”, Nat. Comm.). We had cited this paper in the Methods but not in the Results, but on reflection we now include this reference in the Results.*

Line 165: Including highly correlated variables as the authors have done is fine for predictive modelling, which the mapping exercise is. But it removes the biological interpretability of individual effects due to violating the assumption of independence among variables in regressions.

Authors' response: *We agree with this point. In fact, we did remove strongly correlated variables ($r > 0.75$). In response to another reviewer, we added additional text for our rationale for keeping certain variables, so the statement now reads (L530-535):*

“We removed highly correlated variables ($r > 0.75$), prioritizing variables that represent climate extremes, climate variability, habitat variability, last glacial maximum (LGM) climate stability, and human influence on the environment. We prioritized these classes of variables because the influences of environmental extremes and variability across space and time are more likely to shape local communities than averages 150,151.”

Line 147: It is incorrect and misleading to call this “intraspecific genetic diversity” here and throughout. The textbook definition of intraspecific variation is variation among individuals of the same species.

Authors' response: *We believe the reviewer is referring to line 247, in which case there appears to be a misunderstanding. We are indeed quantifying intraspecific genetic diversity and summarizing it across sampled taxa within grid cells. Within each grid cell, we group sequences by their OTU and calculate average pairwise differences (GD) for each OTU and then summarize this using GDM and GDE. For example, the distribution of GD for an assemblage of 5 OTUs may look like this: $[GD_{OTU1} = 0.0023, GD_{OTU2} = 0.0014, GD_{OTU3} =$*

0.005, $GD_{OTU4} = 0.0022$, $GD_{OTU5} = 0.0045$]. Each of these GD values is “intraspecific genetic diversity”, which is what we are summarizing with the metrics GDM and GDE. Figure 1 b) demonstrates this graphically. If there was some confusion, we did not calculate the average pairwise differences across all sampled individuals per grid cell and call this “intraspecific genetic diversity”, which would be more akin to phylogenetic diversity. We hope this clarifies the issue.

Line 263: should state something like “which highly functional and is under strong selection in most species”

Authors’ response: *We agree that adding the functional nature of mtDNA is necessary and that it is under selection across most animal species. The revised text is (L376-381):*

“It is important to note that both metrics are calculated using mtDNA, which is a single, highly functional, non-recombining locus. Using mitochondrial DNA has many important advantages 12, yet relying on a single-locus marker has drawbacks. Notably, the genetic diversity that mtDNA captures is from a single draw from the stochastic coalescent process 63, and due to the lack of recombination, patterns of mitochondrial diversity can be impacted by selection and taxon-specific mtDNA mutation rates 52,61–64”

Line 275: I cannot see what average evenness and richness can tell us about single species ranges. No range information is included in the study. I suggest that all reference to relationships between GD and GDE and ranges be removed as speculative and outside of analyses and data.

Authors’ response: *We believe the reviewer is referring to our summaries of genetic diversity (GD). In this case, while range information is not included in the study, we are connecting our observed patterns (lower GDE in the tropics vs. the subtropics) with existing patterns in the literature, as well as theoretical expectations (genetic diversity increases with range size and latitude in animals, Pelletier & Carstens 2018 and Exposito-Alonso 2022, cited in the manuscript). We include cautionary language (“Rapoport’s Rule... **might partially explain this result...**”) because we acknowledge that range sizes are not the sole reason for mtDNA genetic diversity variation. In light of confusion about the relevance of GDE vs GDM for this pattern, we add additional reasoning about why we find a correlation with GDE and not GDM (L 273-277):*

“Furthermore, the inflation of GD in a larger portion of species within a grid-cell serves to increase evenness (GDE) by bridging the difference in GD between high GD and low GD species within the sampled assemblage. Although we do not find a correlation between latitude and GDM, we acknowledge a positive correlation between GDE and GDM that may contain some signal of an increase in GDM with latitude (Fig. 2d,e).”

Comment from the authors: *The below comments are from the reviewer 2 replying to our replies to their original (first round) comments. Their original comments are in standard formatting, our original replies are in green, their reply to our replies are bolded, and our replies to their new replies are italicized in the format “**Authors’ response:** reply text”.*

While I am supportive of and interested in the questions addressed in this paper and think

analyses of these data can certainly be made publishable, I cannot support the publication of the paper in its current format. My primary concern is that the data analyzed are inappropriate for addressing the questions posed. mtDNA are not neutral, do not recombine, are not well correlated with demography, are not correlated with neutral genome-wide diversity or adaptive potential (e.g., Allio et al 2017 Mol Biol Evol; Edwards and Bensch 2009 Mol Ecol; Galtier et al. 2009 Biol Lett; Galtier et al 2009 Mol Ecol). For example, in *Drosophila*, the mt genome plays an outsized role in clinal adaptation and is suspected to be heavily involved with climatic adaptation (see Quantifying the relative contributions of the X chromosome, autosomes, and mitochondrial genome to local adaptation Lasne et al. 2019 Evolution). Such data are unsuited for testing demographic hypotheses should be analyzed and interpreted as if they represent adaptive genetic diversity (see Yiming et al. 2021 Ecology). The firm conclusion of the Galtier et al 2009 ME review and references therein are that mtDNA cannot be used as a general population genetic measure of the diversity of anything but mitochondrial DNA.

Author response: While we acknowledge the reviewer's point about selection interacting with mtDNA, we think that this opinion is overstated in the context of a macrogenetics study of this scale. The stated shortcomings were currently acknowledged in the previous version of the manuscript, but they do not outweigh the practical advantages of being able to use mtDNA on such a large global scale and are not in line with evidence that mtDNA patterns do generally relate to important processes beyond "anything but mitochondrial DNA", (with a healthy dose of probabilistic uncertainty).

I am happy we agree on the critical issues of non neutral data, no recombination, little correlation with demography, no correlation with neutral genome-wide diversity or adaptive potential (both of which are critical for conservation and management. I disagree that the opinion is overstated. This is a significant list of major issues that the availability of data does not make go away. Indeed the aggregation methods used exacerbate these issues as described below. I also disagree that the issues were well addressed in this draft or the previous draft. I have read this draft a number of times, and while additional citations are given they are not well discussed. This is well established literature with strong consensus and many more references than I gave. I encourage the authors to fully engage with this literature.

Authors' response: The Edwards and Bensch 2009 paper was part of a vigorous debate about the merits of mtDNA data for the inference of detailed complex demographic histories used for phylogenetic/phylogeographic reconstruction used at that time. That particular debate has long been settled, but we are asking much less of this locus in calculated genetic diversities rather than reconstructing the detailed demographic history of each OTU. Rather, we argue that aggregating large numbers of OTUs per grid cell at large geographical scales reflects patterns of mtDNA that could relate to coarse-level processes that span demography, selection, and selectively neutral genetic mechanisms.

The Edwards and Bensch debate was in the context of phylogenetics but was fundamentally about whether mtDNA can be used to infer evolutionary processes. The consensus on both sides was that in many cases it was useful for describing pattern but it was not suitable for inferring process. The entire paper is framed around evolutionary processes and their relation to mtDNA diversity. The process vs patterns consensus clearly holds for pop gen and the current study. Regarding averaging sequence from many species mtDNA and nDNA evolution are uncorrelated (as we

agreed above) so mtDNA can only be used to draw inferences about mt. Second the level of evolution is the population (both adaptively and neutrally). No matter grid cell size variation the average diversity and evenness of hundreds of species cannot reflect population level processes. If these coarse level process (demography, neutral and adaptive evolution) are the focus of the paper and the authors think their data reflects them it authors must explain how with support. I cannot see an easy way to evolutionary interpretability.

Authors' response: One of the coauthors of our manuscript had numerous correspondences with the lead author Scott Edwards about the Edwards and Bensch 2009 paper, and as stated in the first paragraph of that paper, the topic was about inferring "the phylogeographic history of avian species". The word "phylogenetic" is not in the paper. The paper does focus on phylogeographic processes, and in 2009, Edwards takes the correct side that using nDNA from many loci is superior. We agree, but Edwards and Bensch are not really taking the extremist position that mtDNA has zero information content about these processes. After all, mtDNA patterns often reflect important features of demographic history despite selection (e.g. humans, *Drosophila* etc.). While lack of statistical correlation between nDNA and mtDNA across taxa is an important thing to consider, this by itself does not demonstrate that there is zero correlation between mtDNA patterns and phylogeographic processes.

There is a response to Edwards and Bensch 2009 in the same issue (Barrowclough and Zink 2009 <https://doi.org/10.1111/j.1365-294X.2009.04271.x>). This response shows that rather than having zero information content about demographic history, even with all the acknowledged shortcomings, mtDNA divergence patterns tend to be leading indicators of population divergence even though ultimately one would obtain divergence time estimates with narrower confidence intervals given data from large number of nuclear loci (as predicted by theory). Indeed, despite the know shortcomings, mtDNA can help detect population divergence, which could be one of the many processes underlying GDM and GDE (e.g. if there is some divergence and isolation within an OTU that is within a grid cell, GD could become inflated for this OTU).

For a good example of connecting intraspecific mtDNA diversity to coarse-level population histories, a very recent macrogenetic study of ~15,000 European butterfly mtDNA sequences from ~300 species (Dapporto et al. 2019, <https://doi.org/10.1111/1755-0998.13059>) found a pattern of depleted diversity in previously glaciated areas, a classic prediction of founder effect expansion after the last glacial maximum (i.e., the classic idea championed by the late Godfrey Hewitt). Rather than this observation only being related to the mitochondrial DNA of butterflies as the reviewer here suggests, it is more likely that the community history of post-Pleistocene expansion had a detectable impact on mtDNA diversity patterns. One could counter that we are merely seeing the effect of linked positive selection occurring in the previously glaciated areas, yet these two processes (demographic founder effect and positive linked selection) are not mutually exclusive and are both worthy of discussion.

I am aware of this paper. I agree that there are interesting spatial patterns in the data. However, the authors analyzed species separately (I believe) and did not average values across species. This paper does not support the authors approach.

The authors first prediction is:

"First, since high selection, absence of recombination, erratic mutation rate and stochastic

variation in population size make overall mtDNA divergence highly unpredictable (Allio et al., 2017; Galtier et al., 2009; Nabholz et al., 2009; Romiguier et al., 2014), we expect to find no correlations between mtDNA diversity (haplotype diversity) and species traits related to population size, dispersal capability, number of generations and climatic tolerance (prediction 1).”

As I've hoped to communicate my concerns are well known and established and clearly expressed in this paper.

Further the authors found:

“Despite the theoretical expectation that levels of neutral genetic variation should increase with effective population size, no relationships between range size (a probably good proxy for population size) and mitochondrial DNA diversity has emerged in our extensive data set”

"Moreover, the fall of the neutrality assumption for mtDNA implies the importance of mtDNA variation in influencing functional traits. Recent evidence proved that these mitochondrial-derived traits are involved, among others, in local adaptation to climatic conditions (Toews et al., 2014)."

This paper found spatial variation in their data.

Authors' response: We thank the reviewer for these quotes. Later in the paper, the authors highlight the important paradox that could explain some of the results in our study and do indeed directly relate to neutral demographic history. Specifically, range size would not at all correlate with GD in areas with extreme changes in range due to late Pleistocene glacial cycling. Furthermore, despite the distortions caused by selection at mtDNA functional loci, the authors do find a footprint of neutral history in the butterfly data related to population changes since the LGM (as is found in many mtDNA-based phylogeographic datasets).

Another example of mtDNA data reflecting important features of evolutionary history is from humans. While mtDNA data from humans are unable to detect such things and admixture from human/Neanderthal interbreeding, the genetic diversity and divergence patterns of human mtDNA have always reflected the main features of neutral demographic history such as origins in Africa with subsequent expansion into the rest of the planet, something that has largely been corroborated with whole genome data.

I apologize if I am missing something here. I do not see the relevance of this for the current study. They almost certainly used haplotype diversity and not nucleotide diversity (as in the above reference) and this is a very different question in a single species. As noted above mtDNA can inform pattern, sometimes, for some species, in some situations. This work is about spatial variation and not conservation, biodiversity etc.

Authors' response: We believe the study is relevant because it is an instance of mtDNA reflecting a pattern consistent with a demographic process supported by other sources of

data. Our study is also about spatial variation (summarized across species) and is interpreted in the light of frameworks where this variation can be informative (conservation, biodiversity, etc.). Additionally, nucleotide diversity and haplotype diversity are different, yet comparable approaches to estimate the parameter of interest- genetic diversity.

While we strongly acknowledge the study by Galtier et al. 2009, observed mtDNA patterns reflect combined demographic and selection processes in addition to a large stochastic component (i.e., the coalescent). This is consistent with the general observation of a large amount of noise in macrogenetic studies (not to mention variation in mutation rates and generation times etc).

I disagree that the Galtier paper is strongly acknowledged. It is cited only twice and only in the context of mtDNA sequences being a single locus. This critique was much more substantial with a whole section entitled “the worst marker ever”. The opening 2 paragraphs and the closing 2 paragraphs do not seem to me to acknowledge Galtier’s critique.

Authors’ response: *We agree that consideration for the Galtier paper and problems with mtDNA in general deserved more space, so we modified the Introduction and Discussion accordingly. In response to other review comments in this same vein, we dedicate two full paragraphs in the Discussion to the potential pitfalls of using mtDNA, while also justifying our choice of using mtDNA in the present study (L376-400):*

“It is important to note that both metrics are calculated using mtDNA, which is a single, highly functional, non-recombining locus. Using mitochondrial DNA has many important advantages 12, yet relying on a single-locus marker has drawbacks. Notably, the genetic diversity that mtDNA captures is from a single draw from the stochastic coalescent process 63, and due to the lack of recombination, patterns of mitochondrial diversity can be impacted by selection and taxon-specific mtDNA mutation rates 52,61–64. However, the pattern of genetic diversity at any single locus is ultimately impacted by multiple processes (phylogeographic demographic history, mutation rate, and linked selection) rather than any one process alone 125. Moreover, despite the likelihood for linked selection and high coalescent variance to distort mtDNA diversity patterns, mtDNA data routinely retain key features of species-specific demographic and phylogeographic histories 126–129. Lastly, emergent patterns of mitochondrial genetic diversity across assemblages demonstrably recapitulate well-known biodiversity patterns like the latitudinal biodiversity gradient, indicating meaningful biological signal 21,23,26.

While intraspecific mtDNA diversity has been shown to have a weak, insignificant correlation with whole genome genetic diversity in a group of 38 European butterfly species 130 and is generally insufficient for making detailed inference of demographic history or phylogenetic reconstruction 62, sampling the mitochondrial genetic diversity of thousands of taxa per locale with rapidly increasing data availability is a promising first step towards understanding an important component of the biodiversity of assemblages at global spatial scales 10,12,65,66, and will be especially valuable until whole-genome data become more readily available through projects like the Earth BioGenome Project (Lewin et al. 2018) and GEOME (Riginos et al. 2020). While taking these important cautions into consideration, we view macrogenetics as a developing field and thus basic expectations regarding observed patterns are still not established. Our study is a step in the direction of establishing this foundational knowledge and suggests avenues for ways to test specific hypotheses.”

We additionally modify multiple sentences in the Introduction to express caution in our interpretation of observed patterns and emphasize the evolutionary forces impacting mtDNA diversity:

(L78-80) “While being mindful of these limits in our study design and data 14,15,19,52,53, we present the first global class III macrogenetic analysis of insect intraspecific mitochondrial DNA patterns.”

(L85-89) “Therefore, it is important to note that while mtDNA has useful properties¹², mtDNA diversity patterns are impacted by multiple processes that distort the patterns shaped by phylogeographic demographic histories, including selection, high coalescent variance, and mutation rate variation across taxa^{52,61–64}. While taking these complex dynamics into consideration...”

The other papers the reviewer cites are interesting studies that we now cite: “These single-locus markers have drawbacks: notably, the genetic diversity that mtDNA captures is from a single draw from the stochastic coalescent process operating within a lineage and may be subject to selection in addition to neutral demographic processes 62–67.” Allio et al. 2017 shows that mtDNA rates are variable and faster than nuclear DNA rates, which in itself does not contradict any specific inferences or conclusions we make.

I disagree. You average across species with very different mtDNA mutation rates (sometimes as much as 100 fold variation). This variability directly means these quantities should not be averaged. They are also uncorrelated with nDNA evolution. The therefore cannot be used to make the same inferences that neutral nDNA can make.

Authors’ response: *We have restricted our interpretation of the data to only include mitochondrial genetic data, rather than make broad statements that include nDNA. We specified “mtDNA” or “mtDNA diversity” or “mitochondrial genetic diversity” at appropriate sections throughout the manuscript.*

However, variable rates could have impacted the observed patterns and we now cite this paper (see above) to highlight this potential confounder. The results of Lasne et al. 2019 are only based on a single taxon and therefore should not be used to make generalizations across all insects. The study of Yiming et al. is especially interesting, but the data is MHC, a locus that is thought to be impacted by balancing selection (which would not specifically impact haploid mtDNA).

Lasne was just an example and I hope the authors will search further. For example please see also Hurst and Jiggins Proc B

“We review here the evidence for indirect selection on mtDNA in arthropods arising from linkage disequilibrium with maternally inherited symbionts. **We note first that these symbionts are very common in arthropods** and then review studies that reveal the extent to which they shape mtDNA evolution. **mtDNA diversity patterns are compatible with neutral expectations for an uninfected population in only 2 of 19 cases.** ... We therefore conclude that **these elements often confound the inference of an organism’s evolutionary history from mtDNA data and that mtDNA on its own is an unsuitable**

marker for the study of recent historical events in arthropods.”

Authors’ response: *This paper came out in 2005, again, in the era when there was a lot of enthusiasm for championing the soon to explode nDNA datasets with the growth of next generation sequencing. However, the authors also admit that their results are preliminary:*

“This judgment is preliminary because a full test of this issue requires testing the congruence of mtDNA and nuclear DNA patterns of variability across a range of clades, where symbiont presence has not been ascertained and not (as in our reviewed studies) been used as a proband to seek the effects. We do, however, regard the above studies combined with the known high incidence of Wolbachia as an evidential ‘smoking gun’ that urges precaution in the first instant, pending formal testing.”

This issue was reexamined in mtDNA barcode data (Smith et al 2012; <https://doi.org/10.1371/journal.pone.0036514>). Smith et al found that across > 2 million insect entries to BOLD, Wolbachia was only detected in < 1% of the cases, and that:

“We conclude that the presence of the Wolbachia DNA in total genomic extracts made from insects is unlikely to compromise the accuracy of the DNA barcode library; ...”

Text from the manuscript: “In contrast, species richness and GD may be decoupled due to the “cradle” hypothesis that predicts higher speciation rates, more population structure, and smaller range sizes leading to Rapoport’s Rule, the tendency for species’ range sizes to increase with

increasing latitude. Second, we might predict that the influence of Late Quaternary climate fluctuations to have an impact on GD through population demographic processes influenced by cyclical variation in precipitation, temperature, and glaciation patterns where areas with more stable climatic histories are predicted to have increased GD. Finally, we consider the influence of human disturbance on patterns of assemblage-wide GD, which we expect to decrease in magnitude in areas of high human influence.”

Each of these hypotheses acts through demographic processes acting on populations. mtDNA is not suited to test them.

Author response: We agree that mtDNA is ill-suited for detailed inference of complex demographic and/or phylogenetic history. As laid out in our previous response (and examples) above, basic coarse-level demographic processes are completely decoupled from mtDNA diversity patterns.

I suspect that the final sentence is a typo, but I agree  . However the authors have not, at least to my satisfaction, demonstrated that arbitrarily sampled sequences averaged across species disregarding populations can say much if anything about evolutionary processes, coarse or not. Much of the writing does not suggest coarseness or tentativeness in with conclusions.

Authors’ response: *The final sentence was indeed a typo— our apologies. We added caveats and tentative language to emphasize that our findings, while limited by the use of single-locus mtDNA data, still convey some information about biogeographic processes.*

The Galtier review is particularly compelling and comprehensive and those criticisms of using

mtDNA as a measure of general genetic diversity have, to my knowledge, only grown stronger. If the authors disagree with this substantial literature and feel pragmatism is warranted those criticisms need to be directly confronted in text if the reader is to be able to evaluate the paper on its own. The strong conclusion of the paper can be summed up in this quote: "For all these reasons, mtDNA is perhaps intrinsically the worst population genetic and phylogenetic molecular marker we can think of."

Author response: We thank the reviewer for pointing out that we should clarify the limitations of connecting mtDNA with demographic history. We acknowledge that it is a noisy relationship, yet not an absolutely empty one"

"These single-locus markers have drawbacks: notably, the genetic diversity that mtDNA captures is from a single draw from the stochastic coalescent process operating within a lineage and may be subject to selection in addition to neutral demographic processes 62–67."

"While intraspecific mtDNA diversity data is insufficient for making detailed inference of demographic history or phylogenetic reconstruction 67, we instead treat it as a variable to study on the macroscale, rather than a population genetic marker. Macrogenetics is a relatively new field and thus basic patterns are still not established. Our study is a step in this direction and suggests avenues for ways to test specific hypotheses."

I admit that do not understand what the authors are trying to say here. The manuscript until this point has been fully about population genetics. I can't imagine a scenario where a paper about nucleotide diversity and spatial distribution of a population genetic marker isn't about population genetics. Given the agreed upon significant weaknesses of the marker the authors need to make the case that they can get a coarse patterns in aggregate rather than assert that they can. I detail specific problems with their metric below.

Authors' response: We agree that the statement "rather than a population genetic marker" might be misunderstood and is out of place given that genetic polymorphism is under the larger umbrella of population genetics. We removed the statement to emphasize that we mean we don't intend mtDNA to convey precise information about complex demographic and selective processes (i.e. a population genetic marker in the traditional sense).

The authors of this manuscript correctly point out that their approach follows the methods of previous studies, some of which were published in this journal (e.g., Theodoridis et al. 2020). These papers too, to varying degrees, did not address the substantial shortcomings of using mtDNA to estimate genetic diversity. Galtier also predicted the continued use of mtDNA as a population genetic marker:

"Despite the many concerns we express about the use of the mitochondrial marker in molecular ecology, we are confident that scientists will keep on analysing mtDNA variation in the wild. The reason why we think mtDNA will keep its popularity is its practical convenience".

Author response: The paper by Galtier et al. made some important findings, but was specifically referring to the use of mtDNA for single-species population genetic studies and/or phylogenetic studies. In contrast, we are not using mtDNA as a population genetic marker in the traditional way Galtier et al. are referring to. We are using available mtDNA for a macrogenetic study aggregating intraspecific diversity patterns from large numbers of co-distributed OTUs and correlating this with a suite of environmental variables. Using these observed correlations to infer eco-evolutionary and demographic processes is beyond the direct scope of our study or any other macrogenetic study to date (with the possible exception

of Exposito-Alonso et al. 2022, DOI: 10.1126/science.abn5642). Instead, we are using the results to discuss possible explanations of these correlations (i.e., we are not inferring them with a process explicit model ; see DOI: 10.1126/sciadv.abj2271 for review). Therefore, our study is a valuable and important first step that generates hypotheses that will inspire and lead to future studies.

I think the major point of disagreement has not been well communicated. mtDNA cannot capture demography etc at the within species level as we clearly agree. The authors assert that taking the average of a number across species that does not do the job they wish it would in a single species is somehow better or different. My contention is that averaging does not generate the ability for mtDNA capture these processes even in a coarse way and the authors have not cited data to support their contention that it does. These are population level process. Given millions of years of independent evolution mtDNA with very different mutation rates I think it is clear that averaging only makes it worse. The authors do not present the context of their findings as ‘possible explanation’ (see for example the opening two paragraphs and the closing two paragraphs)

Authors’ response: We think despite shortcomings, there tends to be some positive information about coarse-level demographic history from mtDNA (beyond the biology of mtDNA), which we outline in other responses.

Despite my harsh criticisms of mtDNA work, I do appreciate there is value in exploring global patterns of mtDNA diversity. I suspect that there are many interesting processes related to the biology of mtDNA underlying some of the patterns detected. However, the sentiment the impressive volumes of data for mtDNA allow these markers to overcome their shortcomings is, to the best of our knowledge, not well supported by data. If mtDNA cannot measure the quantities of interest increasing the volume of data will not help (Paz-Vinas et al 2021 Ecol Let; Galtier et al 2009 ME).

Author response: We are glad that the reviewer found value in our study that uncovers statistical correlations between mtDNA GD and environmental variables. Following the logic of Pilowski et al. 2022, our phenomenological model finds some unexpected correlations that can suggest the mechanisms that produce them. Actual causation (be that of regional demographic histories or histories of selection) are points of discussion and ultimately hypotheses that are testable by process-based modeling in future studies, not points of direct inference or conclusion.

Again the authors very generally do not present their findings as tentative.

Authors’ response: Many of our findings were originally presented as tentative, but to make this more clear, we added additional language that emphasized the tentative nature of our study throughout the manuscript. We’ve sampled a few examples below:

(L237-243): “Although our approach is correlative and does not link predictor variables with observations as one could from a process-explicit model ⁷⁵, these results suggest that forces

underlying intraspecific mitochondrial genetic diversity could be inherently different from those driving the classical negative latitudinal gradients in species richness and phylogenetic diversity found in most arthropod taxa, including ants, butterflies, and spiders^{76,77 78–80}, as well as plants⁸¹, which are expected to be strongly linked to insect biogeographic patterns”

(L264-266): “Rapoport’s Rule, the tendency for species’ range sizes to increase with increasing latitude^{89,90}, might partially explain this result if species with larger ranges tend to harbor greater genetic diversity^{91,92}”

(L297-301): “However, a deflation of GD in temperate regions could also be driven by widespread positive selective sweeps on the mtDNA genome across insect species that expanded into these previously glaciated areas after the LGM, a scenario that is non-mutually exclusive to founder-effect dynamics⁹⁹.”

Putting aside the appropriateness of this measure of mDNA as a population genetic marker there are also significant methodological issues with this work. Again the authors have followed previous studies but these studies share the same flaws. Schmidt and Garroway 2021 (Con Gen) note that mean genetic diversity of arbitrary numbers of species aggregated from within arbitrary sized grid cells is not a meaningful population genetic or conservation relevant quantity. This and the previous studies the authors followed take the mean genetic diversity of across all species sampled per cell regardless of the fact that the species to be averaged will have extremely different life and evolutionary histories, which will produce different equilibrium genetic diversities, and the fact that mtDNA mutation rates can vary extensively.

Author response: We appreciate the argument from Schmidt and Garroway 2021, but note that we do at least restrict our study to insects and that our second summary statistic is actually a way to measure the variance in GD that can come from variation in life and evolutionary histories (GDE).

I’m afraid this response does not help. I can think of no instance where averaging any sort of genetic diversity across species would be appropriate. No study of genetic diversity using SNPs or microsats would average any genetic metric across species. I cannot see such a paper getting out to review. As noted below and by others there is an approach that would enable not averaging. The authors could treat species as a random effect and allow slopes and intercepts to vary. This is easy in the brms package that they use. You could then calculate the relationship between genetic diversity of each species and any other quantity and ask whether there are overall consistent general relationships. This avoids averaging and because relationships are calculated within species things like variable diversity, histories, and mutation rates would not matter.

Authors’ response: Although it would be less straightforward to average a GD measure across species for microsats or genome-wide SNP data for various reasons, it has been done and published (Singhal et al. 2017; <https://doi.org/10.1098/rspb.2016.2588>). We would like to clarify our approach, repeated from an earlier response. We do not average pairwise nucleotide differences across all species within a grid cell, but we average pairwise differences within

each OTU (GD) found in a grid cell and then summarize those averages across species using GDM and GDE. We've copied an excerpt from the earlier response below for reference:

“Within each grid cell, we group sequences by their OTU and calculate average pairwise differences (GD) for each OTU and then summarize this using GDM and GDE. For example, the distribution of GD for an assemblage of 5 OTUs may look like this: [$GD_{OTU1} = 0.0023$, $GD_{OTU2} = 0.0014$, $GD_{OTU3} = 0.005$, $GD_{OTU4} = 0.0022$, $GD_{OTU5} = 0.0045$]. Each of these GD values is “intraspecific genetic diversity”, which is what we are summarizing with the metrics GDM and GDE. Figure 1 b) demonstrates this graphically. Perhaps the reviewer thought that we calculated the average pairwise differences across all sampled individuals per grid cell and called this “intraspecific genetic diversity”. We did not do this, and if we did it would have been some more akin to phylogenetic diversity.”

In case the reviewer is correctly interpreting our method of aggregation, the alternative approach is impossible given the distribution of sampled insect OTUs. Almost all of the OTUs are only found in one grid cell (~85%) which means that fitting any kind of random effect will not be possible. The reason is that mixed effects models need at least two observations to have a real number for the estimate of variance. One really would need at least three observations to have any real meaningful estimate of variance (and you need somewhere between 4 and 6 observations per random effect to meet statistical best practices).

Additionally this method of data aggregation means that different numbers of species and indeed different species are being compared. Millette et al grouped mtDNA sequences by species within grid cell and I suggest the authors explore this method. It too is not ideal but here nucleotide diversity is interpretable. The authors are careful to call their metric mean genetic diversity until the discussion where they slip into making inferences about intraspecific variation. Again the previous studies call mean genetic diversity intraspecific genetic diversity too. I cannot see how the mean of multiple species gives information about intraspecific variation. The Millette approach is a measure of intraspecific mtDNA genetic diversity—this does not alleviate concerns about the inappropriateness of mtDNA as a population genetic marker.

Author response: Our GDM and GDE are both derived from the classic average pairwise distances (per OTU), which is sometimes referred to as nucleotide diversity and is essentially a useful measure of intraspecific or intrapopulation or intra-location per species variation in genetic diversity (depending on what unit one chooses to sample). GDM is the average of this per grid cell and GDE is the evenness. It has desirable and robust sampling properties (as opposed to allelic richness). We clarified this by saying “mean intraspecific variation” instead of just saying “intraspecific variation” alone.

Again the authors appeal to previous studies with little argumentative back up. These GDM and GDE metrics are not classic having only been used in the handful for recent macrogenetic studies.

Authors' response: I think our language may not have been clear- we are saying that average pairwise distances within an OTU is a classic measure, and GDM and GDE are summaries of that classic measure. If one is interested in observing how the average and variance of intraspecific genetic diversity varies across space, then GDM and GDE would be a

way to observe this. Additionally, our sequences are grouped by species within each grid cell and we agree that nucleotide diversity is interpretable with this approach.

I appreciate the intention behind using genetic diversity and evenness to partition diversity patterns into different components that together are more informative than a single metric. This is a widely used method in ecology to attribute biodiversity differences due to changes in total abundance across an assemblage and changes in the species abundance distribution.

In population genetics we take similar approaches comparing evenness metrics (e.g., expected heterozygosity) to richness metrics (allelic richness) to infer, for example, population size changes.

However, the genetic diversity mean and evenness metrics used here are not completely analogous to their use in macroecology and in my view they don't measure what they purport to measure.

Authors' response: GDE is the same as the first Hill number for GDs across a sample of species rather than the typically used abundances. This has been developed for GD in a couple of publications led by one of our coauthors, and among many things, has used simulations to show joint predictions of Hill1 for GD and abundances across a number of community assembly models (Overcast et al. 2019, <https://doi.org/10.1111/jbi.13541>; <https://doi.org/10.1111/1755-0998.13514>).

We draw our inspiration for GDE from (Chao et al 2014) who developed a system of equations based on Hill numbers that allow for direct comparability in quantifying diversity across different domains of community data (e.g. species diversity, functional diversity, taxonomic diversity). The core idea is that in all cases species have properties (abundance, traits, phylogenetic commonness/rarity) and we want some measure of 'diversity' for the distribution of these properties within communities that is comparable. That is what they did, and what we did (in previous work; see Overcast et al 2019; Overcast et al 2021) is only to see that genetic diversity is another property of a species (which Chao et al did not consider) that could be reasonably treated in exactly the same way. Jost (2010) had previously shown that a useful measure of species abundance evenness (within the framework of Hill numbers) can be obtained by dividing Hill numbers of order q by species richness (S ; see Section 3 in Jost), and so what we did is the exact analog, dividing the first Hill number of genetic diversity by S to obtain GDE.

On the other hand, we would argue that 'genetic diversity mean' (GDM) has no meaningful analog in macroecology (abundance mean?), yet we include this in our analysis for two reasons: 1) to aid in comparability to previous class III macrogenetic studies; and 2) to show that previous macrogenetic studies are not capturing an important component of community-scale diversity (GDE).

I am open to being corrected, but my reasoning for this view is below:

L97-101: Existing macrogenetic studies, which are inherently correlative, have only described average intraspecific genetic diversity (GDM), and are therefore unable to determine whether high average genetic diversities are due to high diversity within most community members or to the effects of a few taxa with extremely high diversity (Fig. 1; see Methods)."

As the authors note, because the mean cannot distinguish a high value from many species having relatively high GD or from few species having extremely high GD, it therefore explicitly does not measure intraspecific genetic diversity. It cannot be used to test questions related to patterns of intraspecific genetic diversity. It is also uninformative about evolutionary processes acting across species because similar patterns across cells can arise for completely different reasons. A cell might have high diversity because there are several species whose abundances (and therefore genetic diversity) increase with temperature, whereas in a neighboring cell genetic diversity might increase because one species originated in this region and has extremely high diversity there relative to other species.

***Authors' response:** GDM and GDE are summaries of the distribution of intraspecific genetic diversities across OTUs. We highlight how GDE complements GDM, and interpret how these complimentary summaries may tell us something about insect biogeography and community history. They can be a useful first step to understand factors underlying patterns of intraspecific mtDNA genetic diversity because we are correlating patterns of environmental variables with patterns of intraspecific mtDNA genetic diversity, where the patterns of intraspecific mtDNA genetic diversity are the summarized by GDE and GDM. While the specific scenario you mention may be true in some cases, this is one of many scenarios that highlight the challenges of identifiability and noise when interpreting the relationship we observe, (which we acknowledge). However, it is not impossible for some meaningful signal to be present, (such as the deflated values of GD in areas recently glaciated), although we make room for the possibility of the non-mutually exclusive scenario where wide-spread positive selective sweeps on the mtDNA occurred across many temperate latitude insects that expanded from refugia after the LGM (as was made by Ilves et al 2010; <https://doi.org/10.1111/j.1365-294X.2010.04790.x>) (L297-301):*

"However, a deflation of GD in temperate regions could also be driven by widespread positive selective sweeps on the mtDNA genome across insect species that expanded into these previously glaciated areas after the LGM, a scenario that is non-mutually exclusive to founder-effect dynamics⁹⁹."

We agree that the field of macroecology and population genetics are inherently challenged by identifiability of model parameters (e.g. N_e or μ can never be known from observed estimates of Θ ($4N_e\mu$), which we find and interpret accordingly.

Regarding the mean as an assemblage level summary: The distribution of genetic diversity is right skewed (lines 464-466: "The distribution of GD values within an area of low GDE can take a variety of shapes, but the most common in our observed data is markedly L-shaped whereby most OTUs have low or zero GD along with a small

number of OTUs with large GD (Fig. 1d).”

The mean is not an appropriate summary statistic for skewed distributions. The authors nevertheless use the mean and use this fact as a rationale for using evenness.

Authors’ response: To keep our results in line with the literature, we used a square root transformation of the mean, which corrects for the underlying skewed distribution of the data.

Coming to evenness: this is estimated taking the Hill number approach, where evenness is the Shannon index divided by the number of OTUs for a cell (line 453). This metric mathematically depends on the number of sampled species in the cell. The authors have done sensitivity analyses based on different minimum numbers of OTUs (between 10-200 OTUs/cell), but these tests cannot test sensitivity for evenness because evenness necessarily decreases with increasing numbers of OTUs in a cell. And, there is still a lot of wiggle room for sample size effects if the maximum sample size is over 1000 OTUs with a 200 OTU minimum. It therefore makes absolute sense to me that evenness is significantly correlated with latitude (lower at higher absolute latitude) and with freezing (higher evenness in areas that rarely freeze). These are where more taxa are sampled. I would be interested in seeing whether these relationships disappear if the number of taxa is modelled as another covariate to control for species sample size.

Authors’ response: Our approach for calculating GDE works by dividing the exponential of Shannon’s entropy by the number of OTUs (i.e. “species”) for a cell, which controls for species richness and is a major reason why we use GDE. Additionally, we confirm this expectation in Supplementary Fig. 16 and Supplementary Table 5, showing that the number of OTUs in a cell does not significantly correlate with GDE or GDM. Given these pieces of information, we decided to leave this variable out of our model.

Besides these issues with metrics, the authors’ statements in the rebuttal do not match the revised manuscript. The authors say that they do not test hypotheses, but generate them. This contradicts the clearly worded hypotheses to be tested on lines 104-119, and statements made throughout the manuscript (e.g. L289-290: “We are hypothesizing that both larger ranges and climatic stability could have a positive effect on GD, yet they are different processes and do not necessarily co-occur”). Again, neither of the demographic hypotheses proposed here can be appropriately tested with only mitochondrial data.

Authors’ response: The referred to phrase is a hypothesis generated from the study rather than an a priori hypothesis, but we agree that the language might be misleading as the word “hypotheses” has a range of meanings. We updated our language to more appropriately read (L278-279):

“If both larger ranges and climatic stability have a positive effect on GD, they may not co-occur.”

Additionally, we felt that the biogeographic hypotheses presented in the Introduction contribute to the misunderstanding that we are directly testing relationships with range sizes, species richness, etc., so we moved all discussion of them to the Discussion and rewrote this section of the Introduction to reflect the correlative nature of the study (L109-118):

“After calculating these two metrics across all OTUs for each cell, we explore how they correlate with global environmental and geographic variables. If GDM and GDE follow classical global biodiversity trends, we expect GDM and GDE to decrease with latitude 69 and human disturbance 70, and increase with climate stability 71. Once these environmental correlates of intraspecific insect mitochondrial genetic diversity are found based on information extracted from insect sampling efforts across the globe, we use them to predict patterns of insect GDM and GDE in undersampled regions, which includes most of the planet. We discuss and interpret our results in the context of classic biogeographic patterns such as the latitudinal diversity gradients in insect species richness, average and variability in range sizes, and late Pleistocene climate cycles.”

Pas-Vinas et al note that arbitrarily aggregating sequences across large spatial areas create implausible populations. This is true in the extreme. Populations are not defined genetically and contain multiple species. Most inferences in this paper refer to population level processes (evolutionary change, relationships to environments, drift etc).

Author response: We agree that summarizing genetic variation across OTUs within arbitrary grid cells is a compromise out of pragmatism. We agree with the reviewer’s point brought up by Pas-Vinas et al is an interesting dynamic to consider for macrogenetic studies. Macrogenetic studies are inherently different from single species population genetic studies with different goals and approaches but there is a shared inherent abstraction of what a “population” is in relation to a geographically defined sample. They are rarely congruent, yet methods based on population genetic theory can still yield useful results despite that populations are rarely homogenous or well defined (Barton et al., 2019; <https://doi.org/10.7554/eLife.45380>). Following this interesting point suggested by the reviewer, we now quantify and report that the vast majority (~75%) of the sampled OTUs are only found in single grid-cells (Supp. Fig. 11). Coalescent theory with migration (i.e., the structured coalescent; Nordborg and Krone 2001 and Nordborg 1997) predicts that local sampling can capture variation associated with unsampled areas and that this can lead to higher average pairwise distances (i.e., higher GD). We don’t see this as a bug, but as a feature that enables us to potentially capture information about species with larger and/or more isolated ranges. In relation to this important dynamic brought up by the reviewer, we now cite a recent macrogenetics paper that directly looks at links between range size and GD empirically and theoretically (Alonso-Espisito et al. 2022).

This helps for sure. Thank you.

Paz-Vinas also showed that sequence deposition is often incomplete and idiosyncratic authors tending to deposit only new sequences in a region. This too will significantly affect their measures of genetic diversity.

Author response: We are well aware that authors (and participants in the BOLD initiative)

neglect to deposit duplicate allele copies, and this is one reason why we chose our metric of GD to be average pairwise distances. This metric has several desirable statistical sampling properties (Tajima 1983) including approaching its expectation with only 5-10 samples. Therefore, missing duplicate samples, while positively biasing this summary metric will likely have a negligible effect. As mentioned to reviewer #1, we explore and verify this explicitly with simulations given a range of sample sizes and have included it in the new version of our manuscript, in the Supplementary Materials, and main text:

“To explore this sampling dynamic explicitly, we conducted coalescent simulation experiments comparing how the calculation of GD varies given identical samples with and without duplicate alleles removed. These simulations showed that retaining only unique haplotypes resulted in a small, consistently upward bias in GD values. Additionally, increasing values of effective population size (N_e) decreased this bias, with estimates of GD with and without duplicate alleles converging for N_e values greater than $\sim 10e5$ (Supplementary Materials, Supplementary Fig. 15)”

This also helps. Thank you.

The predictive maps look good. However, I am very leery about interpolating the globe from just 187 grid cells. The posterior checks look good but that just means the model can predict the data used to generate it, which is important! But predicting different continents with different species groups and geological histories with such a small sample seems fraught.

Author response: We have new results based on a more clear modeling procedure as well as a range of minimum threshold for number of OTUs power per grid cell. As with the original manuscript, we still do not extrapolate to environments beyond what are used to train the model. In addition, we added a model validation step to our modeling process, where we train the model and predict it to test data that was withheld. The data are stratified according to continent to enforce the most even spatial sampling possible. We highlight potential spatial biases in prediction by coloring observed versus predicted plots by continent and find no obvious spatial patterns in the residuals (Supp. Fig. 3 & 6). We do agree that our study has a biased focus on North America and Europe, and we now mention this in the Discussion: “Sampling biases, including the overrepresented North American and European sampling we find, may also influence this relationship, although we find no evidence of an influence in our data (Supplementary Fig. 14, Supplementary Table 5).” “Additionally, we find that OTUs occupy more grid cells between 40° and 60° latitude (Supplementary Fig. 14), matching our expectation for larger ranges at higher latitudes, although we acknowledge that sampling bias may contribute to this pattern as well.”

Great, thank you. This also helps. Sample size remains low for global prediction but this check is useful.

Finally, I am concerned with the conservation claims made in this paper given the use of mtDNA. Claims from previous papers (e.g., Theo) are already being misapplied. Their mtDNA is now included in UN Biodiversity planning despite the fact that it is not useful for

conservation. In conservation we are interested conserving adaptive potential. Neutral genome-wide genetic diversity is not well correlated with adaptive potential and organellar genetic diversity is not correlated with either. Strong conservation claims made from analyses of this marker are thus dangerous.

Author response: We agree that conservation claims should be made with caution and therefore we do not make any strong claims regarding how our study should directly inform conservation decisions. While anything gained from our study should be done with care, macrogenetic studies like ours are a potential first step to help inform conservation goals (along with other information), which is why we cited DeWoody et al. 2021 and and Reed & Frankham 2003. Likewise, studies like ours could aid in monitoring efforts.

I disagree that studies like this should aid in monitoring. That in fact is my explicit concern. They should not. mtDNA is simply not correlated with adaptive genetic diversity, genome wide diversity, or demography in a way that is useful for conservation. If the authors disagree I request that they argue their point with references. This is clearly laid out in the references provided.

Author response: We now point to these mtDNA specific references in the main text of the paper (Hobern et al. 2021; Santini et al. 2021; Petit-Marty et al. 2021) and a counterpoint (Schmidt et al. 2023), L(50-54).

It is also incorrect to state that the authors do not make strong conservation claims. There are uncaveated claims throughout.

For example the below claims are provided as context for their results:

Line 369: Genetic diversity is critical to the survival of insects and their complex interactions with other organisms. High genetic diversity may facilitate adaptation to changing climates, 123–125 emerging diseases, and pollutants: three (of many) potential drivers of the “insect apocalypse” 50. In addition, genetic diversity contributes to the diversity and stability of species interaction networks by affecting niche space and competition 126, community structure 127, and network complexity 128. At larger ecological scales, insect genetic diversity may reflect ecosystem function and structure as reliably as other traditional macroecological metrics such as species richness 129. It can also augment the resilience of ecosystems that provide continuing services for humankind 14, such as disease management, curbing the spread of invasive plants, aiding sustainable agriculture, pollinating food crops, and controlling pests 13.

Author response: We have added (L389-400):

“While intraspecific mtDNA diversity has been shown to have a weak, insignificant correlation with whole genome genetic diversity in a group of 38 European butterfly species 130 and is generally insufficient for making detailed inference of demographic history or phylogenetic reconstruction 62, sampling the mitochondrial genetic diversity of thousands of taxa per locale with rapidly increasing data availability is a promising first step towards understanding an important component of the biodiversity of assemblages at global spatial scales 10,12,65,66,

and will be especially valuable until whole-genome data become more readily available through projects like the Earth BioGenome Project (Lewin et al. 2018) and GEOME (Riginos et al. 2020). While taking these important cautions into consideration, we view macrogenetics as a developing field and thus basic expectations regarding observed patterns are still not established. Our study is a step in the direction of establishing this foundational knowledge and suggests avenues for ways to test specific hypotheses. ”

Line 388: “By modeling relationships between environmental data and two complementary measures of intraspecific genetic diversity, GDE and GDM, we can make assemblage-level genetic diversity predictions for data-poor regions of the planet, while flagging and masking those with high uncertainty ¹³⁴ (Fig. 2).”

Author response: We have modified this text to remind readers that the study uses mtDNA GD rather than genetic diversity in general (i.e., genome-wide) (L401-404):

“By modeling relationships between environmental data and two complementary measures of intraspecific genetic diversity, GDE and GDM, we make assemblage-level mitochondrial genetic diversity predictions for data-poor regions of the planet, while flagging and masking those with high uncertainty ¹³¹ (Fig. 2)”

Reviews of reviews

I would still not subscribe that mtDNA diversity, in general, bear *no* information about demographic and evolutionary processes, and, in particular, I do not agree that neutral genetic variation is useless. The authors do not claim to measure adaptive potential, they (with shortcomings that the colleague and myself have highlighted) attempt to test for historical and demographic hypotheses.

Author response: We would further say that we don't take it this far. Our study uses correlative methods, but we do discuss putative historical, evolutionary and demographic hypotheses . We treat genetic diversity as an interesting variable to study on the macroscale, likely shaped by historical processes impacting demography (as well as by selection, and neutral mutational mechanisms). But we do not view it as a population genetic marker for directly testing demographic hypotheses. This is a relatively new field (macrogenetics) and thus basic patterns are still not established. Our study is a step in this direction and suggests avenues for ways to test specific hypotheses.

We agree that it doesn't bear “no information” however the references we supplied show that it bears vary little information in most contexts and is clearly not systematically related to demography or evolutionary processes. It is very clearly not related to genome wide diversity which all claims in the paper hinge on.

I certainly think that neutral genetic diversity IS useful and my primary criticism and

the primary reason it does not relate to demography and evolutionary past well is that mtDNA is NOT neutral. I would be much happier if it was although averaging across species would remain incorrect. mtDNA genes underlie very important respiratory processes. Hints demography and evolutionary pasts will be eradicated when it is averaged across hundred of species that have diverged for millions of years.

The authors do make many claims about adaptive potential. For example:

Line 369: Genetic diversity is critical to the survival of insects and their complex interactions with other organisms. High genetic diversity may facilitate adaptation to changing climates, 123–125 emerging diseases, and pollutants: three (of many) potential drivers of the “insect apocalypse” 50. In addition, genetic diversity contributes to the diversity and stability of species interaction networks by affecting niche space and competition 126, community structure 127, and network complexity 128. At larger ecological scales, insect genetic diversity may reflect ecosystem function and structure as reliably as other traditional macroecological metrics such as species richness 129. It can also augment the resilience of ecosystems that provide continuing services for humankind 14, such as disease management, curbing the spread of invasive plants, aiding sustainable agriculture, pollinating food crops, and controlling pests 13.

These are strong claims that cannot be tested with mtDNA. No study of any single non neutral gene from any part of the genome would acceptably make this claim. I believe Taking the average of such a gene’s across hundreds of species makes even less sense.

In any case, we strongly agree that the limitations brought forward by the referee and the literature she/he cites should be better acknowledged in the study, along with the highlighted technical issues entailed by the use of raw genbank/bold data (e.g. the bias due to the fact that many authors only deposited unique sequences in databanks).

Author response: We have added further discussion of limitations and have now included simulations showing that omission of duplicate alleles has a minor impact on the bias (Supp. Materials; Supp. Fig. 15).

I think this is beter done although the too strong claims remain

I also completely agree on the suggestion to explore Millette et al's approach of considering species identity (which I should have mentioned myself), while it looks to me that the issue of aggregating populations into "arbitrary" units mentioned by Pas-Vinas is more a concern regarding other details of Millette's approach.

Author response: We now show that the vast majority of OTUs are restricted to one grid cell, such that the Millette approach wouldn't work well. We also explain that the grouping into grid cells of arbitrary size is a useful compromise and less of a concern than stated with respect to prediction under the coalescence with migration/isolation or metapopulation. If ranges exceed the grid cell, local sampling is predicted to yield higher GD under a wide range of circumstances (Nordborg and Krone 2001; Charlesworth and Charlesworth 20210, chapter 7

for details), while absolute isolation leading to local coalescence is also not problematic.

The Milette approach would be straightforward to implement and would make the paper acceptable for publication in my opinion.

Already in my original report I made clear that the study requires substantial upgrade before being considered for publication. The concerns raised by the colleague certainly do not improve my judgement, but I would not say that, in principle, I am against publishing any such study just because it is based on mtDNA.

Author response: A point-by-point response to the reviewer's original comments can be found above.

I agree that mtDNA work merits publication. I did before and continue now to encourage the authors to develop hypothesis related to the biology of mtDNA

In general, I partially agree with the other reviewer. In fact, there are other markers or genomic techniques that can be used to study population genetics as the other reviewer has listed, however the manuscript focuses on intraspecific genetic diversity across species at a global scale, which is recently called macrogenomics. In order to do it, the authors used available sequences of *cox1* that covered the world, which is probably the only fragment that is massively sequenced and can be used in a global scale study of intraspecific genetic variation. *Author response: We agree that our use of the potentially problematic mtDNA is a useful compromise that enables the scale of sampling to be done at the macrogenetic scale. Using something like microsatellites mined from the literature would be dwarfed by our study in scale and scope, and also come with its own baggage with regards to relation to neutral or adaptive processes and or rate variation/saturation.*

I (with apologies) reiterate that the volume of data does not overcome the significant limitations with the analytical approach (averaging across species), the marker (which the reviewer notes they agree upon below), and the much too strong inferences made.

Although the other reviewer is right in "mtDNA are not neutral, do not recombine, are not well correlated with demography, are not correlated with neutral genome-wide diversity or adaptive potential", *cox1* provide an accurate species delimitation because the genetic divergence clusters species separately and the cladogenesis is well captured.

We are happy the reviewer agrees but do not see a way to the strong claims made in the paper from this agreement. We agree mtDNA is often good for species delimitation but that is not what the authors of the paper use it for. They use it for a measure of genetic diversity which is highly non standard outside of recent macrogenetics work

Moreover, intraspecific genetic diversity and structure of *cox1* corresponds to biogeography and dispersion (e.g., Papadopoulou et al, ME, 2009; Baselga et al., Nat Comm, 2013; Múrria et al, ME, 2017) and this fragment is widely used in macrogenetics (Leigh et al, Nat Rev Gen, 2021). Unfortunately,

data available for neutral markers such as microsat or the mt genome of *Drosophila* that the other reviewer suggested, is limited and currently it is available for a small number of species, but the data available is far for a global scale study. Moreover, I have my doubts if microsat data can be easily compared across species because the markers used are commonly specific for each species. Until genome information is not available for thousands of species at the population level, i.e., several individuals sequenced for each population, we cannot use mt genome sequencing in macrogenetics. Also the authors are not clearly interested in adaptive genetics or general genetic diversity, as the other reviewer indicated, instead authors want to test macroecological patterns of intraspecific genetic variation for as much number of species as possible covering the widest area. For this reason, authors specifically analyzed the intraspecific DNA barcode diversity, which is clear in the manuscript. *Author response: As mentioned above, we largely agree with all of this. As the reviewer describes, we use the only data source with the geographic and taxonomic scale available to answer this question. While not perfect, is the standard approach in macrogenetics thus far in other taxa and provides a baseline for future work.*

I maintain that mtDNA is interesting but it's treatment in this study and the inferences drawn (as with previous studies) are inappropriate. The authors very clearly make claims about adaptive potential directly and indirectly when they cite the conservation relevance of their marker.

2. Do you think the mtDNA data analyzed are appropriate for addressing the questions posed? Despite the mtDNA data analyzed has limitations, I think the data used can address the questions posted and the study is well-performed. However, assuming the limitations of *cox1* data, I suggest authors rephrase the hypothesis based on the comments by the other reviewer. In fact, some of the hypothesis based on demographic processes cannot be directly tested using the available data. Improving this section is critical. As the other reviewer, I agree in "I suspect that there are many interesting processes related to the biology of mtDNA underlying some of the patterns detected." I may focus the new hypothesis on these processes.

Author response: As previously mentioned, we don't actually test these hypotheses via inferences of underlying processes, but instead we conduct a correlative study using a phenomenological model and discuss possible interpretation and processes that could be more directly tested via process-explicit models in future studies. In other words, our study does not really test hypotheses, but rather it generates them (Pilowsky et al. 2022 for a discussion of the distinction between these two types of modeling approaches).

I just don't see this distinction as clearly communicated in the paper or doing the argumentative work that the authors think it does. If the data is unsuited to testing the hypotheses why would it be suited to generating them. They are poor measures of all the quantities of interest at best as is agreed upon by all reviewers and the authors. I have argued strongly with references that they are worse than poor measures. The support for the approach seems to be related to data availability and not appropriateness.

Another interesting criticisms by the other reviewer is the metric used: mean GD in each grid. Authors must to argue extensively why this metric is appropriate, and I think it is.

Moreover, authors can explain if they used the abundance of each haplotype in a grid or only the data of each unique sequences, which can introduce bias as the other reviewer suggested, specially when authors used the mean GD. This is also critical.

Author response: In the manuscript we have explained that we use a classic metric (average pairwise differences) that is robust to small sample sizes (unlike allelic richness) and now provide simulation experiments showing that this metric is also not strongly biased by the removal of duplicate alleles (Supp. Materials, Supp. Fig. 15) We have also addressed these two issues directly with other reviewers.

As noted above the average of this metric taken across species is not classic and not demonstrated to be robust.

Reviewer 4

Reviewer #4 (Remarks to the Author):

After carefully reading the main text and the revision, I can see an important improvement of the manuscript. Now it is clearer and well addressed and it can be an important contribute to understand large scale pattern of intraspecific genetic diversity of insects. The patterns found are relevant for advancing this discipline and understanding global patterns of diversity.

As required for the reviewers, authors have managed to improve the description of the methodology, which now is clearer than previous versions. Specially, criteria for sequence selection is much clearer than the previous version and the statistical analyses used for validating their datasets are robust and well explained.

Authors' response: *We thank the reviewer for their positive feedback and helpful suggestions.*

Minor revision:

- Lines 254-256: I cannot see the relations between the lack of correlation with GDE and GDM, and the departure from the expectations of SGDC, because the number of OTUs are not considered in the current study.

Authors' response: *Thanks for pointing this out, we skipped a step in presenting our logic. We have modified the text to read (L248-258):*

“The negative quadratic latitudinal correlation of GDE in two of the three most sampled orders (Diptera and Lepidoptera) and the lack of an overall correlation with GDM suggests a departure from expectations of species genetic diversity correlation (SGDC) predictions, given the expected negative linear correlation of species richness and latitude 32,83–85. However, many confounding factors could affect how species diversity metrics relate to GDM and GDE, and these factors may have both positive and negative effects, leading to large variation in the direction and strength of SGDCs 86, especially at a global scale in such a large taxonomic

group such as insects. Sampling biases, especially for the overrepresented North American and European regions, may also influence this relationship, although we find no evidence of a correlation between sampling effort and the two GD metrics in our data (Supplementary Fig. 14, Supplementary Table 5).”

Lines 267-270: I may remove these two sentences from “Macrogenetics is a relatively new...”. There are several expectations well established, however we are still dealing how to test them.

Authors’ response: *We agree that this was confusing, so we have removed these sentences.*

- Legend in figure 1 looks too llong, I suggest cut here

Authors’ response: *We appreciate the suggestion. We decided to keep these descriptions but make edits to tighten the prose.*

- Check reference 11.

Authors’ response: *Thank you, we have fixed the formatting.*

REVIEWERS' COMMENTS

Reviewer #1 (Remarks to the Author):

First of all, I apologize for taking longer than expected in submitting this review.

I appreciate the authors' new effort to improve the structure of the manuscript.

In particular, the introduction, which I suggested to modify, now reads really well and is definitely more streamlined.

Honestly, I still find that the "complementary" value of GDM and GDE is not actually explained there, though. One may read that both are expected to positively correlate with the same predictor (although I reckon that the references cited about these expectations actually had GDM in view, rather than GDE). If so, we should expect that GDM and GDE also correlate with each other and, ultimately, have little "complementarity".

More in general, I find that the back-and-forth between GDM and GDE, especially in the discussion, often lacking a clear distinction about which processes should affect one or the other, may still result in a bit of confusion for some readers and there is space for some little change.

I would, for example, suggest reducing the space given in the discussion to the analysis of latitudinal patterns to focus on the (in my opinion) more relevant, ecologically relevant predictors.

I don't want to insist too much on this, though, and, all in all, I think the manuscript is now fit for publication.

I add some more detailed comments below which I hope may be still useful for some final touches.

Authors' Responses

Authors' response:

We agree that the logic of testing the "freeze" variable separately was not made clear. It was highly correlated with other climate variables in the predictor set of the full model and we now state this clearly (L559-561): "We tested this variable separately from the full predictive model because it is highly collinear with the deep-time temperature variability metric ($r > 0.75$) included in the model."

My comment:

I am still not really convinced that it is a great idea to have it tested at all if it is strongly correlated with other predictors. It is much like testing the same thing twice in different ways.

By the way, I think that testing temperature in a similar way is somehow more meaningful just because we all *take for granted* that latitude *per se* is nothing but a proxy for actual explanatory variables.

Abstract

L30: replace "that" with ", which".

L36-37: I suggest "*positively* correlate".

Introduction

L 120: Probably, writing "GDM and GDE were calculated from *COI sequences of* insects native to.." would make the sentence easier to read.

L 120-123: Why not break the sentence in two?

L 133: I am not sure whether this average also includes those cells with no samples at all. It might be good to make it more clear.

L 140-147: Is it **absolute** latitude? Clarify.

L141 (and L 154): I think it should be "**spatially** modified t-test".

L142-143: Although the test does not assume a dependent and an independent variable, this sentence sounds really bad to me... Why not "correlation of GDM with either absolute latitude or squared latitude was not significant"? In my view, it would also make clear that this test is **not** a linear model where one may have a linear **and** a quadratic term for latitude. It took me a while to figure it out!

Also, I am very surprised that these results were so different between GDM and GDE given that, as far as one can see in Fig. 2, their latitudinal trend is visually similar..

Discussion

L 235: Why is this relation so surprising? Were there any specific predictions?

L 256-257: It is not very clear. I guess the bias may be generated by N America and Europe having idiosyncratic patterns wrt to Asia (the only other continent extending in the northern hemisphere), rather than from a specific relation between.

L 264-277: Here is a good example of the potential confusion between GDM and GDE... The authors, very reasonably, say that larger ranges may be associated with higher ***genetic diversity***. They also suggest that "consistently higher values of GD ... aggregate to higher GDE", which also makes sense. Why, then, should we only observe the latitudinal pattern in GDE and not in GDM? The explanation suggested in the following lines seems rather weak and gets back to the point I raised above. If GDM and GDE are highly correlated, they are not complementary. One should rather attempt to explain this very strong correlation...

L278: Replace "If" with "Although", it will clarify that the sentence does not imply a condition-consequence relationship.

L306-307: It is a bit disappointing that there is no attempt to explain this, which is the only instance where GDM and GDE diverge..

L313-323: It seems like the role of range size has been already discussed at L266 and following..

L326-328: Isn't it a bit weird to make **predictions** regarding the analysis at hand within the Discussion?

L334: Climate stability predicts stable demography for each species, which may translate in higher GD **relative to other populations of the same species**. I am strongly convinced that high GDW (and low GDM) should be predicted in very unstable climates (where all populations result from recent colonization). Please revise this sentence.

L335-337: Good point. Is it possible that GDE is mostly driven by how taxonomically homogeneous the sampling in each grid cell is?

L381-383. I do not understand this sentence. Why "however"? How does it relate to the previous?

Reviewer #2 (Remarks to the Author):

I very much appreciate the author's very hard work in addressing my concerns. I think this is a much better focus for the paper. The caveats are upfront and easy for readers to see, and the justification for proceeding despite those caveats is mostly good.

While I have ongoing concerns I think the reader can better judge those concerns for themselves with this version so I'll leave most of those out. I have three remaining thoughts at this point.

Line 356 - "Genetic diversity is critical to the survival of insects and their complex interactions with climates 112–114. High genetic diversity may facilitate adaptation to changing, emerging diseases, and pollutants: three (of many) potential drivers of the "insect apocalypse" 58. In addition, genetic diversity contributes to the diversity and stability of species interaction networks by affecting niche space and competition 115, community structure 116, and network complexity 117. At larger ecological scales, insect genetic diversity may reflect ecosystem function and structure as reliably as other traditional macroecological metrics such as species richness 118. It can also augment the resilience of ecosystems that provide continuing services for humankind 119, such as disease management, curbing the spread of invasive plants, aiding sustainable agriculture, pollinating food crops, and controlling pests 120.

I would delete this paragraph. As we have discussed at length mtDNA is not correlated with genome-wide diversity and does not come into play with most of what is mentioned here. It has no role in adapting to disease, pollution is a stretch--certainly in general across this many species, maybe climate a bit but that is not made clear. mtDNA clearly has nothing to do with species interactions, community structure or any kind of network complexity. Resilience of ecosystems and ecosystem services is also highly speculative.

Line 451- By modeling relationships between environmental data and two complementary measures of intraspecific genetic diversity, GDE and GDM, we make assemblage-level mitochondrial genetic diversity predictions for data-poor regions of the planet, while flagging and masking those with high uncertainty (Fig. 2). These genetic diversity maps have the potential to fill a knowledge gap that far exceeds the undersampling and taxonomic uncertainties underlying vertebrate and plant macroecological studies 132,133. They can also highlight genetic diversity as an important biodiversity component that has yet been assessed for relatively few taxa 17, while focusing attention on a data-deficient group with evidence of global population declines and strong connections to ecosystem functions and services 134. Taken together, GDM and GDE are promising biodiversity metrics for documenting and understanding 410 "the little things that run the world" 135.

I would also delete this. I think the authors have done a good job advocating that they have done good work with a bad data availability situation. I would not then advocate for promise of this approach. I think deleting is best but if anything advocate for advances and not repetition.

I also remain concerned about GDE. I understand that it has been published previously but the response reinforces my concern.

"Authors' response: Our approach for calculating GDE works by dividing the exponential of Shannon's entropy by the number of OTUs (i.e. "species") for a cell, which controls for species richness and is a major reason why we use GDE. Additionally, we confirm this expectation in Supplementary Fig. 16 and Supplementary Table 5, showing that the number of OTUs in a cell does not significantly correlate with GDE or GDM. Given these pieces of information, we decided to leave this variable out of our model."

I'll reiterate that this is my concern. "evenness necessarily decreases with increasing numbers of OTUs in a cell.". Species richness can drive these results and the author responses do not suggest otherwise.

again thank you for humouring my comments seriously and sincerely. They have been extensive, I

know. While I'll choose to remain anonymous, I will buy the authors a tea or beer at any conference I see them at. Good luck with this and everything.

Reviewer #1 (Remarks to the Author):

First of all, I apologize for taking longer than expected in submitting this review.

I appreciate the authors' new effort to improve the structure of the manuscript.

In particular, the introduction, which I suggested to modify, now reads really well and is definitely more streamlined.

Authors' response: Thank you, we appreciate your feedback to improve the text.

Honestly, I still find that the "complementary" value of GDM and GDE is not actually explained there, though. One may read that both are expected to positively correlate with the same predictor (although I reckon that the references cited about these expectations actually had GDM in view, rather than GDE). If so, we should expect that GDM and GDE also correlate with each other and, ultimately, have little "complementarity".

Authors' response: In theory, the two metrics should contain somewhat independent information, since GDM describes the average of the distribution, while GDE describes the shape of the distribution. However, we acknowledge that in practice the two are correlated (at least in our instance) because the average is influenced by zero-weighted distributions of genetic diversity, leading to many instances of low GDM distributions reflected by low GDE. We toned down the language by altering the following language in the Introduction:

"GDM and GDE present broadly complementary information when considered together..." (L109-110).

In addition, we removed the reference to complementarity in the Discussion:

"By modeling relationships between environmental data and two measures of intraspecific genetic diversity..." (L425-426).

More in general, I find that the back-and-forth between GDM and GDE, especially in the discussion, often lacking a clear distinction about which processes should affect one or the other, may still result in a bit of confusion for some readers and there is space for some little change.

I would, for example, suggest reducing the space given in the discussion to the analysis of latitudinal patterns to focus on the (in my opinion) more relevant, ecologically relevant predictors.

I don't want to insist too much on this, though, and, all in all, I think the manuscript is now fit for publication.

Authors' response: We realize that our switching between using the two community-wide summary statistics (GDE and GDM) and the per-species/OTU summary GD likely has contributed to this confusion, as "GD" has been used to refer to GDM in previous macrogenetics

literature. To mitigate this, we went through the Discussion and referred to any mention of “GD” as “per-species GD” or “per-OTU GD” depending on the context. This should clarify our Discussion to the reader, as we consistently refer to GDE in the description of expected community-wide patterns, except in a few instances where we mention the lack of correlations with GDM or the relationship between GDE and GDM.

I add some more detailed comments below which I hope may be still useful for some final touches.

Authors' Responses

Authors' response:

We agree that the logic of testing the “freeze” variable separately was not made clear. It was highly correlated with other climate variables in the predictor set of the full model and we now state this clearly (L559-561): “We tested this variable separately from the full predictive model because it is highly collinear with the deep-time temperature variability metric ($r > 0.75$) included in the model.”

My comment:

I am still not really convinced that it is a great idea to have it tested at all if it is strongly correlated with other predictors. It is much like testing the same thing twice in different ways.

By the way, I think that testing temperature in a similar way is somehow more meaningful just because we all *take for granted* that latitude *per se* is nothing but a proxy for actual explanatory variables.

Authors' response: We acknowledge the statistical redundancy in analyzing the two patterns separately, but believe testing the freeze-line variable separately is warranted to allow for direct comparisons with existing literature (e.g. insect diapause and macroecological studies such as White et al. 2019, Nat. Comm.) and to intuitively present a geographical pattern that is not immediately evident with our analysis with large numbers of continuous explanatory variables. To make the statistical redundancy clear for the reader, we added this text to the Methods section, (L585-588):

“Given this correlation, we acknowledge the redundancy in the statistical tests, but we believe that including two perspectives on the geographical structure of genetic diversity allows for direct comparison with existing patterns in the literature that are not apparent otherwise.”

Abstract

L30: replace “that” with “, which”.

Authors' response: Fixed, thanks.

L36-37: I suggest “*positively* correlate”.

Authors’ response: Fixed, thanks.

Introduction

L 120: Probably, writing “GDM and GDE were calculated from *COI sequences of* insects native to..” would make the sentence easier to read.

Authors’ response: Agreed, fixed!

L 120-123: Why not break the sentence in two?

Authors’ response: We agree with the suggestion and the sentence now reads like this:
“GDM and GDE were calculated from COI sequences of insects native to the 193 km x 193 km equal-area resolution raster grid cells in which they were sampled. To account for the potential impact of sampling biases, six thresholds for the minimum number of OTUs per grid cell (10, 25, 50, 100, 150, and 200 OTUs) were considered.”

L 133: I am not sure whether this average also includes those cells with no samples at all. It might be good to make it more clear.

Authors’ response: Thanks for pointing this out. The sentence now reads:
“On average, each sampled cell included ten insect orders, 689 OTUs, and 9,859 individuals.”

L 140-147: Is it *absolute* latitude? Clarify.

Authors’ response: we agree this is not clear. We changed the first reference to “Absolute latitude” to clarify.

L141 (and L 154): I think it should be “*spatially* modified t-test”.

Authors’ response: agreed, we changed it in both places. Thanks for the catch.

L142-143: Although the test does not assume a dependent and an independent variable, this sentence sounds really bad to me... Why not “correlation of GDM with either absolute latitude or squared latitude was not significant”? In my view, it would also make clear that this test is *not* a linear model where one may have a linear *and* a quadratic term for latitude. It took me a while to figure it out!

Authors’ response: We agree that the paragraph was not clear about the type of correlation being conducted. We changed the language to reflect your suggestions:
“Absolute latitude was not significantly linearly correlated with GDE ($p_{100} = 0.064$), but squared latitude had a negative relationship with GDE across the globe at $\alpha = 0.05$ (spatially modified t-

test, two-sided; Fig. 2; Table 1; $r_{100} = -0.360$; $p_{100} = 0.022$). In contrast to GDE, neither absolute latitude ($p_{100} = 0.924$) nor squared latitude had a significant correlation with GDM (Fig. 2; Table 1; $p_{100} = 0.767$).

When considering the top three most sampled orders independently, we found that squared latitude had a negative relationship with GDE in Diptera ($r_{100} = -0.468$; $p_{100} = 0.001$) and Lepidoptera ($r_{100} = -0.360$; $p_{100} = 0.043$), but not in Hymenoptera ($p_{100} = 0.352$) (Fig. 3). Absolute latitude did not significantly vary with GDM in all orders (Fig. 3).”

Also, I am very surprised that these results were so different between GDM and GDE given that, as far as one can see in Fig. 2, their latitudinal trend is visually similar..

Authors' response: Although there does appear to be a similar shape, the noise around the relationship between GDM and latitude was much stronger than GDE and latitude and much of the signal in the latitude and GDM relationship is driven by spatial autocorrelation.

Discussion

L 235: Why is this relation so surprising? Were there any specific predictions?

Authors' response: We agree that it is not so much surprising as a deviation from the typical decrease from the tropics to the poles. We removed the word “surprising” from the text.

L 256-257: It is not very clear. I guess the bias may be generated by N America and Europe having idiosyncratic patterns wrt to Asia (the only other continent extending in the northern emisphere), rather than from a specific relation between.

Authors' response: We clarified our expectation with respect impacts on modeling with the following text:

“Sampling biases, especially for the overrepresented North American and European regions, may also influence this relationship by overfitting models towards patterns present in these regions, although we find no evidence of a correlation between sampling effort and the two GD metrics in our data”

L 264-277: Here is a good example of the potential confusion between GDM and GDE... The authors, very reasonably, say that larger ranges may be associated with higher **genetic diversity**. They also suggest that “consistently higher values of GD ... aggregate to higher GDE”, which also makes sense. Why, then, should we only observe the latitudinal pattern in GDE and not in GDM? The explanation suggested in the following lines seems rather weak and gets back to the point I raised above. If GDM and GDE are highly correlated, they are not complementary. One should rather attempt to explain this very strong correlation...

Authors' response: We agree that if GD increases with larger range sizes, then both GDM and GDE should increase. We think the statistical noise inherent in GDM is higher than it is in GDE because outlier values in per-OTU GD (i.e. extremely high or low values) will impact the

estimation of GDM more than GDE. For an illustration of this idea, take this example distribution of genetic diversities across a community:

[0.01, 0.01, 0.005, 0.0]

The GDM of this community is 0.00625, while the GDE is 0.7179365.

If we replace the first 0.01 GD value with 0.1, the distribution is now

[0.1, 0.01, 0.005, 0.0]. In this second case, the GDM is 0.02875, while the GDE is 0.4000929.

This is a 4.6-fold change in GDM, while only a 1.79-fold change in GDE. Given random sampling error of GD in the community, GDM will be a much noisier metric of diversity versus GDE.

To communicate this expectation, we added the following text to the Discussion, following Discussion of Rappoport's Rule and range size correlations (L282-286:

"Furthermore, the inflation of GD in a larger portion of OTUs within a grid-cell serves to increase evenness (GDE) by bridging the difference in GD between high GD and low GD OTUs within the sampled assemblage. This should also lead to higher GDM, but the presence of OTUs with extremely high or extremely low GD are more likely to lead to more statistical noise for the calculation of GDM than GDE."

L278: Replace "If" with "Although", it will clarify that the sentence does not imply a condition-consequence relationship.

Authors' response: Agreed, changed.

L306-307: It is a bit disappointing that there is no attempt to explain this, which is the only instance where GDM and GDE diverge..

Authors' response: We agree that this result deserved more attention. We added additional text to explain a potential scenario that could lead to this result (L322-334):

"Moreover, GDM and GDE have contrasting relationships with seasonally high precipitation (PWM), where GDM increases with PWM, while GDE decreases. The former corresponds with wet, hot regions, while GDE is highest where it is arid and hot (Fig. 2, 4).

GDM's correlation with high MTWM and PWM, low seasonal variation in precipitation (precipitation seasonality) and high long-term climate stability (precipitation variation, precipitation trend, and temperature trend) is reflected by predicted hotspots in the most long-term stable and least seasonal tropical or subtropical forest habitat (Supplementary Fig. 4e, Supplementary Fig. 5). In addition to increased evolutionary speed, the high resource availability and stability of these areas may allow for population persistence and the consequent accumulation of genetic diversity. In contrast, GDE's peak in subtropical, hot, and arid environments could be partially driven by processes related to geographic patterns in physiological tolerances that lead to both the sizes and uniformity of species ranges...."

L313-323: It seems like the role of range size has been already discussed at L266 and following..

Authors' response: while we did explain the general role of range sizes, we took the opportunity to relate the climatic relationships with insect physiology, which indirectly impact range size dynamics. While this does continue the range size discussion, we feel that relationships between insect physiology, and by way of their impact on niche and range sizes, genetic diversity, are important to discuss.

L326-328: Isn't it a bit weird to make *predictions* regarding the analysis at hand within the Discussion?

Authors' response: We agree that this is not appropriate for the Discussion and decided to remove the sentence.

L334: Climate stability predicts stable demography for each species, which may translate in higher GD *relative to other populations of the same species*. I am strongly convinced that high GDW (and low GDM) should be predicted in very unstable climates (where all populations result from recent colonization). Please revise this sentence.

Authors' response: We acknowledge the reviewer's good intuition about what we could predict, but their suggested interpretation would directly contradict our evidence that both GDE and GDM increase with climate stability. Therefore, we elected to keep our original interpretation.

L335-337: Good point. Is it possible that GDE is mostly driven by how taxonomically homogeneous the sampling in each grid cell is?

Authors' response: Thank you for bringing this up! We found that GDE is not significantly correlated with the number of orders (the best measure of taxonomic homogeneity we have available) with a spatially modified t-test and a p-value of 0.08.

L381-383. I do not understand this sentence. Why "however"? How does it relate to the previous?

Authors' response: Agreed, our use of "however" was inappropriate. We removed it so the connection between the two sentences is more obvious.

Reviewer #2 (Remarks to the Author):

I very much appreciate the author's very hard work in addressing my concerns. I think this is a much better focus for the paper. The caveats are upfront and easy for readers to see, and the justification for proceeding despite those caveats is mostly good.

While I have ongoing concerns I think the reader can better judge those concerns for themselves with this version so I'll leave most of those out. I have three remaining thoughts at this point.

Authors' response: We thank the reviewer for the recommendations and agree that this resulted in useful improvement.

Line 356 - "Genetic diversity is critical to the survival of insects and their complex interactions with climates 112–114. High genetic diversity may facilitate adaptation to changing, emerging diseases, and pollutants: three (of many) potential drivers of the "insect apocalypse" 58. In addition, genetic diversity contributes to the diversity and stability of species interaction networks by affecting niche space and competition 115, community structure 116, and network complexity 117. At larger ecological scales, insect genetic diversity may reflect ecosystem function and structure as reliably as other traditional macroecological metrics such as species richness 118. It can also augment the resilience of ecosystems that provide continuing services for humankind 119, such as disease management, curbing the spread of invasive plants, aiding sustainable agriculture, pollinating food crops, and controlling pests 120.

I would delete this paragraph. As we have discussed at length mtDNA is not correlated with genome-wide diversity and does not come into play with most of what is mentioned here. It has no role in adapting to disease, pollution is a stretch--certainly in general across this many species, maybe climate a bit but that is not made clear. mtDNA clearly has nothing to do with species interactions, community structure or any kind of network complexity. Resilience of ecosystems and ecosystem services is also highly speculative.

Authors' response: We are not implying that mtDNA itself has any causal input to any of these things and we already acknowledged that it contains limited (but still potential) signal for some of these patterns and processes. As this is the Discussion section, we choose to discuss the importance of genetic diversity in the broad sense to potentially inspire ambitious initiatives such as the Earth Biogenome Project. To make our emphasis on genetic diversity in general more clear, we added the following text to the beginning of the paragraph:

"While mtDNA genetic polymorphism is only one component of genome-wide genetic diversity, the latter is critical to the survival of insects and their complex interactions with other organisms 112–114."

We believe these components are still important to discuss and left the remainder of the paragraph in the paper.

Line 451- By modeling relationships between environmental data and two complementary measures of intraspecific genetic diversity, GDE and GDM, we make assemblage-level mitochondrial genetic diversity predictions for data-poor regions of the planet, while flagging and masking those with high uncertainty (Fig. 2). These genetic diversity maps have the potential to fill a knowledge gap that far exceeds the undersampling and taxonomic uncertainties underlying vertebrate and plant macroecological studies 132,133. They can also highlight genetic diversity as an important biodiversity component that has yet been assessed for relatively few taxa 17, while focusing attention on a data-deficient group with evidence of global population

declines and strong connections to ecosystem functions and services 134. Taken together, GDM and GDE are promising biodiversity metrics for documenting and understanding 410 “the little things that run the world” 135.

I would also delete this. I think the authors have done a good job advocating that they have done good work with a bad data availability situation. I would not then advocate for promise of this approach. I think deleting is best but if anything advocate for advances and not repetition.

Authors' response: We understand the difference of opinion but choose to keep this text and now add that this paragraph refers to mitochondrial genetic diversity instead of genetic diversity in general. Also, using GDM and GDE together as we have is in no way limited to mtDNA data as it could be extended to summaries of the genetic diversity of any marker (or even at the whole-genome level) so we retain the statement that these metrics have potential utility.

I also remain concerned about GDE. I understand that it has been published previously but the response reinforces my concern.

"Authors' response: Our approach for calculating GDE works by dividing the exponential of Shannon's entropy by the number of OTUs (i.e. "species") for a cell, which controls for species richness, This is a major reason why we use GDE. Additionally, we confirm this expectation in Supplementary Fig. 16 and Supplementary Table 5, showing that the number of OTUs in a cell does not significantly correlate with GDE or GDM. Given these pieces of information, we decided to leave this variable out of our model."

I'll reiterate that this is my concern. "evenness necessarily decreases with increasing numbers of OTUs in a cell.". Species richness can drive these results and the author responses do not suggest otherwise.

*Authors' response: This is probably not intuitive, as it might seem that logically a statistic will get smaller if the denominator (# of OTUs in this case) gets larger. But this is not exactly what is happening here and we should have done a better job explaining it in our original response. The exponent of Shannon's entropy of genetic diversity incorporates species/OTU richness by summing $-\pi * \ln(\pi)$ across all species/OTUs and taking the exponent of that sum, where π is the average number of pairwise nucleotide differences within a species/OTU. Since this is necessarily dependent on the number of species/OTUs, where an increasing number of species/OTUs will lead to increasing evenness, we divide by the number of species/OTUs that are being summed, which effectively removes this bias. If we arbitrarily divide by increasing numbers of OTUs in the cell, then we would expect the evenness to decrease, but increasing the number of OTUs in the cell would also alter the numerator and not necessarily lead to decreased evenness. To confirm this intuition, we further show in our Supplementary Fig. 16 and Supplementary Table 5 the number of OTUs in a cell does not significantly correlate with GDE or GDM.*

again thank you for humouring my comments seriously and sincerely. They have been extensive, I know. While I'll choose to remain anonymous, I will buy the authors a tea or beer at any conference I see them at. Good luck with this and everything.

Authors' response: We are grateful that the reviewer provided useful recommendations that helped improve the manuscript.